

# Data-Model Comparisons of the Tropical Hydroclimate Response to the 8.2 ka Event with an Isotope Enabled Climate Model

Andrea L. Moore[1], Alyssa R. Atwood[1], and Raquel E. Pauly[1]

[1]Department of Earth, Ocean, and Atmospheric Science, Florida State University, Tallahassee, Florida, USA

**Correspondence:** Andrea L. Moore (andilee.moore@gmail.com) and Alyssa R. Atwood (aatwood@fsu.edu)

**Abstract.** The 8.2ka Event, a prominent climate anomaly that occurred approximately 8,200 years before present (8.2ka), has been the subject of extensive research due to its potential implications for understanding the characteristics and mechanisms of abrupt climate change events. We characterize the tropical hydroclimate response to the 8.2ka Event based on a multiproxy compilation of 61 tropical hydroclimate records and assess the consistency between the reconstructed hydroclimate changes and those simulated by a new isotope-enabled climate model simulation of the 8.2ka Event with iCESM. The timing and duration of the hydroclimate anomalies is calculated using two event detention methods, one of which uses a new changepoint detection algorithm to account for age uncertainty. When age uncertainties are explicitly accounted for, significant hydroclimate anomalies associated with the 8.2ka Event are detected in 30% of the records in the compilation, with a mean onset age of $8.28\pm0.12$ka ($1\sigma$), mean termination age of $8.11\pm0.09$ka ($1\sigma$), and mean duration of $152\pm70$ years ($1\sigma$; with a range of 50-289 years), comparing well with previous estimates, and lending support to a regionally-variable tropical hydroclimate response to the 8.2ka Event, with events that span decadal to multi-centennial timescales in the proxy record. Notably, the hydroclimate anomalies are not hemispherically uniform, but rather display rich regional structure. Anomalous conditions are characterized by pronounced isotopic enrichment across East Asia, South Asia, and the Arabian Peninsula. In the Americas, drying and isotopic enrichment occurred in southern Central America, contrasting with isotopic depletion in central/eastern Brazil. In contrast, no robust signatures of the 8.2ka Event were found over the Maritime Continent. Many of these regional patterns generally agree with the new set of iCESM simulations of the 8.2ka Event. In iCESM, the North Atlantic meltwater forcing leads to a broad southward shift in tropical rainfall, resulting in a generally drier Northern Hemisphere and wetter Southern Hemisphere, but with large regional variations in precipitation amount and the isotopic composition of precipitation. Over the oceans, the precipitation $\delta^{18}$O anomalies are generally consistent with the "amount effect", wherein areas characterized by drying have more isotopically enriched precipitation and areas of wetting have more isotopically depleted precipitation. However, the precipitation $\delta^{18}$O anomalies are more decoupled from changes in precipitation amount over land. iCESM captures many of the regional hydroclimate responses observed in the reconstructions, including the large-scale isotopic enrichment pattern in precipitation $\delta^{18}$O in South and East Asia and the Arabian Peninsula, drying and isotopic enrichment in precipitation $\delta^{18}$O in southern Central America, isotopic depletion in parts of northeastern South America, and a muted hydroclimate response in the Maritime Continent. Overall, this study provides new insights into the tropical hydroclimate response to the 8.2ka Event, emphasizing the importance of accounting for age uncertainty in the hydroclimate reconstructions and the value of using isotope-enabled model simulations for data-model intercomparison.



## 1 Introduction

The tropics play a fundamental role in Earth's climate variability, acting as a heat source that drives global weather patterns
via complex atmospheric teleconnections. A key component of the tropical climate system is the Intertropical Convergence
Zone (ITCZ). From a zonal mean perspective, the ITCZ is the ascending branch of the Hadley cell and is characterized by the
convergence of low-level trade winds, ascent, and heavy precipitation near the equator. On a regional basis, ITCZs exist over
the Atlantic and eastern Pacific Ocean, where strong strong sea surface temperature (SST) gradients drive convergence, ascent,
and narrow, well-defined rainbands. Distinct processes govern the large-scale circulation and precipitation in other regions
of the tropics like monsoon systems and the Indian Ocean. Throughout the tropics, rainfall patterns migrate on a seasonal
basis, following the warmer hemisphere. The migrations are regionally variable, with the Atlantic and Pacific ITCZs migrating
between 9°N and 2°N in boreal summer/fall and winter/spring, respectively, while rainfall over the Indian Ocean and adjacent
land masses swings more dramatically between 20°N and 8°S (Schneider et al., 2014). These fluctuations drive distinct wet
and dry seasons through many regions of the tropics, providing critical access to water for roughly 40% of Earth's population
(Penny, 2021). As the tropics comprise some of the most densely populated areas on Earth, it is critical to understand how
tropical precipitation patterns may change in the near future. However, there is currently no agreement across models on how
the ITCZ and monsoons will change with continued greenhouse gas forcing (Biasutti et al., 2018; Geen et al., 2020), in part
due to persistent biases in the representation of the tropical mean state in global climate models (Li & Xie, 2014). Therefore,
improving our understanding of how tropical rainfall patterns respond to external forcing is a critical target in the climate
modeling community.

Our ability to make robust predictions about the climate system is also limited by the relatively short instrumental record.
Paleoclimate proxy records extend the observational record beyond the instrumental era, helping to illuminate the linkages
between climate forcings and the response of the climate system to forcing. Such data are critical for ground-truthing climate
models to observations outside of the short period of direct observational data.

Past periods of abrupt climate change are especially important to examine in the context of evaluating future climate change
risk, as we have no modern analogue with which to compare these events and we cannot rule out the possibility of such abrupt
events in the future. Evidence from paleoclimate records (Arbuszewski et al., 2013; Koutavas & Lynch-Stieglitz, 2004; Rhodes
et al., 2015) and model simulations of past climates (Chiang & Bitz, 2005; Roberts and Hopcroft, 2020) suggest that the
location of the tropical rain bands may have shifted significantly and abruptly in the past (upwards of 7° latitude in certain
regions) associated with changes in ice sheet extent and meltwater forcing (e.g., during Heinrich Events). The most recent such
period of rapid, global climate reorganization occurred approximately 8,200 years before present day (the 8.2ka Event; Alley
et al., 1997) and is thought to have lasted over a period of 100-200 years based on oxygen isotopic data from Greenland ice
cores and tropical speleothems (Morrill et al., 2013). This period occurred during the otherwise stable Holocene epoch (11,700
years ago to present) and was driven by the discharge of around 163,000 km$^3$ of meltwater from proglacial Lakes Ojibway
and Agassiz (remnants of the Laurentide Ice Sheet) into the North Atlantic, which triggered a large-scale salinity anomaly and
resultant reduction in the strength of the Atlantic Meridional Overturning Circulation (AMOC; e.g., Barber et al., 1999; Ellison



et al., 2006). Proxy data and dynamical theory (e.g., Kang et al., 2008; Kang et al., 2009; Schneider et al., 2014) link this event to widespread cooling of the Northern Hemisphere (1-6°C; e.g., Ellison et al., 2006; Kobashi et al., 2007) and an associated southward shift of tropical rainfall patterns, with hydroclimate anomalies lasting anywhere from decades to centuries (e.g.,

Rohling and Palike, 2005; Morrill et al., 2013).

Morrill et al. (2013) published the most recent multiproxy compilation of high-resolution paleoclimate data related to the 8.2ka Event, incorporating 262 paleoclimate records from 114 global sites. Their synthesis demonstrated a regionally-variable hydroclimate response to the 8.2ka Event. Some of the robust regional hydroclimate changes included drying in Greenland, the Mediterranean, the Maritime Continent (Ayliffe et al., 2013; Chawchai et al., 2021), and across Asia (Wang et al., 2005;

Dykoski et al., 2005; Cheng et al., 2009; Liu et al., 2013); while wetter conditions prevailed over northern Europe, Madagascar (Voarintsoa et al., 2019), and northeastern South America (Aguiar et al., 2020). Together, these data provide evidence for an anti-phased hemispheric precipitation response, with a strengthening of the South American summer monsoon (SASM) and a weakening of the Asian (AM) and East Asian summer monsoons (EASM).

Building on this work, Parker and Harrison (2022) used a statistical technique called breakpoint analysis to identify the

75 timing, duration, and magnitude of the 8.2ka Event in 73 high-resolution, globally-distributed speleothem $\delta^{18}$O records from the Speleothem Isotope Synthesis and Analysis database (SISALv2; Comas-Bru et al., 2020). They identified significant isotopic excursions near 8.2ka in over 70% of their records and determined a median duration of global hydroclimate anomalies of approximately 159 years. Parker and Harrison (2022) inferred several regionally-coherent tropical hydroclimate anomalies from their synthesis, based on broad patterns of isotopic depletion across South America and southern Africa and isotopic

enrichment in Asia, from which they inferred a weakening of Northern Hemisphere monsoons, strengthening of Southern Hemisphere monsoons, and a mean southward shift of the ITCZ as the most plausible mechanism for transmitting the effects of the 8.2ka Event throughout the tropics.

There are several limitations to these studies which are addressed in the updated proxy compilation presented here. Chiefly, Morrill et al. (2013) rely upon an a priori event window in classifying the climate response to the 8.2ka Event, and do not take

radiometric age uncertainty of the proxy records into account. While Parker and Harrison (2022) consider the effects of age uncertainties on their compilation, they did not propagate these uncertainties through their breakpoint analyses. Neither study includes comparisons with isotope-enabled climate model simulations, a critical tool for validating proxy record interpretations. Further, tropical records comprise less than half of each compilation and since the publication of those studies, many new records have been generated in data-sparse regions that are key to understanding the complexities of tropical precipitation

variability. Finally, recent studies (e.g., Atwood et al., 2020) have demonstrated significant regional variability in the tropical precipitation response to a variety of forcings, including North Atlantic meltwater events, calling into question the usefulness of invoking a southward shift in the zonal mean ITCZ as the primary mechanism driving hydroclimate changes in response to the 8.2ka Event, as invoked in the reconstructions of Morrill et al. (2013) and Parker and Harrison (2022).

This study seeks to provide new insights into the tropical hydroclimate response to the 8.2ka Event, by compiling an updated

set of hydroclimate-sensitive proxy records complete with age model uncertainty and integrating them with new statistical tools to quantitatively evaluate how tropical rainfall patterns responded to this period of abrupt global climate change. We further




assess how well the proxy reconstructions compare to a new isotope-enabled model simulation of the 8.2 ka Event. In doing so, we hope to improve our understanding of the tropical hydroclimate response to abrupt AMOC disruptions and provide a critical benchmark for climate models that are used in projections of future climate change.

## 2 Methods

### 2.1 Synthesis of Published Datasets

To assess the tropical hydroclimate response to the 8.2ka Event, we developed an updated compilation of published, high-resolution, continuous, well-dated proxy datasets, collating records that span the period 7ka-10ka, cover latitudes from 30°N to 30°S, and which are sensitive to some aspect of hydroclimate variability. Records were identified through an in-depth literature review, searching public data repositories (e.g., the NOAA National Centers for Environmental Information and World Data Center PANGAEA databases), and incorporated from previous compilations (e.g., Morrill et al., 2013). All records were reformatted into the Linked Paleo Data framework (LiPD; McKay and Emile-Geay, 2016) to facilitate analyses of age uncertainty and quantitative event detection.

To constrain the timing and duration of the abrupt hydroclimate anomaly associated with the 8.2ka Event, the datasets in this compilation were screened to meet the following criteria: (i) data resolution of 50 years or better over the period of 7ka-10ka; (ii) based on hydroclimate-sensitive proxy data interpreted by authors as reflecting precipitation amount or intensity, the isotopic compositions of environmental water (including precipitation, lake water, and seawater), effective moisture, lake level, fluvial discharge, or sea surface salinity (SSS); and (iii) contain at least three radiometric dates over the 7ka-10ka interval. Emphasis was placed on collecting water isotope-based records to enable more direct comparison with isotope-enabled climate model simulations.

The compilation was organized into three categories based on the climate interpretation of the various proxy records: proxies which reflect the isotopic composition of precipitation ($P_{iso}$), proxies which reflect effective moisture (EM; P-E), and proxies which reflect precipitation amount and/or intensity ($P_{amt}$). This categorization scheme enables more robust interpretations of the proxy records and facilitates data-model comparison as our understanding of water isotopes and their manifestations in paleoclimate archives continues to advance (Konecky et al., 2020).

### 2.2 Age Model Development

Published radiometric age data were used to develop an ensemble of age-depth models for each dataset using Bayesian methods. Where available (Table A1), we employed the age ensembles developed by the Past Global Changes (PAGES) Speleothem Isotope Synthesis and Analysis (SISAL) working group, using version 2 of their database (Comas-Bru et al., 2020). For records for which these age ensembles were not available due to lack of inclusion in the SISALv2 database or comprising a lacustrine or marine sediment archive, we developed age-depth models using the geoChronR package in R version 4.2.1 (McKay et al., 2021). All radiometric dates were obtained from the original publications and screened for updated age data



where available. For records originating from the Northern Hemisphere tropics, radiometric dates were calibrated using the Northern Hemisphere calibration curve, IntCal20. Dates of records originating from the Southern Hemisphere tropics were

130 calibrated using the Southern Hemisphere calibration curve, SHCal20. For each record, 1000 age-depth model iterations were run to generate a Markov Chain Monte Carlo (MCMC) age ensemble, which produces median age values and quantile age ranges, facilitating the propagation of age-model uncertainties through subsequent analyses.

In order to reduce uncertainty arising from the differences in age modeling algorithms offered through geoChronR, we prioritized the use of BACON (Blaauw and Christen, 2011) across our records, including those in the SISALv2 database,

where available. If a BACON age ensemble was not constructed for a SISALv2 dataset, we employed the Bchron (Haslett and Parnel, 2008) or copRA (Breitenbach et al., 2012) ensembles instead.

## 2.3 Detection of the 8.2ka Event

Two event detection methods were used in this study, as detailed below. The start, end, and duration of the hydroclimate anomalies associated with the 8.2 ka Event were calculated for all records in which events of the same sign were detected in

both event detection methods. This was done to leverage the strengths of each detection method and provide a more robust reconstruction of the hydroclimate response to the 8.2ka Event.

### 2.3.1 Modified Morrill Method

For each record's published time series, we apply a modified version of the event detection methods described in Morrill et al. (2013) as a control to compare against our actR results (hereafter referred to as MM). Using the period from 7.4ka to 7.9ka as

a reference period, we calculate the mean and the standard deviation over that interval. From there, we define the upper and lower bounds by the two-sigma level. We repeat this process for a second reference period from 8.5ka to 9.0ka. We take the final upper and lower bounds as the most extreme values between the two reference periods. Then we use the 7.9-8.5ka period as the 8.2ka Event detection window.

Over this period, any values which exceed the upper or lower bound are marked as the 8.2ka Event, with the timing of the

150 event defined by the ages of the proxy values that exceed those bounds. In order for an excursion to be considered part of the 8.2ka Event, the excursions must last at least 10 years. If multiple events are detected within the 7.9-8.5ka window, they are combined into a single event if there are no more than three data points or thirty years separating the different excursions. This modification is necessary to account for the varying sampling resolutions present within and between several of the records in our compilation. If multiple events of differing signs are detected within the 8.2ka Event window, the event with the largest

z-score is chosen as the representative hydroclimate response. The magnitude of the event is defined by the largest absolute value z-score within the event detection period.





### 2.3.2   actR Method

A second event detection method was used to account for age model uncertainties in the proxy records. Past studies (e.g., Morrill et al., 2013) of the 8.2ka Event employed statistical techniques to detect excursions in the proxy records using an a priori assumption that the North Atlantic meltwater perturbation propagated globally at exactly 8.2ka and lasted no more than 200 years. To better constrain the timing, duration, and magnitude of the 8.2ka Event in this study, we employed an event detection algorithm based on the changepoint package in the newly developed Abrupt Change Toolkit in R (actR; McKay and Emile-Geay, 2022). This algorithm detects abrupt shifts in the mean of a time series based on a prescribed number of age model ensembles (generated in geoChronR), the minimum length of a segment (in years) over which mean shifts in the time series are detected, and a user-defined changepoint detection method and weighting penalty function. A minimum segment length of 50 or 100 years was assigned for each record in the proxy compilation to minimize short-lived transitions in the noisy proxy records, with the assumption that the 8.2ka Event signal in each of the records lasts at least 50 years.

Detected changepoints were summarized over 10-year-long windows. The Pruned Exact Linear Time (PELT; Killick et al., 2012) changepoint detection method was chosen for its computational efficiency and dynamic programming approach to accurately identify the location and number of changepoints in time series data. The Modified Bayesian Information Criterion (MBIC; Zhang and Siegmund, 2007) was chosen as the penalty weighting function to balance the goodness of fit of the model to the data with the complexity of the model and the number of changepoints. These methods effectively minimize the detection of spurious changepoints within each ensemble. Each time series ensemble was tested against a robust null hypothesis using surrogate proxy data generated by an isospectral noise model. By construction, the surrogate data have the same power spectrum as the original data, but phase scrambling destroys any autocorrelation that was present in the original time series. If autocorrelation is detected in a segment of the original time series ensemble, it fails the null hypothesis test, and any changepoint detected within that segment is excluded from the result. This test helps to ensure that the detected changepoints are statistically significant and not just the result of random variation. Both age and proxy data uncertainties are propagated through each ensemble, improving the robustness of the result. For each record, 1000 age model ensembles were generated and tested against 100 surrogate time series.

Two types of events were characterized based on the actR results. 'Significant events' are defined by the presence of two consecutive changepoints with $p < 0.05$ over the 7.9-8.3ka window ("start" and "end"). If more than two consecutive changepoints exist over that window, the two with the lowest p-values and highest probability are used. The difference between "start" and "end" dates is used to calculate event duration, which we assume to be no greater than 300 years. The magnitude of "events" is determined by the greatest absolute value z-score in each record's median age ensemble time series between the actR-derived "start" and "end" dates, with interpretation based on the sign of the z-score corresponding to the interpretation direction of the original authors. 'Tentative events' are defined by the presence of two consecutive changepoints with $p < 0.1$ over the 7.7-8.5ka window ("start" and "end"). Events lasting more than  300 years are removed from consideration. If more than two events are detected within that window, the event with the start date closer to 8.2ka is chosen as the final 8.2ka event.



## 2.4 iCESM Simulations

The National Center for Atmospheric Research's (NCAR) water isotope-enabled Community Earth System Model (iCESM1.2; Brady et al., 2019) is a state-of-the-art, fully-coupled GCM designed to simulate water isotopes across all stages of the global hydroclimate cycle. It employs the CAM5.3 atmospheric model, with a gridded resolution of 1.9° latitude x 2.5° longitude and 29 vertical levels. Land processes are modeled by CLM4, at the same nominal 2° resolution. CLM is coupled to a River Transport Model which routes runoff from the land into oceans and/or marginal seas. Both the POP2 ocean model and the CICE sea ice model have a common grid size of 320 x 384 with a nominal 1° resolution near the equator and in the North Atlantic. While iCESM faithfully captures the broad quantitative and qualitative features of precipitation isotopes, it is known to have a global bias toward depleted precipitation $\delta^{18}$O (median bias of -2.5‰; Brady et al., 2019).

We performed a new 8.2ka Event meltwater-forced ("hosing") simulation and an early Holocene control simulation ("ctrl") using iCESM1.2. iCESM enables explicit tracking of water isotopes throughout the global water cycle, facilitating quantitative comparisons between model output and water isotope-based proxy records. These simulations were based on the freshwater forcing scenario recommended by Otto-Bliesner et al. (2017) for their proposed Paleoclimate Modeling Intercomparison Project 4-Coupled Model Intercomparison Project 6 (PMIP4-CMIP6) 8.2ka simulation. The climate of the early Holocene is characterized by different orbital configurations, including a larger obliquity and slightly higher eccentricity than present day. In addition, precession resulted in increased seasonality of insolation in the Northern Hemisphere, with greatest insolation receipt in boreal summer (Wu et al., 2018; Hu et al., 2019). These factors produced warmer Northern Hemisphere summers, especially in mid to high latitudes, which are thought to have promoted the retreat of the remnant Laurentide Ice Sheet (LIS) (Otto-Bliesner et al., 2017 and references therein).

To simulate early Holocene climate, the model was forced with prescribed greenhouse gas concentrations (CH$_4$ = 658.5 ppb, CO$_2$ = 260.2 ppm, and N$_2$O = 255 ppb), orbital configurations (eccentricity = 0.019524°, obliquity = 24.2030°, and longitude of perihelion = 99.228°), and a reconstruction of the ice sheet extent (Peltier et al., 2015) representative of conditions at 9ka. A control simulation ("ctrl") was initialized from an earlier 400-year-long 9ka simulation and was run for 100 model years using these parameters. The 8.2ka Event simulation ("hose") was branched from year 100 of this control run. Initially, a simulated 2.5Sv meltwater flux (meltwater $\delta^{18}$O = -30‰; Zhu et al., 2017) was applied across the northern North Atlantic Ocean (50–70°N) for 1 year, followed by 0.13Sv flux for 99 years to approximate the abrupt drainage of Lakes Agassiz and Ojibway and eventual collapse of the LIS at Hudson Bay (Otto-Bliesner et al., 2017). Monthly surface air temperature, precipitation amount, and precipitation $\delta^{18}$O variables were extracted from each simulation for analysis. To isolate the global response to the simulated 8.2ka Event in the model, yearly time series of temperature (°C), precipitation amount (mm/day) and precipitation amount-weighted precipitation $\delta^{18}$O (‰) were obtained. Anomalies for each variable were calculated by subtracting the final 50 years of the "ctrl" simulation from the final 50 years of the "hose" simulation.





## 3 Results

### 3.1 Data Compilation

61 tropical hydroclimate proxy records have been compiled in this study, covering 17 IPCC-designated scientific regions (Fig. A1; Iturbide et al., 2020). Compared to the compilation in Morrill et al. (2013), there is substantial improvement in hydroclimate
proxy data coverage across the Caribbean, Central America, South America, South and East Asia, and the Maritime Continent.

The compilation comprises 42 speleothem records ( 69%), 14 lacustrine records ( 23%), and 5 marine records ( 8%). When categorized into the three hydroclimate interpretation groups, the compilation consists of 43 $P_{iso}$ records (70.5%), 11 EM records (18%), and 7 $P_{amt}$ records (11.5%; Fig. 1; Table 2). For the purpose of this study, records which fully meet all inclusion criteria are designated as Tier 1 records (n = 50, 82%), forming the basis for the data-model intercomparison. Records which
fail to meet either the minimum paleodata resolution or radiometric date requirements are classified as Tier 2 records and are included as supporting datasets (n = 10, 16%). One record (MWS1; Dutt et al., 2015) failed to meet both of these requirements, thus it is designated as a Tier 3 record, and has been excluded from further analysis.

### 3.2 Timing, Magnitude, and Duration of the 8.2ka Event in the Proxy Compilation

The approximate start, end, and duration of hydroclimate anomalies associated with the 8.2 ka event were calculated for all
235 records in our compilation in which events of the same sign were detected in both our modified MM and actR event detection methods. This was done to provide a more robust reconstruction of the hydroclimate response to the 8.2ka Event than that which either method would achieve alone. This final set of records comprises 30 of the 61 records (49%) in our compilation. The remaining 31 records in our compilation displayed a lack of agreement in the sign or presence of an event and are thus excluded from further analysis.

Of the 30 records that exhibit agreement between the two detection methods, significant hydroclimate events were detected in 18 records (34% of all Tier 1 and 10% of all Tier 2 records), with the remaining 12 records showing no event in either detection method (14% of all Tier 1 records and 50% of all Tier 2 records). Globally, the detected hydroclimate anomalies had an average start age of 8.28ka, and average termination age of 8.13ka, and an average duration of 152 years. The longest events occurred in the foraminifera $\delta^{18}O$ record from the Gulf of Mexico (LoDico et al., 2006; MD022550; Fig. A39; 289 years) and
the speleothem record from Chongqing, China (Yang et al., 2019; HF01; Fig. A23; 259 years). The Chinese lacustrine magnetic susceptibility record of Hillman et al. (2021; F14; Fig. A17) has the earliest event onset age of 8.49ka, with a termination at 8.34ka, for a total duration of 152 years, while the Chinese speleothem record of Dykoski et al. (2005; D4Dykoski; Fig. A15) has the latest event onset age at roughly 8.11ka, terminating near 8.04ka, for an event duration of 62 years. Because the event detection algorithm in actR can be compromised by highly variable sampling resolution, in records with highly variable
resolution, we used the MM method to determine the event onset, termination, and duration. This applies to only two records: the speleothem record from Dykoski et al. (2005; D4Dyoski, Fig. A15) and the speleothem record from Neff et al. (2001; H5; Fig. A21).





Drier and/or isotopically enriched events were detected in 13 of the 30 records in the final compilation, including six records from East Asia (e.g. D4Dykoski ($+2.8\sigma$) and F14 ($+5.8\sigma$)). Similarly, drying/isotopic enrichment was seen in three speleothem
records from the Arabian Peninsula, with the largest shift ($+3.5\sigma$) detected in the record of Cheng et al. (2009; H14; Fig. A22) between 8.08-8.21ka. The two speleothem records of Chawchai et al., 2021 from Klang Cave, Thailand (TK07, Fig. A57; TK20, Fig. A58) showed similarly high levels of isotopic enrichment ($+3.1\sigma$ and $+2.5\sigma$) between approximately 8.16-8.30ka. Finally, an isotopic enrichment of $+3.4\sigma$ between 8.05-8.19ka in the Costa Rican speleothem record of Lachniet et al. (2004; V1; Fig. A61) and a negative excursion in titanium content indicative of a drying event ($-4.0\sigma$) in the Guatemalan lake sediment
record of Duarte et al. (2021; Core5LI; Fig. A10) from 8.09-8.16ka suggests a hydroclimate response to the 8.2ka Event in southern Central America (south of the Yucatan Peninsula; Fig. 6).

Wetter and/or isotopically depleted events were detected in five of the 30 records in the final compilation. Namely, the Madagascar speleothem records of Voarintsoa et al. (2017; ANJB2; Fig. A4) and Duan et al. (2021; ABC1; Fig. A3) showed negative isotopic excursions of $-3.0\sigma$ and $-2.5\sigma$, respectively, while the two Brazilian speleothem records from Lapa Grande
Cave (Strikis et al., 2011; LG11; Fig. A32 and Padre Cave (Cheng et al., 2009; PAD07; Fig. A43) exhibited negative isotopic excursions of $-2.9\sigma$ and $-2.7\sigma$, respectively (Table 5). In addition, a large isotopic depletion event ($-3.8\sigma$) was detected in the foraminifera $\delta^{18}$O record from the Gulf of Mexico (LoDico et al., 2006; MD022550; Fig. A39).

We found no significant hydroclimate response in the remaining 12 records of our compilation, with both the MM and actR event detection methods in agreement that no event occurred. This category included three lake sediment records from the
270 Yucatan Peninsula (Fig. 6; LC1 [Hodell et al., 1995; Fig. A31], Curtis6VII93 [Curtis et al., 1998; Fig. A13], LagoPuertoArturo [Wahl et al., 2014; Fig. A29]), two speleothem records from Southeast Asia/the Maritime Continent (Fig. 7; KMA [Berkelhammer et al., 2012; Fig. A27], SSC01 [Carolin et al., 2016; Fig. A53]), and two speleothem records from Brazil (Fig. 5; RN1 [Cruz et al., 2009; Fig. A49], TM6 [Ward et al., 2019; Fig. A59]).

### 3.3 Regional Coherency of the Reconstructed Hydroclimate Changes

There is substantial regional coherency in the spatial pattern of reconstructed hydroclimate anomalies (Fig. 2), though they do not conform to the anticipated hemispheric dipole pattern typically associated with the 8.2ka Event, i.e. a generally drier/isotopically enriched Northern Hemisphere and a wetter/isotopically depleted Southern Hemisphere. Under both the MM and actR event detection methods, prominent drying/enrichment occurred across East and Southeast Asia, as well as the Arabian Peninsula. These dry conditions are interspersed with areas of no change in parts of the Maritime Continent and eastern
India/Tibetan Plateau. No robust signatures of the 8.2ka Event are observed over the Maritime Continent. Central and South America display more of a hemispheric dipole pattern, with dry/enrichment events occurring north of the equator in Costa Rica and Guatemala, contrasting with wet/depletion events south of the equator in central/eastern Brazil. However, there are also regions in northern and central Brazil that exhibit no hydroclimate response. The proxy records thus present a far more complex, regionally-specific hydroclimate response to the 8.2ka Event than a simple hemispheric dipole pattern.





## 3.4  Global Signature of the 8.2 ka Event in iCESM

We now compare these reconstructed hydroclimate patterns to those simulated by iCESM under 8.2ka meltwater forcing. The surface temperature response is characteristic of the "bipolar seesaw" pattern (i.e., a colder northern hemisphere and a warmer southern hemisphere that is most pronounced in the Atlantic Ocean), consistent with a reduction in northward heat transport by AMOC (Fig. A2). Anomalously cool surface temperatures, reaching as low as -20°C where freshwater forcing was applied, stretch across the northern North Atlantic Ocean, down the western coasts of Europe and North Africa, and into the tropical Atlantic via the North Atlantic Subtropical Gyre. Surface air temperatures across the Southern Hemisphere show a positive anomaly of up to 3°C, with the largest warming occurring in the South Atlantic. Over the continents, surface air temperatures cool in all regions except localized parts of northern South America, West Africa, and the southernmost regions of South America and Australia.

Accompanying these temperature anomalies are notable anomalies in precipitation amount, precipitation $\delta^{18}$O, and effective moisture (Fig. 3). Precipitation decreases while effective moisture increases throughout much of the North Atlantic, with the responses most pronounced in the regions with greatest cooling. The increase in effective moisture in this region indicates that the evaporation reduction outpaces the precipitation reduction (Fig. 3c). In the tropics, the largest precipitation anomalies appear in the tropical Pacific and Atlantic basins, with a southward shift of the Pacific and Atlantic ITCZs occurring in response to the freshwater forcing (Fig. 3b). These shifts are characterized by a weakening of the northern extent of the ITCZs and an enhancement of the southern extent. The most pronounced drying occurs over central America and the eastern tropical Pacific, including Costa Rica and Panama, while the largest wet anomalies occur across the southern tropical Pacific. A notable hemispheric dry/wet dipole pattern is also observed in the tropical Atlantic, extending over northeastern South America. This pattern is less pronounced but still present over the tropical Indian Ocean and Africa. In contrast, no such dipole occurs over the western Pacific or Maritime Continent.

These temperature and precipitation anomalies project strongly onto the amount-weighted precipitation $\delta^{18}$O values (Fig. 3a). The greatest precipitation $\delta^{18}$O anomalies occur in the northern reaches of the North Atlantic Ocean, reaching up to -8‰ in association with the strong regional cooling of the North Atlantic, as well as the addition of highly depleted (-30‰) meltwater to the surface ocean of the "hosing" site, and subsequent evaporation and rainout. In the tropics, precipitation $\delta^{18}$O anomalies closely follow the changes in precipitation amount over the tropical Atlantic and central/eastern Pacific Oceans, with negative precipitation $\delta^{18}$O anomalies south of the equator and positive precipitation $\delta^{18}$O anomalies north of the equator. A pronounced dipole pattern is also evident over northern South America, where anomalously increased (decreased) rainfall amounts correspond to negative (positive) precipitation $\delta^{18}$O anomalies in the southeastern (northwestern) region of South America. In the Middle East, India, Tibetan Plateau, and parts of Southeast Asia, modest drying is accompanied by pronounced positive precipitation $\delta^{18}$O anomalies. However, not all regions are well described by the amount effect. In the Caribbean and Central America, there is a positive relationship between precipitation $\delta^{18}$O and precipitation amount, characterized by strong drying and isotopic depletion (associated with the strong upwind cooling and meltwater addition in the North Atlantic). There



also appears to be no clear relationship between precipitation amount and precipitation $\delta^{18}$O anomalies over Africa, East Asia, the Western Pacific, and Maritime Continent.

## 3.5    Data-Model Comparisons

The proxy locations span a total of 15 IPCC scientific regions (Fig. A1). The regions with densest coverage of Tier 1 proxy data are Central America, northeastern South America, East Asia, and Southeast Asia/Maritime Continent. These four regions were thus targeted for the data-model comparisons. The proxy records within each region were compared to the model-simulated anomalous annual mean precipitation amount, amount-weighted precipitation $\delta^{18}$O, and effective moisture (P-E) to investigate the data-model agreement in the four target regions.

In East Asia (Fig. 4; Table 3; Table 4), five speleothem records display an isotopic enrichment event that broadly corresponds to the simulated large-scale isotopic enrichment pattern in precipitation $\delta^{18}$O that stretches across South Asia and the Arabian Peninsula in iCESM (Fig. 4a,b). This broad enrichment pattern in the model corresponds well with the similarly broad isotopic enrichment pattern found in the reconstructions, spanning East Asia, the Arabian Peninsula and southern Thailand. In iCESM, the Chinese speleothem records are located near the node of an east-west dipole pattern in precipitation $\delta^{18}$O in eastern China, which is part of a larger pattern of isotopic depletion in South Asia and isotopic enrichment in the subtropics and extratropics of the North Pacific. In addition to these speleothem records, there is a single lake sediment record in southern China that displays a notable drying event, indicative of reduced precipitation, however this record contrasts with the results from iCESM, which show no notable change in precipitation in this region (Fig. 4b,e).

Northeastern South America displays only moderate proxy-model agreement (Fig. 5). Two of the four speleothem records there contain large $\delta^{18}$O depletion events, corresponding with the large-scale isotopic depletion signal in precipitation $\delta^{18}$O in iCESM across northeastern South America. However, two other speleothem records in the region–one in the Nordeste region of Brazil and one in central Brazil–show no significant hydroclimate anomalies during the 8.2ka Event, in contrast with the results from iCESM.

Subject to the small sample size in Central America (Fig. 6), broad agreement is found between the simulated and reconstructed hydroclimate anomalies, with the dry event detected in the Guatemalan lake sediment record of Core5LI (Duarte et al., 2021) corresponding with the reduced precipitation simulated throughout central America in iCESM. In addition, a positive (enriched) precipitation $\delta^{18}$O event found in the Costa Rican speleothem record (V1) is not inconsistent with the simulated isotopic enrichment in precipitation $\delta^{18}$O in the southernmost extent of Central America in iCESM, forming part of the wide enrichment signal that stretches across the eastern Pacific. However, like the East Asian records, this speleothem record sits at the nodal point of a pronounced east-west dipole pattern in precipitation $\delta^{18}$O in iCESM, characterized by isotopic enrichment in the eastern tropical Pacific and widespread isotopic depletion in the tropical North Atlantic that stretches into the Caribbean and Mesoamerica. Lastly, the lack of a detected event in three lake sediment records from the Yucatan Peninsula (LagoPuertoArturo, Curtis6VII93, LC1) also agrees with the simulated weak EM response in that region in iCESM.

Broad data-model agreement is also found in Southeast Asia and the Maritime Continent (Fig. 7), where one speleothem record in the Thailand peninsula contains a notable isotopic enrichment event, in agreement with the simulated large scale





enrichment signal in precipitation $\delta^{18}$O in iCESM in South Asia (Fig. 7a,d). Two other speleothem records in Sumatra and Borneo show no significant hydroclimate anomalies, in general agreement with the weak simulated precipitation $\delta^{18}$O anomalies in iCESM in this region, which reflect the weak response in precipitation $\delta^{18}$O throughout the western Pacific and Maritime

Continent (Fig. 3a).

## 4 Discussion

### 4.1 Comparison to Previous Hydroclimate Compilations

The spatial pattern of the hydroclimate responses found in this study broadly agrees with those presented in Morrill et al. (2013) and Parker and Harrison (2022). Both sets of reconstructions, as well as the present study, find large-scale drying across

East Asia and the Arabian Peninsula, as well as a robust wet/depletion signal in central eastern Brazil; in Morrill et al. (2013) and Parker and Harrison (2022) this signal coincides with a dry/enrichment event in northern South America in agreement with the simulated hydroclimate response in iCESM (Fig. 3). All three reconstructions also agree on drying/enrichment in southern Central America, while both the present study and Parker and Harrison (2022) find a dipole pattern of wet/isotopically depleted conditions in the Caribbean/Gulf of Mexico and dry/isotopically enriched conditions in southern Central America (Fig. 3a-c).

There is a reasonable level of agreement among all three compilations regarding the timing and duration of the hydroclimate anomalies. The age ensembles produced in the current analysis yield a mean start age of 8.28±0.12ka (1$\sigma$) and a termination age of 8.11±0.09ka (1$\sigma$) for the tropical hydroclimate response to the event. This timing agrees within age uncertainty to the timing of the event established from northern Greenland ice core data (8.25ka to 8.09ka; Thomas et al., 2007). In a compilation of eight absolutely-dated speleothems from South America and East Asia, Cheng et al. (2009) calculated an event

onset at 8.21ka and termination at 8.08ka. Similarly, Parker and Harrison (2022) used a compilation of 275 absolutely-dated speleothems to calculate the start of the global event at 8.22±0.01ka and its termination at 8.06±0.01ka. Importantly, the present study is the first to adopt a more comprehensive approach to age uncertainty in the proxy records through the generation of age ensembles in our analysis. The larger uncertainty in the timing of the event identified in this study is likely due to the inclusion of the age ensembles, as well as the multiproxy nature of our dataset, which includes lower resolution lake and

marine sediment records, in addition to the higher resolution speleothem records.

The duration of the hydroclimate anomalies identified in this study is also consistent with previous estimates. Parker and Harrison (2022) estimate durations ranging from 159 to 166 years, while Cheng et al. (2009) report a duration of 150 years, both of which fall within the range estimated from layer-counted Greenland ice cores (160.5±5.5 years; Thomas et al., 2007). In their multi-proxy dataset, Morrill et al. (2013) report a larger range in event duration, spanning from 40 to 270 years. In the

380 present compilation, the mean duration of the tropical hydroclimate anomalies is 152±70 years (1$\sigma$), with a range of 50-289 years, lending support to the multi-decadal to multi-centennial range of timescales found in earlier studies. Importantly, this work is the first to explicitly account for age uncertainty through all phases of the event detection analysis.

The multiproxy nature of this compilation, the explicit accounting of age uncertainties, and a geographic focus on the tropics likely contributed to a lower percentage of records which show robust signals of the 8.2ka Event (30%) relative to previous





studies (e.g., 70% of the global speleothem records compiled by Parker and Harrison, 2022). Proxy datasets from the tropics
are likely to record lower magnitude climate anomalies relative to those observed in the North Atlantic and Europe, which
experienced more pronounced climatic impacts from their proximity to the meltwater forcing that caused the 8.2ka Event.
Moreover, while certain records may exhibit significant changes between 7.9ka and 8.3ka, the lack of sufficient age constraints
in some records preclude the identification of robust events that can be tied to the 8.2ka Event (as these records display wide
ranging age ensembles that fail the null hypothesis test in actR).

## 4.2   Simulation of the 8.2 ka Event in iCESM

Proxy data play a crucial role in reconstructing local climate, but such data are often sparsely distributed, particularly through
the tropics and the Southern Hemisphere, prohibiting a complete global picture of the 8.2ka Event. Ocean-atmosphere coupled
General Circulation Models (GCMs) provide an invaluable tool for testing the response of the climate system to various forc-
ings, and provide an estimate of the climate response in regions with scarce proxy data. Moreover, climate model simulations
that track water isotopes through the hydrologic cycle facilitate more direct comparisons between proxy and model data.

In agreement Two lower-resolution isotope-enabled GCM simulations have previously been conducted to investigate the 8.2ka Event.
LeGrande and Schmidt (2008) used the Goddard Institute for Space Studies ModelE-R (GISS ModelIE-R) to evaluate the
response of global temperatures, precipitation amount, and precipitation $\delta^{18}$O values to a slowdown of the AMOC. GISS
ModelIE-R is a fully-coupled GCM from the IPCC AR4 era, featuring a 4° x 5° horizontal resolution atmosphere model
coupled with an ocean model of the same resolution, comprising 20 and 13 vertical layers, respectively. LeGrande and Schmidt
(2008) performed a 1,000-year preindustrial control simulation and a suite of twelve meltwater forced experiments, applying a
range of forcings (1.25 Sv to 10 Sv) over the Hudson Bay for 0.25 to 2 years. They found that this range of meltwater forcings
inhibited North Atlantic Deep Water (NADW) formation and reduced the strength of the AMOC for up to 180 years.

In agreement with the results from iCESM, LeGrande and Schmidt (2008) found large precipitation $\delta^{18}$O anomalies over the
meltwater source area in the North Atlantic in the decade following the meltwater forcing, which they similarly attributed to
the evaporation and rainout of the isotopically depleted meltwater in the region. They observed reasonable agreement between
their simulations and proxy records of temperature and hydroclimate, with the simulations containing larger meltwater forcing
exhibiting better agreement with the proxies (emphasizing the importance of considering an ensemble of simulations to find the
best fit to proxy reconstructions). Regarding the tropical hydroclimate response, they identified bands of enriched (depleted)
precipitation $\delta^{18}$O anomalies in the northern (southern) tropics as a result of a southward shift in tropical rainfall. Notable
patterns of $\delta^{18}$O enrichment were identified in Northeastern Africa, through the Middle East, south Asia, and the Thailand
peninsula, which they attributed to large-scale changes in the hydrologic cycle, including shifts in moisture source and moisture
transport pathways.

In a more recent set of simulations, Aguiar et al. (2021) used the University of Victoria Earth System Climate Model
version 2.9 (UVic ESCM2.9) with the addition of oxygen isotopes to test proxy-model agreement under a range of empirically
derived freshwater forcing scenarios. UVic ESCM2.9 uses the Modular Ocean Model version 2, with a horizontal resolution
of 3.6°x1.8° and 19 vertical levels. The version of the UVic ESCM2.9 model used in this study possesses a simple two-




dimensional atmospheric energy moisture balance model, which limits its ability to accurately represent precipitation $\delta^{18}$O

values. Aguiar et al. (2021) compared the sea surface temperatures and seawater $\delta^{18}$O values from 28 simulations with 35 proxy records to place new constraints on the amount and rate of freshwater forcing in the North Atlantic. Their analysis revealed that a two-stage meltwater experiment with a background flux of 0.066Sv over 1,000 years (9-8ka), followed by an intensification to 0.19Sv over 130 years (8.31-8.18ka), best replicated the anomalies observed in the proxy records.

The iCESM simulation illustrates clear signatures of the global 8.2ka Event that, at the largest scales, are broadly consistent

with the GISS and UVic simulations described above, including the hemispheric dipole pattern in temperature and associated southward shift of the tropical rainbands. On regional scales, the tropical rainfall patterns display substantial regional heterogeneity, with a southward shift of the tropical ocean rain bands, drying in the major NH monsoon regions of South Asia and West Africa, and wetting in parts of the South American Summer Monsoon. Tropical precipitation $\delta^{18}$O values display strong signatures of the 8.2ka Event, including opposing patterns of precipitation $\delta^{18}$O values between northern South

America and northeast Brazil (e.g., Zhu et al., 2017) and large precipitation $\delta^{18}$O anomalies over the meltwater region (e.g., LeGrande and Schmidt, 2008; Bowen et al., 2019). Dry (wet) anomalies correspond with enriched (depleted) precipitation $\delta^{18}$O values in some tropical regions, implicating the "amount effect" as the driving force behind the isotopic signal, but a decoupling of precipitation amount and precipitation $\delta^{18}$O anomalies occurs over many tropical continental regions, indicating that other processes such as changes in moisture source, moisture transport pathways, water recycling over land, and/or

changes in precipitation seasonality, dominate the isotopic signal in those regions. The model simulations lend support to the proxy reconstructions in demonstrating that the tropical hydroclimate response to the 8.2ka Event cannot be described as a simple hemispheric dipole pattern, particularly over continental regions, and that the rich regional structure of the precipitation amount and precipitation $\delta^{18}$O responses must be considered in order to understand the full picture of the tropical hydroclimate response to this event.

**4.3 Data-Model Comparisons**

Subject to the small sample sizes found in the regional data-model comparisons, the results suggest that iCESM captures many of the regional hydroclimate responses observed in the reconstructions, including the large-scale isotopic enrichment pattern in precipitation $\delta^{18}$O in South and East Asia, the muted hydroclimate response in the Maritime Continent, the drying and isotopic enrichment in southern Central America, and the isotopic depletion in precipitation $\delta^{18}$O in parts of northeastern South

America. While qualitative, these areas of agreement between the proxies and model demonstrate that the tropical hydroclimate response to North Atlantic meltwater forcing during the 8.2ka Event was not a simple hemispheric dipole pattern, but is instead characterized by rich regional structure.

However, while there is some qualitative agreement between many of the reconstructed and simulated regional hydroclimate anomalies during the 8.2ka Event, our data-model comparisons are subject to a variety of limitations. For starters, our regional

analyses are limited by small sample sizes and in some regions like East Asia, the point-to-point agreement between the proxy and model data is low even as the regional hydroclimate patterns offer more nuanced context. In addition, our data-model comparisons are necessarily qualitative as many of the proxy records in our compilation are carbonate $\delta^{18}$O records, which do





not solely reflect changes in precipitation $\delta^{18}$O. Rather, these archives incorporate a combination of the isotopic composition of groundwater (for speleothem $\delta^{18}$O records; Lachniet, 2009) or seawater (for marine $\delta^{18}$O records; Konecky et al., 2020)

as well as the environmental temperature, among other factors (LeGrande and Schmidt, 2009; Bowen et al., 2019; Konecky et al., 2019). Thus, future work should integrate proxy system models with water isotope-enabled climate model simulations to develop more quantitative data-model comparisons of the 8.2ka Event. In addition, quantitative metrics like the weighted Cohen's kappa statistic could be used to quantitatively compare the proxy reconstructions to the pseudoproxy data derived from climate models (Cohen, 1960; Cohen, 1968; Landis and Koch, 1977; DiNezio and Tierney, 2013).

However, even when attempting to bridge the gap between models and proxy data using proxy system models and quantitative metrics, robust comparisons remain challenging. Characterizing the point-to-point agreement between the observed and simulated climate anomalies fails to address the well-known hydroclimate biases that exist in GCMs, which arise from factors like course model resolution, idealized topography, and the unresolved physics of cloud formation and convection. Furthermore, proxy data often captures localized climate signals which may not be representative of regional conditions. In contrast,

model data is averaged over the area of a grid cell, which can be large in coarse-resolution models. This can lead to non-trivial biases, particularly in coastal regions and regions of complex topography. Ultimately, the incorporation of additional well-dated proxy records that resolve different aspects of hydroclimate, paired with proxy system models, and ensembles of water isotope enabled climate model simulations of the 8.2ka Event will notably improve our understanding of the characteristics and mechanisms of the tropical hydroclimate response to abrupt climate change events.

**5   Conclusions**

This study has investigated the tropical hydroclimate response to the 8.2ka Event in a new multi-proxy data compilation and state-of-the art isotope-enabled model simulation. Two event detection methods were used in this study. The first method relies on the original age model of each record and uses the 7.9-8.5ka period as the detection window. The second method implements a changepoint detection algorithm that explicitly accounts for age uncertainties in each proxy record. In order to leverage the

strengths of each method and provide a more robust reconstruction of the hydroclimate response to the 8.2ka Event, only records in which events were detected in both event detection methods were used to characterize the hydroclimate response to the 8.2 ka Event.

Robust hydroclimate anomalies were detected in 30% of the compilation across the 7.9-8.5ka interval, with z-scores ranging between $+5.8\sigma$ and $-4.0\sigma$ in East Asia and southern Central America, respectively. 12 records showed no evidence of a

hydroclimate anomaly associated with the 8.2ka Event. The remaining records had conflicting results based on the two event detection methods and were excluded from further analysis. Across the records, a mean onset age of 8.28±0.12ka ($1\sigma$), mean termination age of 8.11±0.09ka ($1\sigma$), and mean duration of 152±70 years ($1\sigma$; with a range of 50-289 years) was found, comparing well with previous estimates, and lending support to a regionally-variable tropical hydroclimate response to the 8.2ka Event, with events that span decadal to multi-centennial timescales. Importantly, this work is the first to explicitly account for

age uncertainty through all phases of the event detection analysis.





Broad agreement is observed between the reconstructions and iCESM model simulations and the results demonstrate that the tropical hydroclimate response to the North Atlantic meltwater forcing was not a simple hemispherically uniform dipole pattern, but is better characterized by rich regional structure. Coherent regional hydroclimate changes identified in proxy records include pronounced isotopic enrichment across East Asia, South Asia, and the Arabian Peninsula. In the Americas, drying and isotopic enrichment occurred in southern Central America, contrasting with isotopic depletion in central/eastern Brazil. In contrast, no signatures of the 8.2ka Event were found over the Maritime Continent.

The isotope-enabled model simulation with iCESM illustrates clear signatures of the global 8.2ka Event that are largely consistent with the proxy records. The characteristic north-south dipole pattern in surface temperature is accompanied by an associated southward shift of tropical rainfall. On regional scales, however, the tropical rainfall changes are highly variable. Major features include a southward shift of the tropical ocean rain bands in the tropical Atlantic, Central and Eastern Pacific, and Indian Oceans (characterized by a weakening of the northern extent and enhancement of the southern extent of the rain-bands), as well as drying in Central America and northern South America and wetter conditions in northeastern Brazil. Modest drying also occurs in the Northern Hemisphere monsoon regions of South Asia and West Africa.

The isotopic composition of tropical precipitation also displays strong signatures of the 8.2ka Event. Over land, precipitation $\delta^{18}$O displays a pronounced dipole pattern in South America, with isotopic enrichment in northern South America and isotopic depletion in northeast Brazil. Large-scale isotopic depletion also occurs over the Arabian Peninsula and South Asia. Over the tropical oceans (namely the tropical Atlantic, Central and Eastern Pacific, and Indian Oceans), a pronounced north-south dipole pattern occurs in precipitation $\delta^{18}$O, with isotopic enrichment corresponding with drier conditions north of the equator and isotopic depletion corresponding with wetter conditions south of the equator. Precipitation amount and $\delta^{18}$O anomalies are more muted in the Western Pacific, Maritime Continent, and Africa.

To quantify data-model agreement, the proxy records were compared to simulated precipitation $\delta^{18}$O, precipitation amount, and effective moisture (P-E) from co-located sites in four regions with the densest coverage of proxy data: Central America, northeastern South America, East Asia, and Southeast Asia. Subject to the small sample sizes found in the regional data-model comparisons, the results suggest that iCESM captures many of the regional hydroclimate responses observed in the reconstructions, including the large-scale isotopic enrichment pattern in precipitation $\delta^{18}$O in South and East Asia and the Arabian Peninsula, th drying and isotopic enrichment in precipitation $\delta^{18}$O in southern Central America, the isotopic depletion in parts of northeastern South America, and the muted hydroclimate response in the Maritime Continent.

These results serve as a first step toward more quantitative data-model comparisons. Recommendations for future studies include adding more well-dated proxy records that resolve different aspects of hydroclimate during the 8.2ka Event, and quantitatively comparing these records with ensembles of water isotope enabled climate model simulations of the 8.2ka Event paired with proxy system models. Future work should also investigate the mechanisms of the observed hydroclimate changes and their isotopic signatures to improve our understanding of the characteristics and mechanisms of the tropical hydroclimate response to abrupt climate change events.



*Author contributions.* ARA and ALM conceptualized the study, with ARA securing funding and providing supervision. ALM performed proxy data curation and led the proxy data analysis with contributions from RP. ALM performed the iCESM simulations and led the writing of the manuscript and the design of figures with contributions from ARA.

*Competing interests.* The authors declare that they have no conflict of interest.

*Acknowledgements.* This work was supported by NSF-EAR 2002444 to ARA. Computing resources were provided to ALM by the Climate Simulation Laboratory at NCAR's Computational and Information Systems Laboratory (CISL). We gratefully acknowledge Julian Sachs for providing early inspiration for the project as well as Jiang Zhu at NCAR for his assistance setting up the iCESM simulations. We also gratefully acknowledge the PMIP4 Community and the Linked Earth team, including Nick McKay and Julien Emile-Geay, for providing access to and invaluable assistance with actR (https://github.com/LinkedEarth/actR).



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





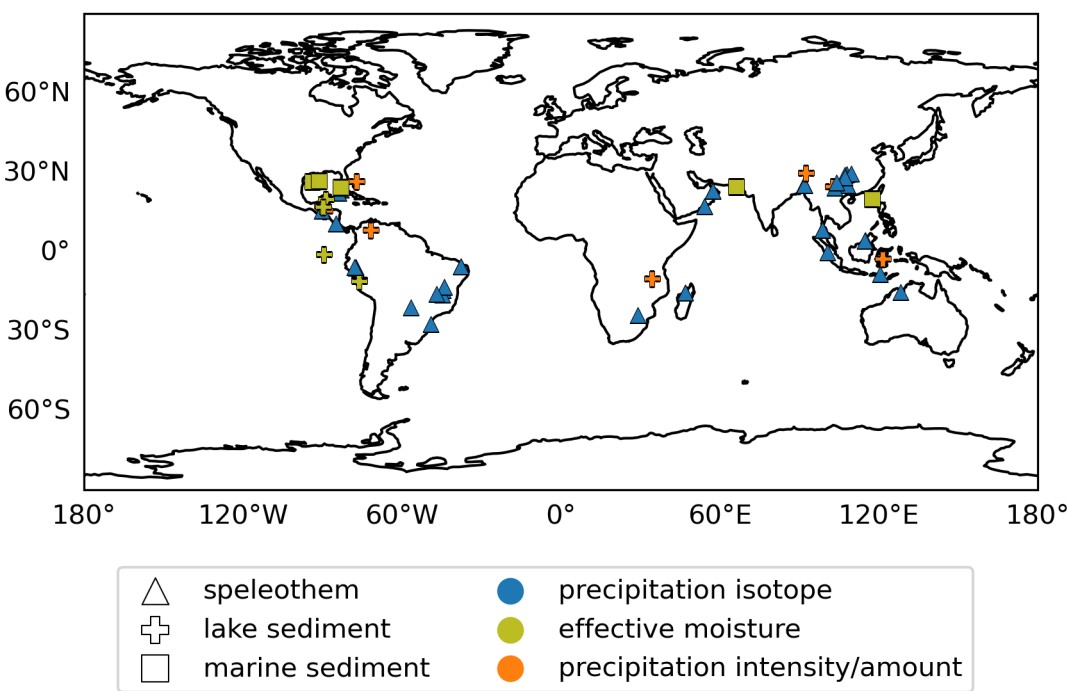

**Figure 1.** The distribution of proxy records comprising each interpretation group included in this study.



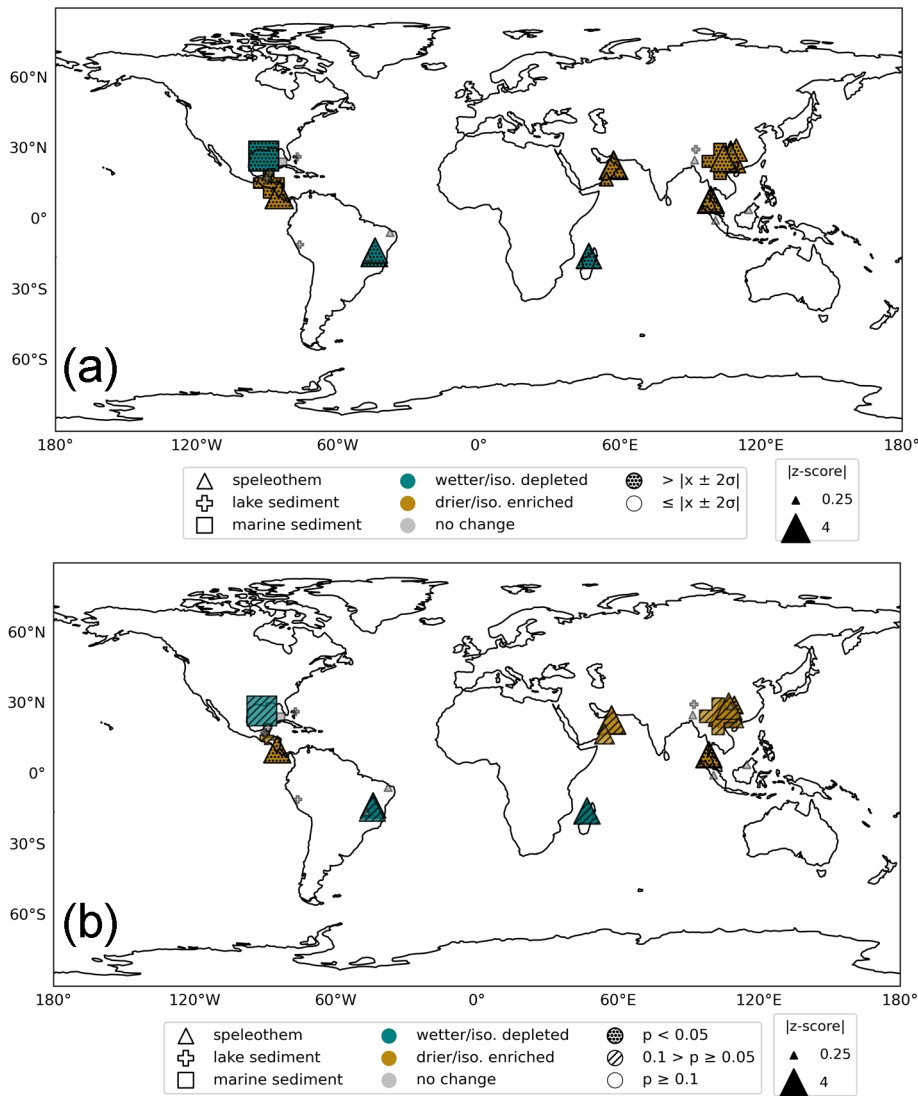

**Figure 2.** (a) Tropical hydroclimate anomalies detected in proxy archives over the 7.9-8.5ka interval using the method described in Morrill et al. (2013). Blue symbols represent wetter (and/or isotopically depleted) conditions while brown symbols represent drier (and/or isotopically enriched) conditions relative to each record's mean climatology over the 7.4-7.9ka and 8.5-9.0ka windows described in the text. Grey symbols indicate the locations of proxy data where no significant 8.2ka hydroclimate event was detected. Symbol size is scaled by $100 \times |$z-score$|$, calculated from the per-record mean and standard deviation over the 7ka-10ka interval. (b) Tropical hydroclimate anomalies detected in proxy archives over the 7.9-8.3ka ("significant") and 7.7-8.5ka ("tentative") intervals. Blue symbols represent wetter (and/or isotopically depleted) conditions while brown symbols represent drier (and/or isotopically enriched) conditions relative to each record's mean climatology over 7ka-10ka. Grey symbols indicate the locations of proxy data where no significant 8.2ka hydroclimate event was detected. Symbol size is scaled by $100 \times |$z-score$|$, calculated from the per-record mean and standard deviation over the 7ka-10ka interval.





**Figure 3.** Summary of the 8.2 ka events detected using our modified Morrill et al. (2013) method (MM; left) and actR (right) for the paleoclimate records included in this study. Records with drier and/or isotopically enriched events are shown in brown, records with wetter and/or isotopically depleted events are shown in green, and records in which no event was detected are shown in grey. Stippling indicates that a "significant" event was detected in a given record by actR with event "start" and "end" times within the 7.9-8.3ka interval at the $p < 0.05$ significance level. Slashed hatching indicates the presence of a "tentative" hydroclimate anomaly, with either a "significant" event detected outside of the 7.9-8.3ka window (between 7.7-8.5ka) or an event within that window where $0.1 > p \geq 0.05$. The archive type is indicated by the symbol (triangle = speleothem, plus = lake sediment, square = marine sediment).Symbol size is scaled by 100 ×|z-score|, calculated from the per-record mean and standard deviation over the 7ka-10ka interval. Symbols are shown mapped over the simulated anomalous (a, d) amount-weighted oxygen isotopic composition of precipitation, (b, e) precipitation amount, (c, f) and effective moisture from the last 50 years of the iCESM "hose" and "ctrl" experiments.

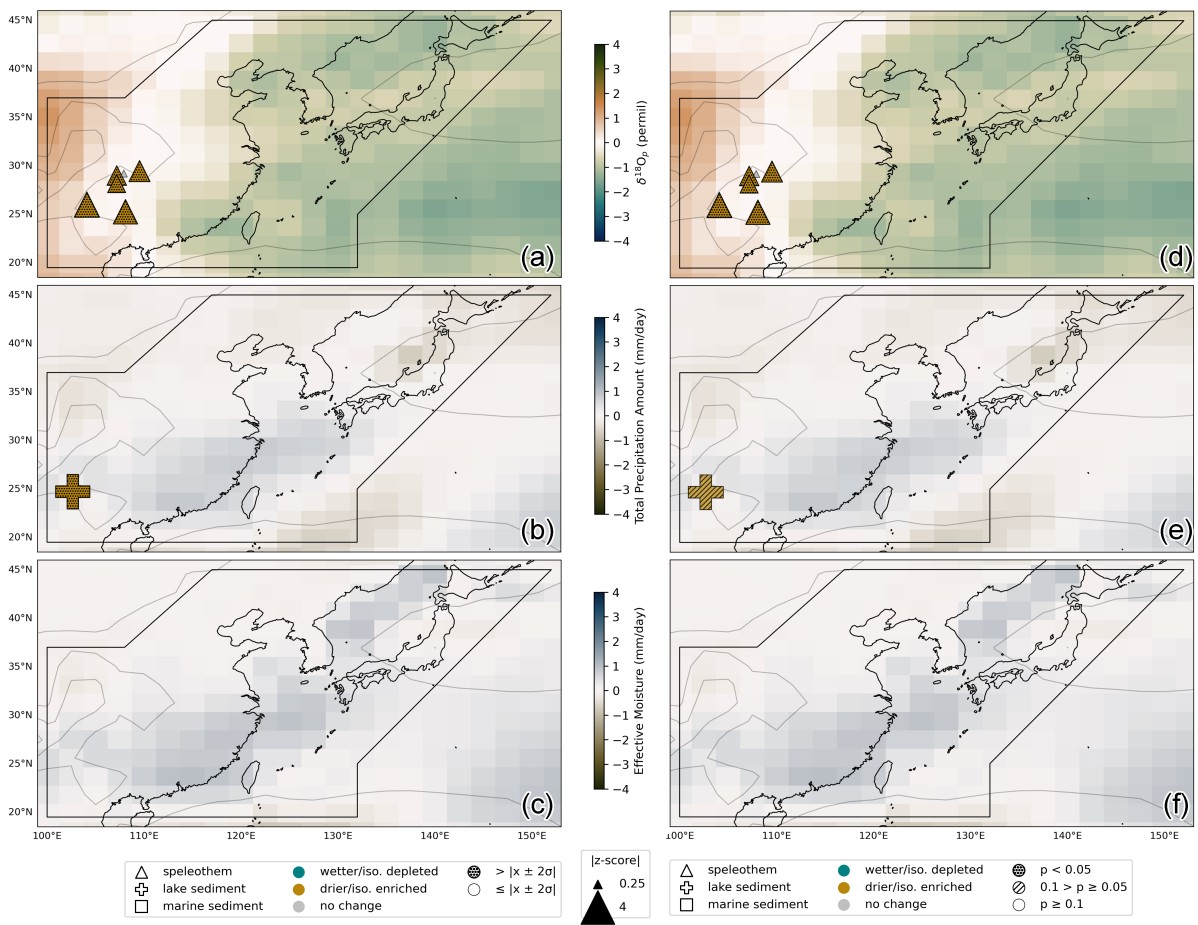

**Figure 4.** Data-model comparison of IPCC region 35: East Asia (box). Model shading represents (a) the precipitation amount-weighted $\delta^{18}O$ anomaly, (b) the precipitation amount anomaly, and (c) the effective moisture (precipitation minus evaporation) between the last 50 years of the "hose" and "ctrl" simulations. Symbols represent paleoclimate proxy archives within the region corresponding to each respective climate variable, where the brown shaded triangles indicate speleothem records with recorded dry hydroclimate/enriched isotopic anomalies during the 8.2ka Event and grey symbols indicate records with no hydroclimate anomalies ("no change") over the 8.3-7.9ka interval. For symbols showing an anomaly associated with the 8.2ka Event, size is scaled by $200 \times |z\text{-score}|$ relative to each record's mean and standard deviation.



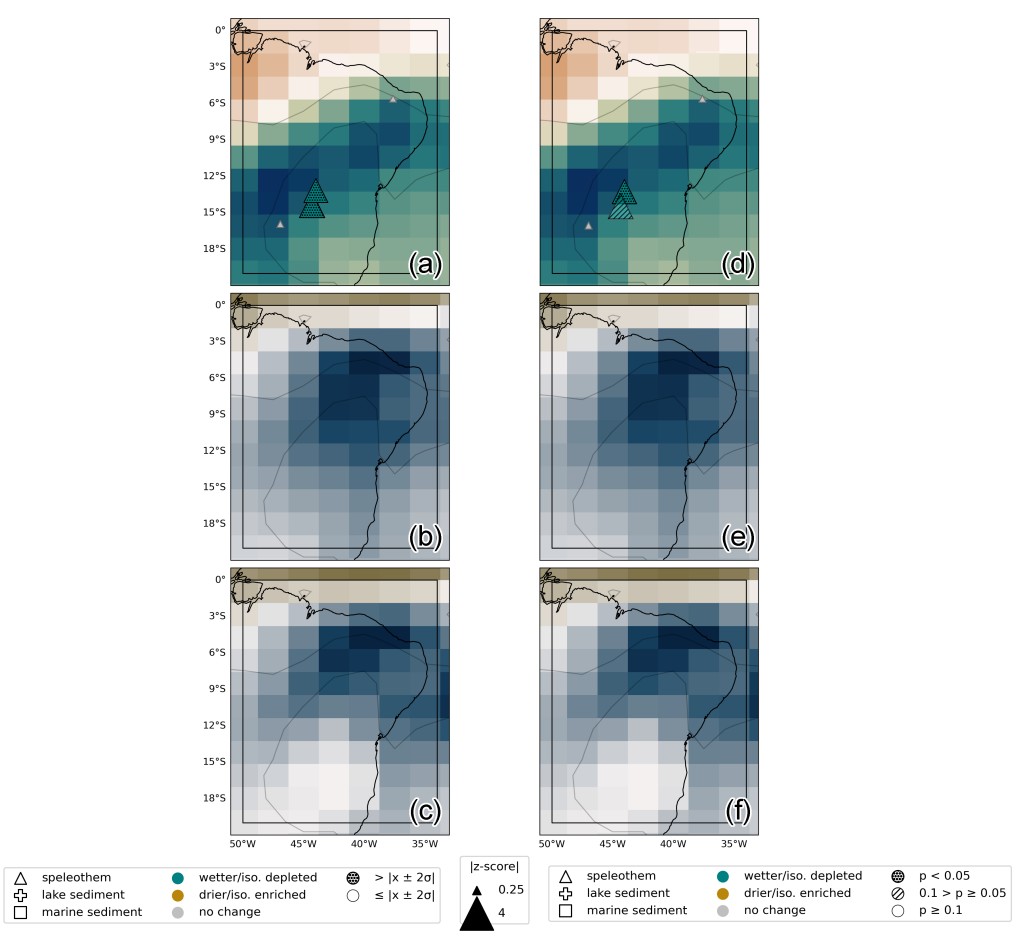

**Figure 5.** As in Fig. 4, but for IPCC region 11: northeastern South America (box).



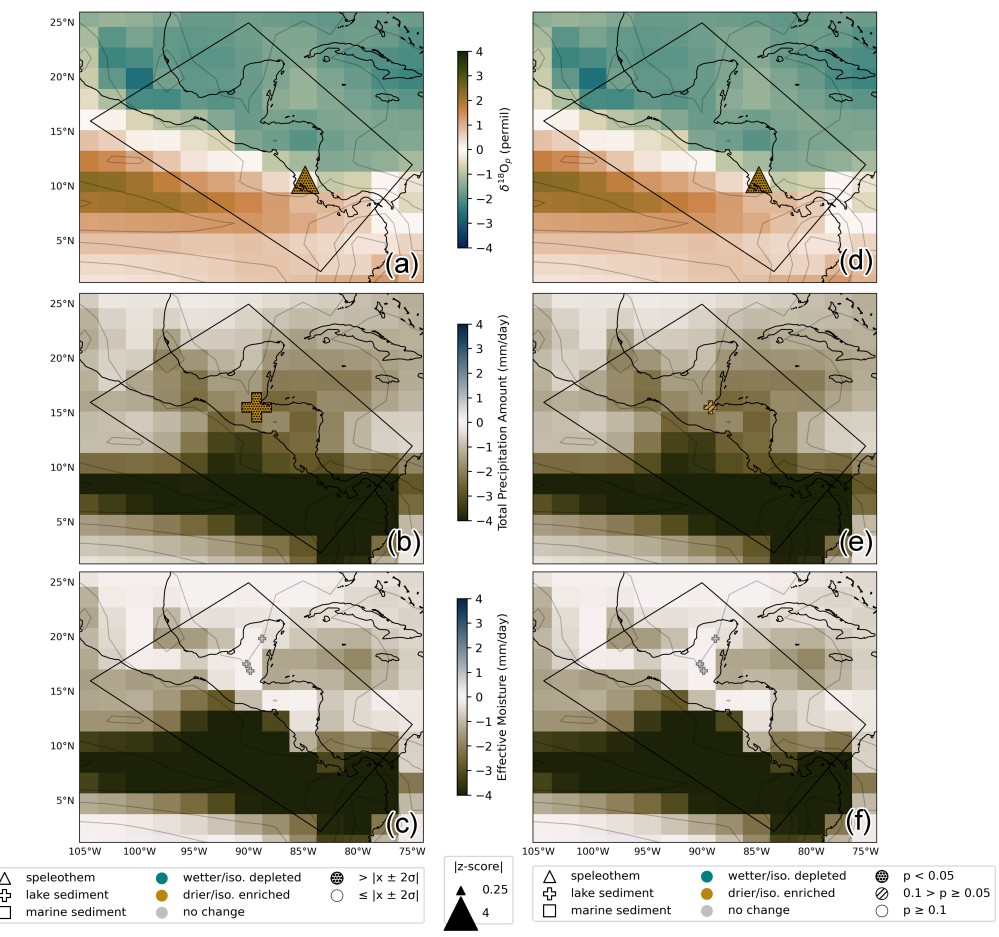

**Figure 6.** As in Fig. 4, but for IPCC region 7: southern Central America (box).



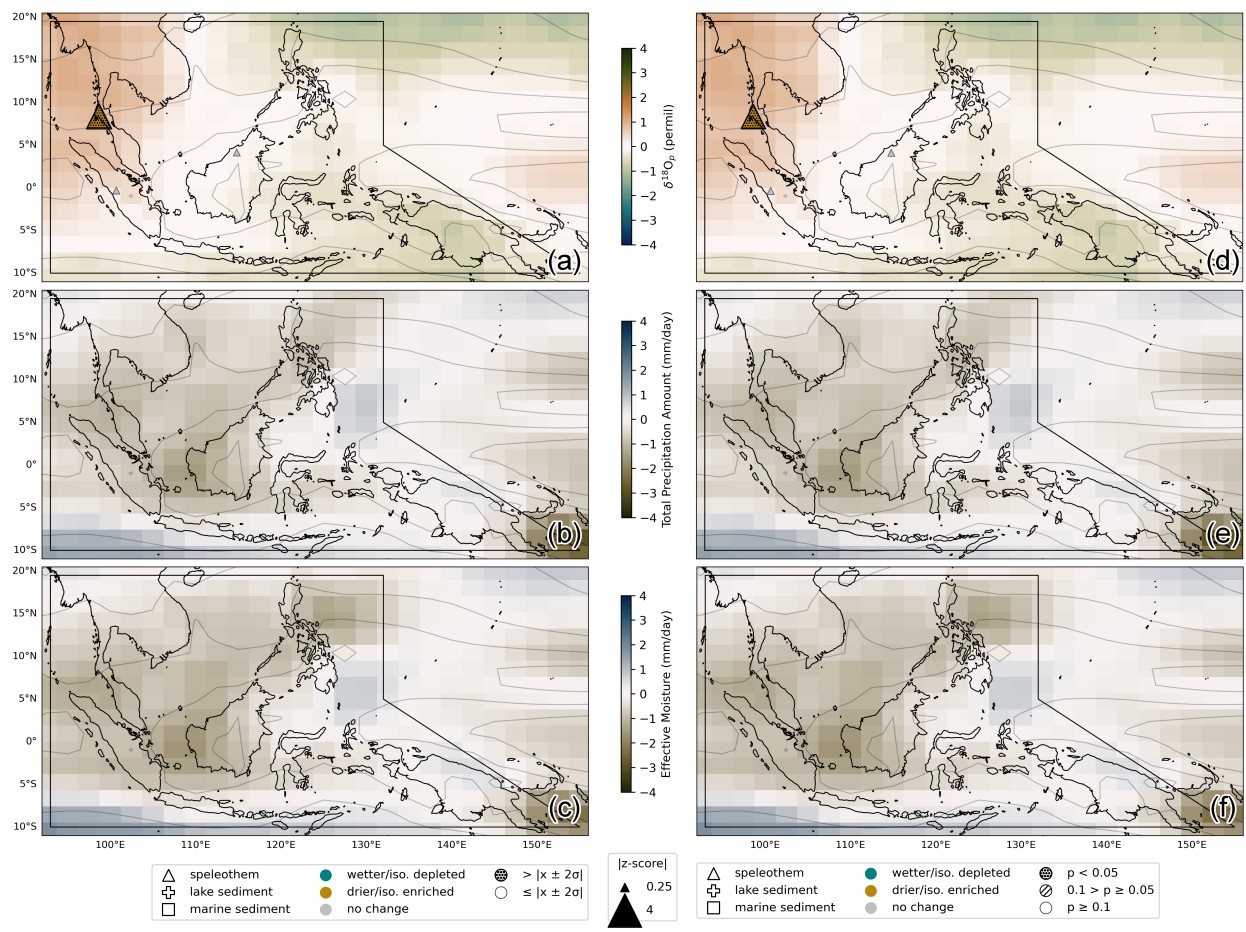

**Figure 7.** As in Fig. 4, but for IPCC region 38: Southeast Asia (box).



**Table 1.** Location metadata for all paleoclimate proxy datasets in this compilation.

| Record ID | Lat | Lon | IPCC Region | Site Name | Reference |
|---|---|---|---|---|---|
| ABC1 | -15.54 | 46.89 | Madagascar | Anjohibe Cave, Madagascar | Duan et al., 2021 |
| ANJB2 | -15.54 | 46.89 | Madagascar | Anjohibe Cave, Madagascar | Voarintsoa et al., 2017 |
| BA03 | 4.26 | 114.96 | S.E. Asia | Malaysian Borneo | Chen et al., 2016 |
| BTV21a | -27.22 | -49.16 | S.E. South America | Botuverá Cave, SE Brazil | Bernal et al., 2016 |
| C7 | 26.57 | -77.12 | E. North America | Great Cistern Sinkhole, Bahamas | Sullivan et al., 2021 |
| CM2013 | 22.38 | -83.97 | Caribbean | Santo Tomas Cave, Cuba | Fensterer et al., 2013 |
| CM2019 | 23.38 | -82.97 | Caribbean | Santo Tomas Cave, Cuba | Warken et al., 2019 |
| Core17940 | 20.12 | 117.38 | E. Asia | South China Sea | Wang et al., 1999 |
| Core5LI | 15.53 | -89.23 | S. Central America | Lake Izabal, Guatemala | Duarte et al., 2021 |
| CP | 22.38 | -83.97 | Caribbean | Dos Anas Cave, Cuba | Fensterer et al., 2013 |
| Curtis6VII93 | 16.92 | -89.83 | S. Central America | Lake Peten-Itza, Guatemala | Curtis et al., 1998 |
| D4Cheng | 25.28 | 108.08 | E. Asia | Dongge Cave, China | Cheng et al., 2009 |
| D4Dykoski | 25.28 | 108.08 | E. Asia | Dongge Cave, China | Dykoski et al., 2005 |
| EJConroy | -0.87 | -89.45 | Equatorial Pacific Ocean | El Junco Lake, Galapagos | Conroy et al., 2008 |
| F14 | 24.69 | 102.67 | E. Asia | Dianchi, Yunan, China | Hillman et al., 2021 |
| FR5 | 29.23 | 107.9 | E. Asia | Furong Cave, China | Li et al., 2011 |
| GB2GC1 | 26.67 | -93.92 | C. North America | Garrison Basin, Gulf of Mexico | Thirumalai et al., 2021 |
| GURM1 | 15.43 | -90.28 | S. Central America | Grutas del Rey Marcos, Guatemala | Winter et al., 2020 |
| H14 | 23.08 | 57.35 | Arabian Peninsula | Hoti Cave, Oman | Cheng et al., 2009 |
| H5 | 23.08 | 57.35 | Arabian Peninsula | Hoti Cave, Oman | Neff et al., 2001 |
| HF01 | 29.02 | 107.18 | E. Asia | Chongqing, Southwest China | Yang et al., 2019 |
| JAR7 | -21.08 | -56.58 | S.E. South America | Jaragua Cave, Brazil | Novello et al., 2017 |
| JPC51 | 24.41 | -83.22 | Caribbean | Florida Straits | Schmidt et al., 2012 |
| KM1 | 25.26 | 91.88 | S. Asia | Mawmluh Cave | Huguet et al., 2018 |
| KMA | 25.26 | 91.88 | S. Asia | Mawmluh cave | Berkelhammer et al., 2012 |
| KN51 | -15.18 | 128.37 | N. Australia | Cave KNI-51, Western Australia | Denniston et al., 2013 (a) |
| LagoPuertoArturo | 17.53 | -90.18 | S. Central America | Lago Puerto Arturo, Maya Lowlands | Wahl et al., 2014 |
| LBA99 | 8.33 | -71.78 | N. South America | Laguna Blanca, Venezuelan Andes | Polissar et al., 2013 |
| LC1 | 19.86 | -88.76 | S. Central America | Lake Chichancanab, Mexico | Hodell et al., 1995 |
| LG11 | -14.42 | -44.37 | N.E. South America | Lapa Grande Cave, Brazil | Strikis et al., 2011 |
| LH2 | 29.48 | 109.53 | E. Asia | Lianhua Cave, Hunan, China | Zhang et al., 2013 |
| LP | -10.7 | -76.06 | N.W. South America | Laguna Pumacocha, Peru | Bird et al., 2011 |
| LR06_B3_2013 | -8.53 | 120.43 | S.E. Asia | Liang Luar cave, western Flores, Indonesia | Ayliffe et al., 2013 |
| LSF19 | -16.15 | -44.6 | N.E. South America | Lapa Sem Fim Cave, Brazil | Azevedo et al., 2021 |

815





| Record ID | Lat | Lon | IPCC Region | Site Name | Reference |
|---|---|---|---|---|---|
| M981P | -10.27 | 34.32 | E. Southern Africa | Lake Malawi, Africa | Johnson et al., 2003 |
| MAW6 | 25.26 | 91.82 | S. Asia | Mawmluh Cave, India | Lechleitner et al., 2017 |
| MD022550 | 26.95 | -91.35 | C. North America | Gulf of Mexico | LoDico et al., 2006 |
| MWS1 | 25.26 | 91.88 | S. Asia | Mawmluh cave | Dutt et al., 2015 |
| NARC | -5.73 | -77.5 | N.W. South America | Cueva del Diamante, Peru | Cheng et al., 2013 |
| NCB | -5.94 | -77.31 | N.W. South America | Cueva del Tigre Perdido, Peru | van Breukelen et al., 2008 |
| PAD07 | -13.22 | -44.05 | N.E. South America | Padre Cave, Brazil | Cheng et al., 2009 |
| ParuCo | 29.8 | 92.35 | Tibetan Plateau | Paru Co, Tibetan Plateau, China | Bird et al., 2014 |
| PET-PI6 | 17 | -89.78 | S. Central America | Lake Petén Itzá, Guatemala | Escobar et al., 2012 |
| PLJJUN15 | -11.04 | -76.11 | N.W. South America | Lake Junín, Peruvian Andes | Woods et al., 2020 |
| Q52007 | 17.17 | 54.3 | Arabian Peninsula | Qunf Cave, Oman | Fleitmann et al., 2007 |
| Q5Cheng | 17.17 | 54.3 | Arabian Peninsula | Qunf Cave, Oman | Cheng et al., 2009 |
| RN1 | -5.58 | -37.64 | N.E. South America | Rainha cave, Brazil | Cruz et al., 2009 |
| RN4 | -5.58 | -37.64 | N.E. South America | Rainha cave, Brazil | Cruz et al., 2009 |
| SG1 | 28.18 | 107.17 | E. Asia | Shigao Cave, China | Jiang et al., 2012 |
| Sha3 | -5.7 | -77.9 | N.W. South America | Shatuca Cave, Peruvian Andes | Bustamante et al., 2016 |
| SSC01 | 4.1 | 114.83 | S.E. Asia | Gunung Mulu National Park, Borneo | Carolin et al., 2016 |
| Staubwasser63KA | 24.62 | 65.98 | S. Asia | Arabian Sea | Staubwasser et al., 2003 |
| T8 | -24.02 | 29.11 | E. Southern Africa | Makapansgat Valley, South Africa | Holmgren et al., 2003 |
| TK07 | 8.33 | 98.73 | S.E. Asia | Klang Cave, Thailand | Chawchai et al., 2021 |
| TK20 | 8.33 | 98.73 | S.E. Asia | Klang Cave, Thailand | Chawchai et al., 2021 |
| TM6 | -16 | -47 | N.E. South America | Tamboril Cave, Brazil | Ward et al., 2019 |
| TOW109B | -2.73 | 121.52 | S.E. Asia | Lake Towuti, Indonesia | Russell et al., 2014 |
| V1 | 10.6 | -84.8 | S. Central America | Costa Rica | Lachniet et al., 2004 |
| XBL29 | 24.2 | 103.36 | E. Asia | Xiaobailong cave, China | Cai et al., 2015 |
| ZLP1 | 26.02 | 104.1 | E. Asia | Zhuliuping Cave, China | Huang et al., 2016 |





**Table 2.** Archive and interpretation metadata for the paleoclimate proxy datasets used in this study. Tier 1 data meet all strict inclusion criteria, while Tier 2 data are deficient in either dating or data resolution over the 7ka-10ka interval. Tier 3 data meet none of the strict inclusion criteria and are not included in quantitative analyses. All foraminifera used in the compilation are G. ruber (white). BSi MAR is the biogenic silica mass accumulation rate, in mg $SiO^2/cm^2yr$.

| Record ID | Tier | Archive | Proxy | Interp. Group | Interp. Dir. | Reference |
|---|---|---|---|---|---|---|
| ABC1 | 1 | speleothem | $\delta^{18}O$ | precip. iso. | inverse | Duan et al., 2021 |
| ANJB2 | 1 | speleothem | $\delta^{18}O$ | precip. iso. | inverse | Voarintsoa et al., 2017 |
| BA03 | 1 | speleothem | $\delta^{18}O$ | precip. iso. | inverse | Chen et al., 2016 |
| BTV21a | 1 | speleothem | $\delta^{18}O$ | precip. iso. | inverse | Bernal et al., 2016 |
| C7 | 2 | lacustrine | grain size | precip. amt. | direct | Sullivan et al., 2021 |
| CM2013 | 1 | speleothem | $\delta^{18}O$ | precip. iso. | inverse | Fensterer et al., 2013 |
| CM2019 | 1 | speleothem | $\delta^{18}O$ | precip. iso. | inverse | Warken et al., 2019 |
| Core17940 | 1 | marine | $\delta^{18}O$ | eff. moisture | inverse | Wang et al., 1999 |
| Core5LI | 1 | lacustrine | Ti | precip. amt. | direct | Duarte et al., 2021 |
| CP | 1 | speleothem | $\delta^{18}O$ | precip. iso. | inverse | Fensterer et al., 2013 |
| Curtis6VII93 | 2 | lacustrine | $\delta^{18}O$ gastropod (Cochliopina sp.) | eff. moisture | inverse | Curtis et al., 1998 |
| D4Cheng | 1 | speleothem | $\delta^{18}O$ | precip. iso. | inverse | Cheng et al., 2009 |
| D4Dykoski | 1 | speleothem | $\delta^{18}O$ | precip. iso. | inverse | Dykoski et al., 2005 |
| EJConroy | 1 | lacustrine | clay (%) | eff. moisture | direct | Conroy et al., 2008 |
| F14 | 2 | lacustrine | magnetic susceptibility | precip. amt. | inverse | Hillman et al., 2021 |
| FR5 | 2 | speleothem | $\delta^{18}O$ | precip. iso. | inverse | Li et al., 2011 |
| GB2GC1 | 1 | marine | $\delta^{18}O$ | eff. moisture | inverse | Thirumalai et al., 2021 |
| GURM1 | 1 | speleothem | $\delta^{18}O$ | precip. iso. | inverse | Winter et al., 2020 |
| H14 | 1 | speleothem | $\delta^{18}O$ | precip. iso. | inverse | Cheng et al., 2009 |
| H5 | 1 | speleothem | $\delta^{18}O$ | precip. iso. | inverse | Neff et al., 2001 |
| HF01 | 1 | speleothem | $\delta^{18}O$ | precip. iso. | inverse | Yang et al., 2019 |
| JAR7 | 1 | speleothem | $\delta^{18}O$ | precip. iso. | inverse | Novello et al., 2017 |
| JPC51 | 1 | marine | $\delta^{18}O$ | eff. moisture | inverse | Schmidt et al., 2012 |
| KM1 | 1 | speleothem | $\delta^{18}O$ | precip. iso. | inverse | Huguet et al., 2018 |
| KMA | 1 | speleothem | $\delta^{18}O$ | precip. iso. | inverse | Berkelhammer et al., 2012 |
| KN51 | 1 | speleothem | $\delta^{18}O$ | precip. iso. | inverse | Denniston et al., 2013 (a) |
| LagoPuertoArturo | 1 | lacustrine | $\delta^{18}O$ | eff. moisture | inverse | Wahl et al., 2014 |
| LBA99 | 1 | lacustrine | magnetic susceptibility | precip. amt. | direct | Polissar et al., 2013 |
| LC1 | 1 | lacustrine | CaCO3 | eff. moisture | direct | Hodell et al., 1995 |
| LG11 | 1 | speleothem | $\delta^{18}O$ | precip. iso. | inverse | Strikis et al., 2011 |
| LH2 | 1 | speleothem | $\delta^{18}O$ | precip. iso. | inverse | Zhang et al., 2013 |



| Record ID | Tier | Archive | Proxy | Interp. Group | Interp. Dir. | Reference |
|-----------|------|---------|-------|---------------|--------------|-----------|
| LP | 2 | lacustrine | $\delta^{18}$O | precip. iso. | inverse | Bird et al., 2011 |
| LR06B32013 | 1 | speleothem | $\delta^{18}$O | precip. iso. | inverse | Ayliffe et al., 2013 |
| LSF19 | 1 | speleothem | $\delta^{18}$O | precip. iso. | inverse | Azevedo et al., 2021 |
| M981P | 2 | lacustrine | BSi MAR | precip. amt. | direct | Johnson et al., 2003 |
| MAW6 | 1 | speleothem | $\delta^{18}$O | precip. iso. | inverse | Lechleitner et al., 2017 |
| MD022550 | 1 | marine | $\delta^{18}$O | eff. moisture | inverse | LoDico et al., 2006 |
| MWS1 | 3 | speleothem | $\delta^{18}$O | precip. iso. | inverse | Dutt et al., 2015 |
| NARC | 1 | speleothem | $\delta^{18}$O | precip. iso. | inverse | Cheng et al., 2013 |
| NCB | 2 | speleothem | $\delta^{18}$O | precip. iso. | inverse | van Breukelen et al., 2008 |
| PAD07 | 1 | speleothem | $\delta^{18}$O | precip. iso. | inverse | Cheng et al., 2009 |
| ParuCo | 2 | lacustrine | Lithics (%) | precip. amt. | direct | Bird et al., 2014 |
| PETPI6 | 1 | lacustrine | magnetic susceptibility | eff. moisture | direct | Escobar et al., 2012 |
| PLJJUN15 | 1 | lacustrine | Ti | eff. moisture | direct | Woods et al., 2020 |
| Q52007 | 1 | speleothem | $\delta^{18}$O | precip. iso. | inverse | Fleitmann et al., 2007 |
| Q5Cheng | 1 | speleothem | $\delta^{18}$O | precip. iso. | inverse | Cheng et al., 2009 |
| RN1 | 1 | speleothem | $\delta^{18}$O | precip. iso. | inverse | Cruz et al., 2009 |
| RN4 | 1 | speleothem | $\delta^{18}$O | precip. iso. | inverse | Cruz et al., 2009 |
| SG1 | 1 | speleothem | $\delta^{18}$O | precip. iso. | inverse | Jiang et al., 2012 |
| Sha3 | 1 | speleothem | $\delta^{18}$O | precip. iso. | inverse | Bustamante et al., 2016 |
| SSC01 | 1 | speleothem | $\delta^{18}$O | precip. iso. | inverse | Carolin et al., 2016 |
| Staubwasser63KA | 1 | marine | foraminifera $\delta^{18}$O | eff. moisture | inverse | Staubwasser et al., 2003 |
| T8 | 1 | speleothem | $\delta^{18}$O | precip. iso. | direct | Holmgren et al., 2003 |
| TA122 | 1 | speleothem | $\delta^{18}$O | precip. iso. | inverse | Wurtzel et al., 2018 |
| TK07 | 1 | speleothem | $\delta^{18}$O | precip. iso. | inverse | Chawchai et al., 2021 |
| TK20 | 1 | speleothem | $\delta^{18}$O | precip. iso. | inverse | Chawchai et al., 2021 |
| TM6 | 1 | speleothem | $\delta^{18}$O | precip. iso. | inverse | Ward et al., 2019 |
| TOW109B | 2 | lacustrine | Ti (cps) | precip. amt. | direct | Russell et al., 2014 |
| V1 | 1 | speleothem | $\delta^{18}$O | precip. iso. | inverse | Lachniet et al., 2004 |
| XBL29 | 2 | speleothem | $\delta^{18}$O | precip. iso. | inverse | Cai et al., 2015 |
| ZLP1 | 1 | speleothem | $\delta^{18}$O | precip. iso. | inverse | Huang et al., 2016 |

820





**Table 3.** Start, end, and duration of 8.2 ka Event calculated from changes in mean detected by actR and our modified Morrill et al. (2013) method for the global compilation and the four regions discussed in this study.

| Region | Statistic | Event Start (yr BP) | Event End (yr BP) | Event Duration (yrs) |
|---|---|---|---|---|
| | $n = 18$ | | | |
| | Average | 8282 | 8130 | 152 |
| Global | Median | 8283 | 8105 | 133 |
| | Min | 8106 | 8029 | 50 |
| | Max | 8489 | 8337 | 289 |
| | SD | 116 | 85 | 70 |
| | $n = 6$ | | | |
| | Average | 8284 | 8133 | 151 |
| East Asia | Median | 8306 | 8071 | 139 |
| | Min | 8106 | 8044 | 62 |
| | Max | 8489 | 8337 | 259 |
| | SD | 138 | 117 | 75 |
| | $n = 2$ | | | |
| | Average | 8291 | 8176 | 116 |
| Southeast Asia | Median | 8291 | 8176 | 116 |
| | Min | 8285 | 8155 | 101 |
| | Max | 8297 | 8196 | 130 |
| | SD | 8 | 29 | 21 |
| | $n = 2$ | | | |
| | Average | 8329 | 8204 | 125 |
| Northeast South America | Median | 8329 | 8204 | 125 |
| | Min | 8215 | 8165 | 50 |
| | Max | 8442 | 8242 | 200 |
| | SD | 161 | 54 | 106 |
| | $n = 2$ | | | |
| | Average | 8175 | 8069 | 106 |
| South Central America | Median | 8175 | 8069 | 106 |
| | Min | 8163 | 8051 | 77 |
| | Max | 8186 | 8086 | 135 |
| | SD | 16 | 25 | 41 |



**Table 4.** Regional and global summary of 8.2 ka events detected by actR and our modified Morrill et al. (2013) classification method, separated by the sign of the anomaly ("wetter", "drier", and "no change").

| IPCC Region | n | % of total | wetter | drier | no change | % of regional records w/ agreed "events" | "significant" actR events | "tentative" actR events | no actR events |
|---|---|---|---|---|---|---|---|---|---|
| Madagascar | 2 | 3.3 | 2 | 0 | 0 | 100 | 1 | 1 | 0 |
| S.E.Asia | 7 | 11.5 | 0 | 2 | 1 | 42.9 | 2 | 1 | 4 |
| S.E.South-America | 2 | 3.3 | 0 | 0 | 0 | 0 | 0 | 2 | 0 |
| E.North-America | 1 | 1.6 | 0 | 0 | 1 | 100 | 0 | 0 | 1 |
| Caribbean | 4 | 6.6 | 0 | 0 | 1 | 25 | 0 | 3 | 1 |
| E.Asia | 10 | 16.4 | 0 | 6 | 1 | 70 | 4 | 4 | 2 |
| S.Central-America | 7 | 11.5 | 0 | 2 | 3 | 71.4 | 1 | 1 | 5 |
| Equatorial.Pacific-Ocean | 1 | 1.6 | 0 | 0 | 0 | 0 | 0 | 0 | 1 |
| C.North-America | 2 | 3.3 | 1 | 0 | 0 | 50 | 1 | 1 | 0 |
| Arabian-Peninsula | 4 | 6.6 | 0 | 3 | 0 | 75 | 1 | 3 | 0 |
| S.Asia | 5 | 8.2 | 0 | 0 | 1 | 20 | 0 | 4 | 1 |
| N.Australia | 1 | 1.6 | 0 | 0 | 0 | 0 | 1 | 0 | 0 |
| N.South-America | 1 | 1.6 | 0 | 0 | 0 | 0 | 1 | 0 | 0 |
| N.E.South-America | 6 | 9.8 | 2 | 0 | 2 | 66.7 | 1 | 2 | 3 |
| N.W.South-America | 5 | 8.2 | 0 | 0 | 1 | 20 | 1 | 2 | 2 |
| E.Southern-Africa | 2 | 3.3 | 0 | 0 | 0 | 0 | 2 | 0 | 0 |
| Tibetan-Plateau | 1 | 1.6 | 0 | 0 | 1 | 100 | 0 | 0 | 1 |
| Global | 61 | 100 | 5 | 13 | 12 | - | 16 | 24 | 21 |



**Table 5.** The timing, duration, magnitude, and interpretation of the 8.2ka Event for records with agreement between MM and actR methods.

| IPCC Region | Record ID | Event Start (yr BP) | Event End (yr BP) | Event Duration (yrs) | MM z-score | actR z-score | Interpretation |
|---|---|---|---|---|---|---|---|
| Madagascar | ABC1 | 8248 | 8029 | 219 | -2.5 | -2.5 | wetter |
| | ANJB2 | 8318 | 8124 | 194 | -2.7 | -3.0 | wetter |
| E.Asia | D4Dykoski | 8106 | 8044 | 62 | 2.8 | 2.8 | drier |
| | F14 | 8489 | 8337 | 152 | 5.5 | 5.8 | drier |
| | FR5 | N/A | N/A | N/A | N/A | N/A | no change |
| | LH2 | 8158 | 8068 | 90 | 2.1 | 1.3 | drier |
| | HF01 | 8332 | 8073 | 259 | 1.8 | 3.0 | drier |
| | ZLP1 | 8339 | 8213 | 126 | 3.0 | 2.9 | drier |
| | SG1 | 8280 | 8062 | 218 | 1.5 | 2.9 | drier |
| Arabian-Peninsula | H14 | 8208 | 8080 | 128 | 3.5 | 3.5 | drier |
| | H5 | 8135 | 8042 | 93 | 2.9 | 3.2 | drier |
| | Q52007 | 8407 | 8199 | 208 | 0.8 | 1.7 | drier |
| Tibetan-Plateau | ParuCo | N/A | N/A | N/A | N/A | N/A | no change |
| S.Asia | KMA | N/A | N/A | N/A | N/A | N/A | no change |
| S.E.Asia | SSC01 | N/A | N/A | N/A | N/A | N/A | no change |
| | TK07 | 8297 | 8196 | 101 | 3.1 | 2.9 | drier |
| | TK20 | 8285 | 8155 | 130 | 2.5 | 2.5 | drier |
| Caribbean | JPC51 | N/A | N/A | N/A | N/A | N/A | no change |
| C.North-America | MD022550 | 8469 | 8180 | 289 | -3.8 | -3.8 | wetter |
| E.North-America | C7 | N/A | N/A | N/A | N/A | N/A | no change |
| S.Central-America | LC1 | N/A | N/A | N/A | N/A | N/A | no change |
| | V1 | 8186 | 8051 | 135 | 3.4 | 3.1 | drier |
| | Core5LI | 8163 | 8086 | 77 | -4.0 | -0.8 | drier |
| | Curtis6VII93 | N/A | N/A | N/A | N/A | N/A | no change |
| | LagoPuertoArturo | N/A | N/A | N/A | N/A | N/A | no change |
| N.W.South-America | LP | N/A | N/A | N/A | N/A | N/A | no change |
| N.E.South-America | PAD07 | 8215 | 8165 | 50 | -2.7 | -2.7 | wetter |
| | LG11 | 8442 | 8242 | 200 | -3.0 | -2.9 | wetter |
| | RN1 | N/A | N/A | N/A | N/A | N/A | no change |
| | TM6 | N/A | N/A | N/A | N/A | N/A | no change |

825



**Appendix A**



**Table A1.** Age model information.

| Record ID | Published Age Model Algorithm | Published 14C Cal. Curve | Contains Hiatus? | Contains Reversal? | In SISALv2? | Age Model Chosen |
|---|---|---|---|---|---|---|
| ABC1 | MOD-AGE | N/A | N | Y | N | Bacon |
| ANJB2 | StalAge | N/A | Y | Y | Y | SISAL Bacon |
| BA03 | StalAge | N/A | N | N | Y | SISAL Bacon |
| BTV21a | unknown | N/A | N | N | Y | SISAL Bacon |
| C7 | Bacon | IntCal13 | N | N | N/A | Bacon |
| CM2013 | StalAge | N/A | N | N | Y | SISAL copRa |
| CM2019 | StalAge | N/A | N | N | Y | SISAL Bacon |
| Core17940 | CALIB 3.0.3 | unknown | N | Y | N/A | Bacon |
| Core5LI | Bacon | IntCal20 | N | Y | N/A | Bacon |
| CP | StalAge | N/A | N | Y | Y | SISAL Bchron |
| Curtis6VII93 | linear interpolation | unknown | N | N | N/A | Bacon |
| D4Cheng | unknown | N/A | N | N | Y | Bacon |
| D4Dykoski | linear interpolation | N/A | N | N | Y | Bacon |
| EJConroy | CALIB 5.0 | unknown | N | N | N/A | Bacon |
| F14 | Bacon | IntCal20 | N | Y | N/A | Bacon |
| FR5 | unknown | IntCal09 | N | N | Y | SISAL copRa |
| GB2GC1 | Bacon | Marine13 | N | N | N/A | Bacon |
| GURM1 | COPRA | N/A | N | N | N | SISAL Bacon |
| H14 | unknown | N/A | N | N | Y | Bacon |
| H5 | unknown | N/A | N | Y | Y | SISAL Bacon |
| HF01 | polynomial fit | N/A | N | N | N | SISAL copRa |
| JAR7 | linear interpolation | N/A | N | N | Y | SISAL Bacon |
| JPC51 | CALIB 6.0 | unknown | N | N | N/A | Bacon |
| KM1 | StalAge | N/A | N | N | Y | SISAL Bchron |
| KMA | StalAge | N/A | N | N | Y | SISAL Bacon |
| KN51 | unknown | N/A | N | Y | Y | SISAL copRa |
| LagoPuertoArturo | CLAM 2.2 | IntCal13 | N | N | N/A | Bacon |
| LBA99 | linear interpolation | IntCal04 | Y | N | N/A | Bacon |
| LC1 | CALIB | unknown | N | Y | N/A | Bacon |
| LG11 | unknown | N/A | N | N | Y | SISAL Bacon |
| LH2 | linear interpolation | N/A | N | N | Y | SISAL Bacon |
| LP | CALIB 5.0 | unknown | N | N | N/A | Bacon |
| LR06B32013 | linear interpolation | N/A | N | N | Y | SISAL Bchron |



| Record ID | Published Age Model Algorithm | Published 14C Cal. Curve | Contains Hiatus? | Contains Reversal? | In SISALv2? | Age Model Chosen |
|---|---|---|---|---|---|---|
| LSF19 | unknown | N/A | Y | N | N | SISAL Bacon |
| M981P | CALIB 4.3 | unknown | N | N | N/A | Bacon |
| MAW6 | COPRA | N/A | N | Y | Y | SISAL Bchron |
| MD02_2550 | CALIB 5.0 | unknown | N | N | N/A | Bacon |
| NARC | linear interpolation | N/A | N | N | Y | SISAL copRa |
| NCB | Isoplot 3 | N/A | N | N | Y | SISAL Bacon |
| PAD07 | unknown | N/A | N | N | N | Bacon |
| ParuCo | CALIB 6.0 | IntCal09 | N | N | N/A | Bacon |
| PET-PI6 | OxCal | IntCal09 | N | N | N/A | Bacon |
| PLJ-JUN15 | Bacon | IntCal13 | N | N | N/A | Bacon |
| Q52007 | linear interpolation | N/A | N | N | Y | Bacon |
| Q5Cheng | unknown | N/A | N | N | Y | Bacon |
| RN1 | unknown | N/A | N | N | Y | SISAL Bchron |
| RN4 | unknown | N/A | N | N | Y | SISAL Bchron |
| SG1 | linear interpolation | N/A | N | N | Y | SISAL Bacon |
| Sha3 | COPRA | N/A | N | N | Y | SISAL Bchron |
| SSC01 | StalAge | N/A | N | N | Y | Bacon |
| Staubwasser63KA | least-squares | IntCal98 | N | N | N/A | Bacon |
| T8 | linear interpolation | N/A | N | N | Y | Bacon |
| TA122 | Bacon | N/A | N | N | Y | SISAL copRa |
| TK07 | Bacon | N/A | N | N | N | SISAL Bacon |
| TK20 | Bacon | N/A | N | N | N | SISAL Bacon |
| TM6 | COPRA | N/A | N | N | Y | SISAL Bacon |
| TOW109B | CALIB 6.0 | unknown | N | N | N/A | Bacon |
| V1 | fifth-order polynomial best-fit age model | N/A | N | Y | Y | SISAL Bacon |
| XBL29 | linear interpolation | N/A | N | N | Y | SISAL Bchron |
| ZLP1 | linear interpolation | N/A | N | N | Y | SISAL Bacon |



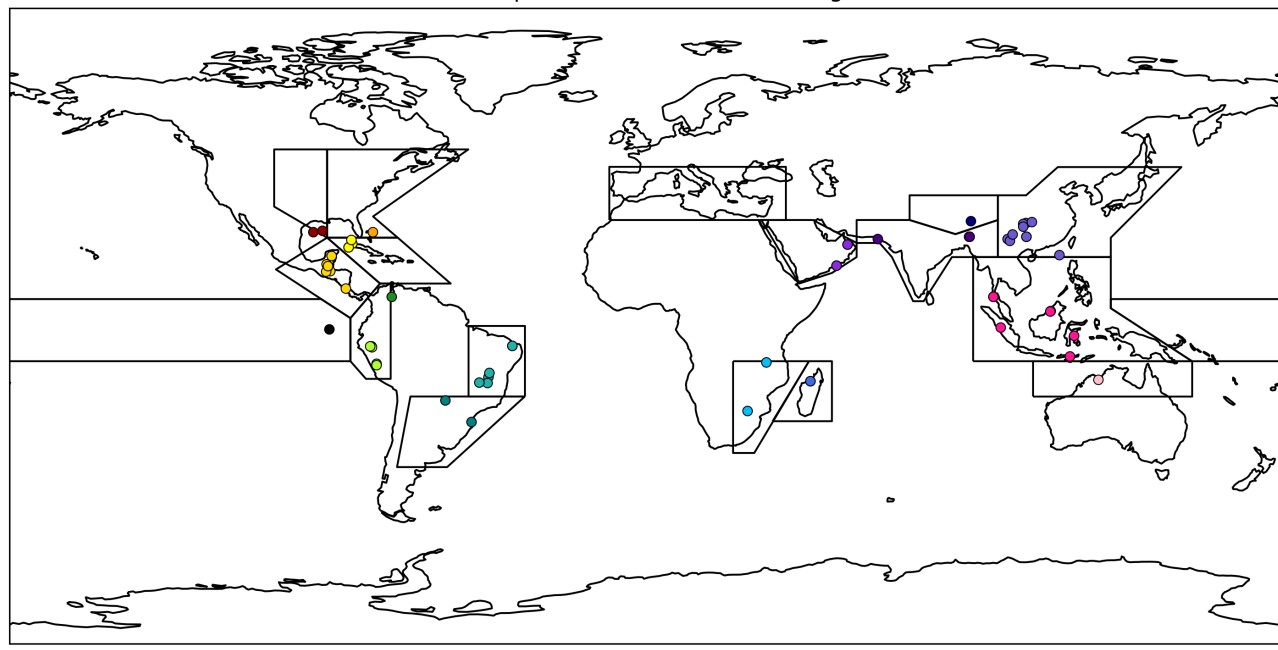

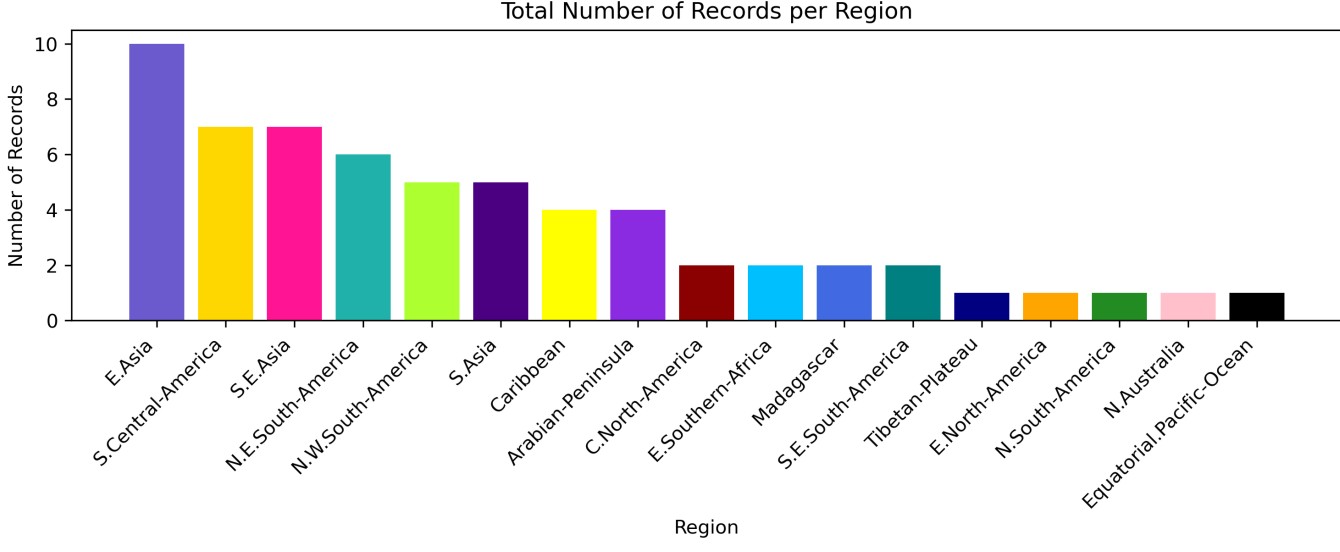

**Figure A1.** Locations of proxy records within climate reference regions defined in Iturbide et al. (2020).





**Figure A2.** The difference in surface air temperatures between the last 50 years of the "hose" and "ctrl" simulations, overlaid by contour intervals indicating the range of temperatures in the "ctrl" simulation over the full 100 years. Blue shaded areas represent anomalously cold regions, while anomalously warm regions are shaded in red on a global (a) and tropical (b) level.




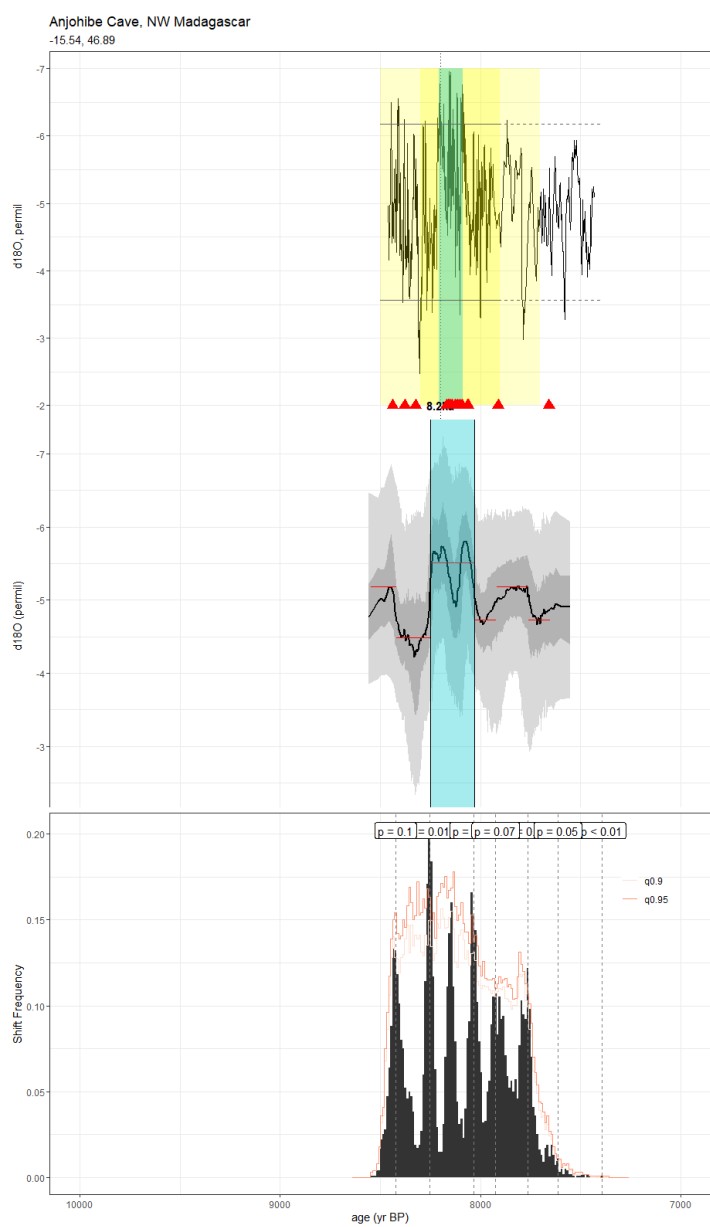

**Figure A3.** A stackplot from the speleothem record of Duan et al., 2021 (ABC1). The top panel shows the raw oxygen isotope time series with 8.3-7.9ka highlighted in darker yellow and 8.5-7.7ka highlighted in lighter yellow. The middle panel shows the same time series with age and paleodata ensemble uncertainty ribbons. The horizontal red lines represent mean values assigned to the data by actR, with discontinuities indicating significant changepoints. The lower panel depicts the frequency of shifts detected in the ensemble dataset (black) relative to 100 null hypothesis surrogate datasets (orange). The age model in the original publication was based on the MOD-AGE algorithm, while the age model used in this synthesis was constructed using the geoChronR package and BACON algorithm.





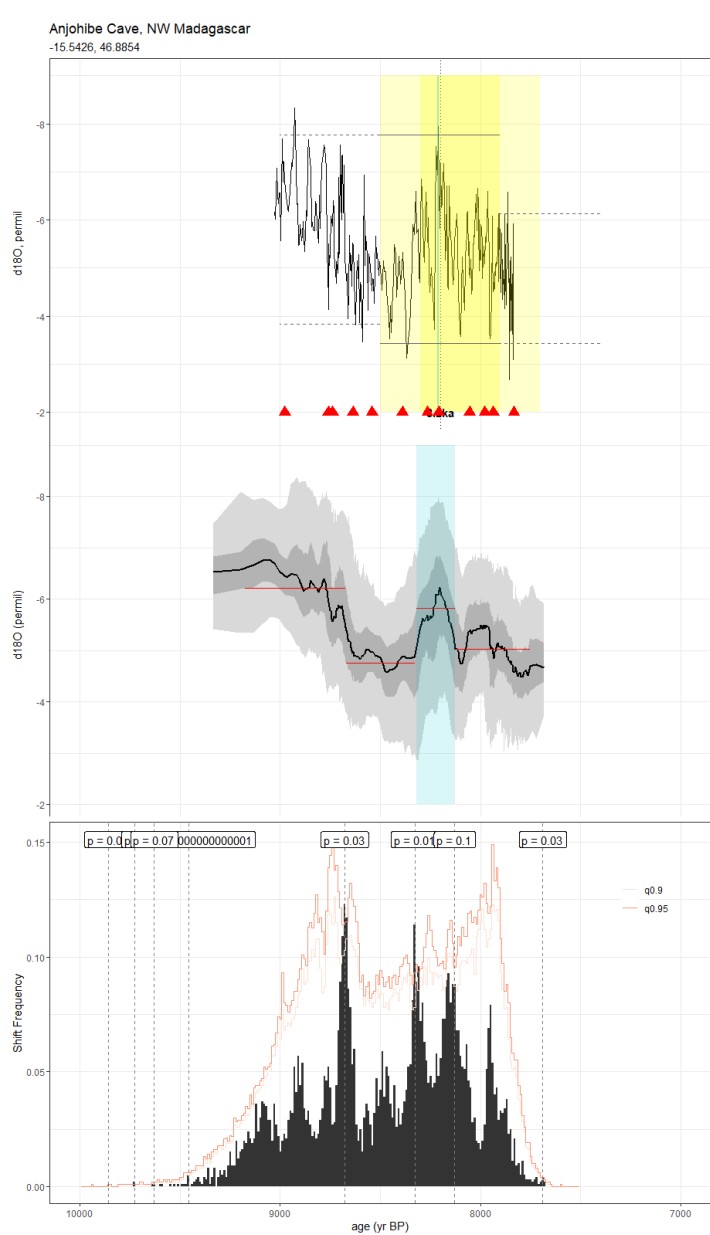

**Figure A4.** As in Fig. A3, but for the speleothem record of Voarintsoa et al., 2017 (ANJB2). The age model of the original publication was constructed using StalAge. Here, we used the BACON age ensemble from SISALv2.




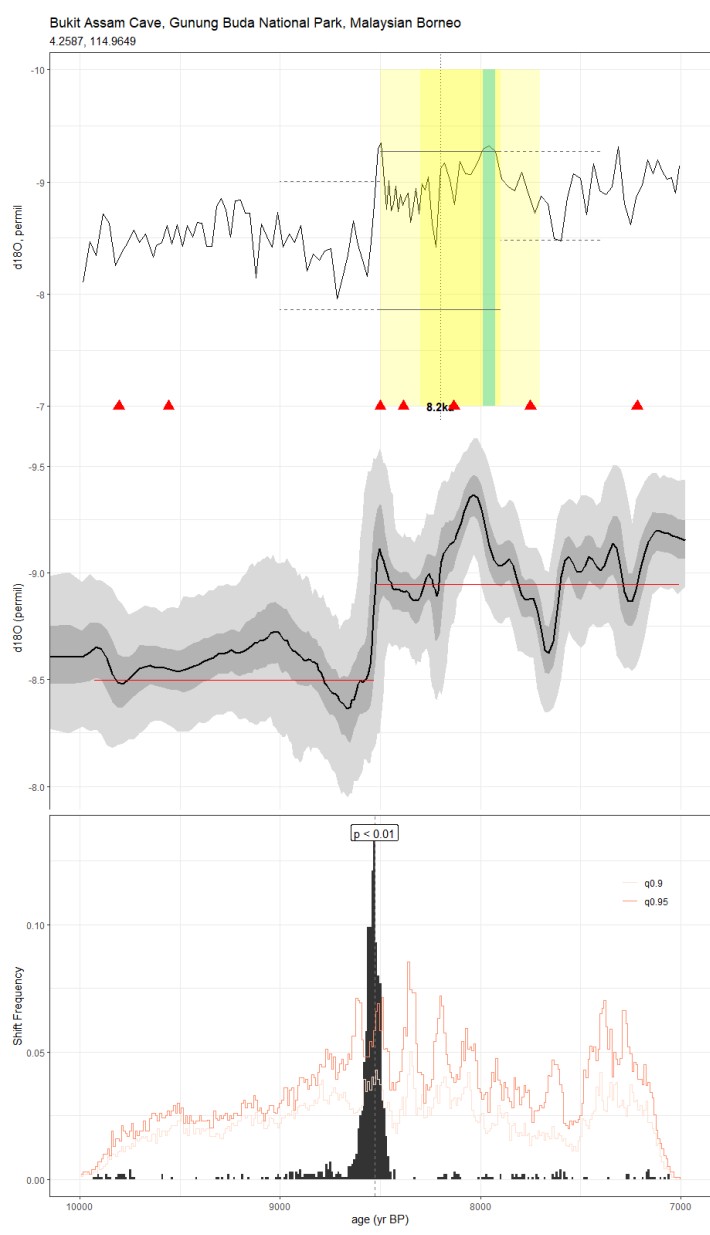

**Figure A5.** As in Fig. A3, but for the speleothem record of Chen et al., 2016 (BA03). The published age model was based on the StalAge algorithm, but here, we use the BACON ensemble from SISALv2.





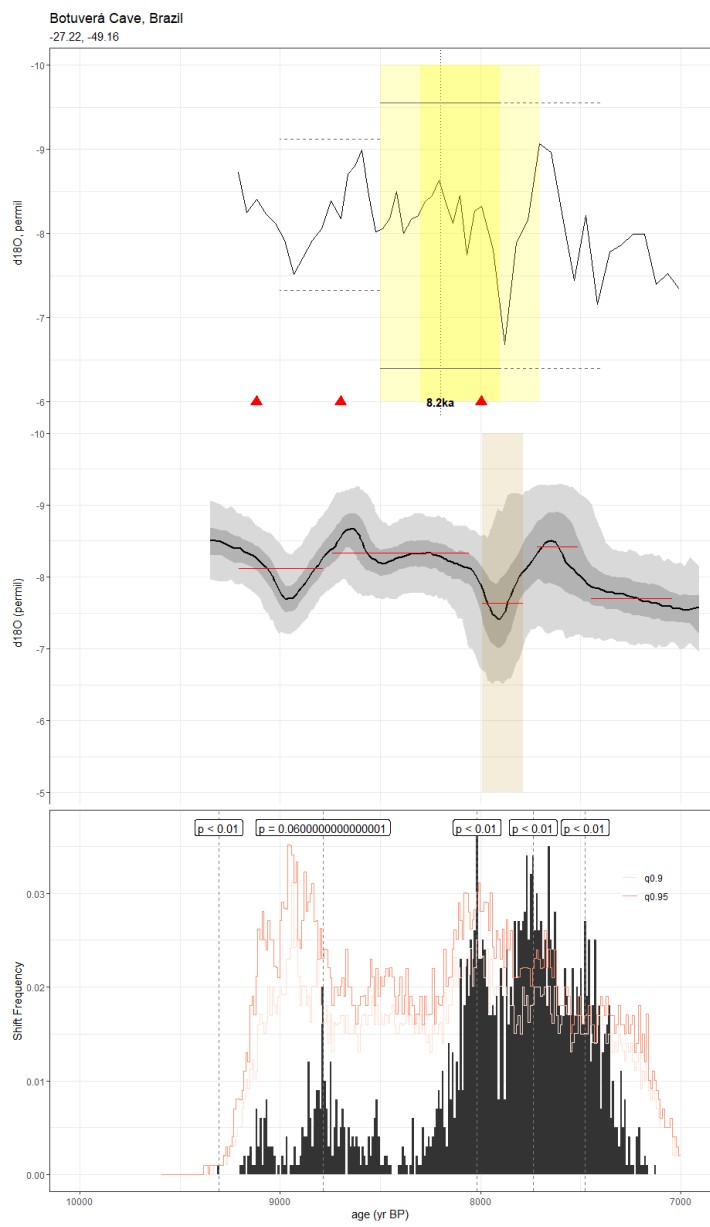

**Figure A6.** As in Fig. A3, but for the speleothem record of Bernal et al., 2016 (BTV21a). Information about the published age model was unreported. Here, we use the SISALv2 BACON age ensemble.





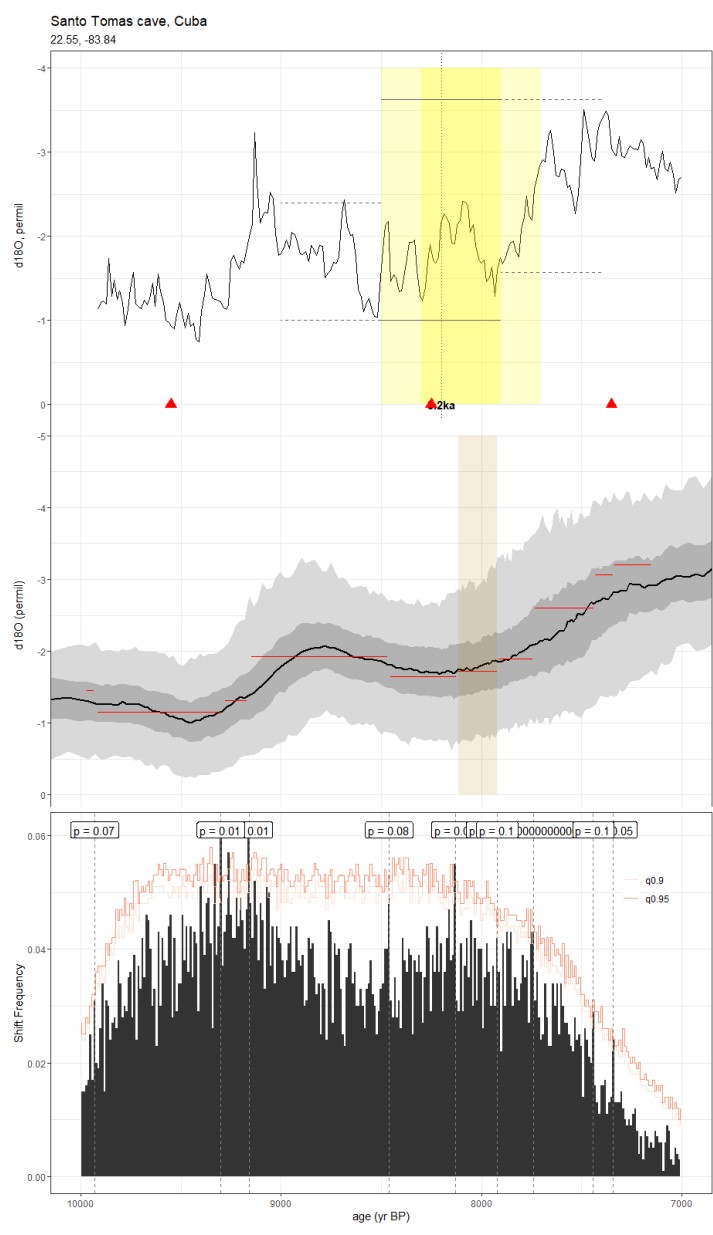

**Figure A8.** As in Fig. A3, but for the speleothem record of Fensterer et al., 2013 (CM2013). The published age model was constructed using the StalAge algorithm. Here, we use the SISALv2 copRa age ensemble.



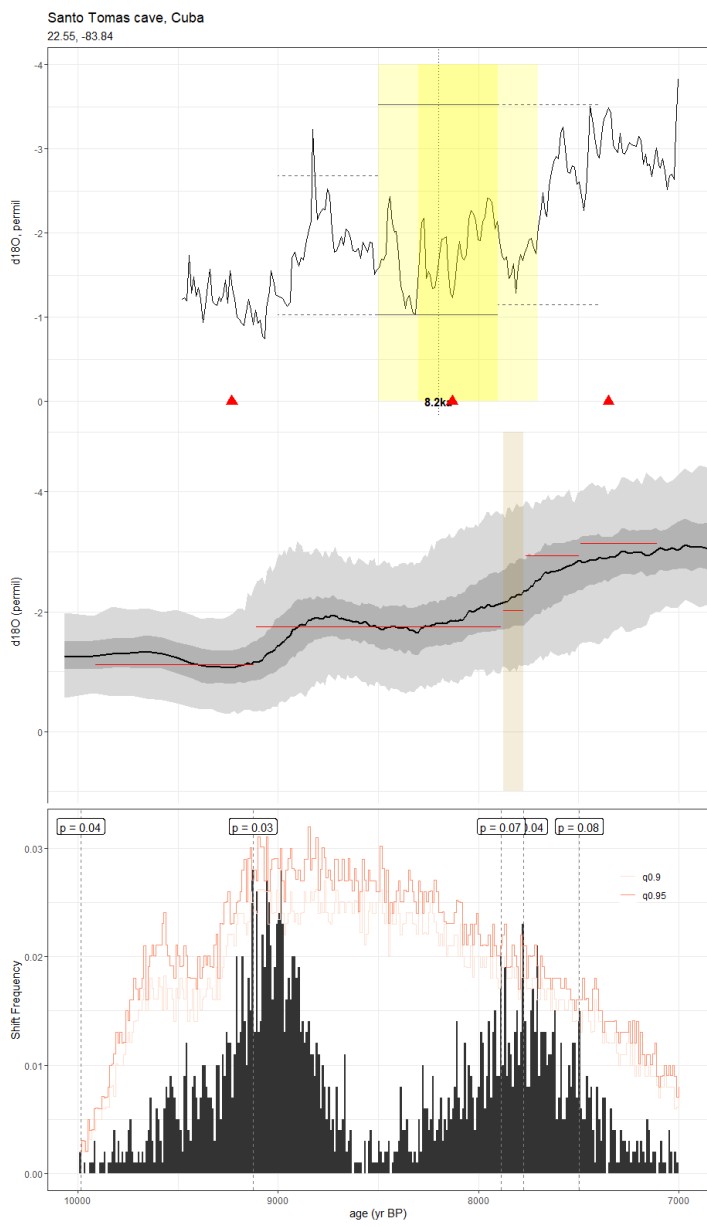

**Figure A9.** As in Fig. A3, but for the speleothem record of Warken et al., 2019 (CM2019). The published age model was constructed using the StalAge algorithm. Here, we use the SISALv2 BACON age ensemble.



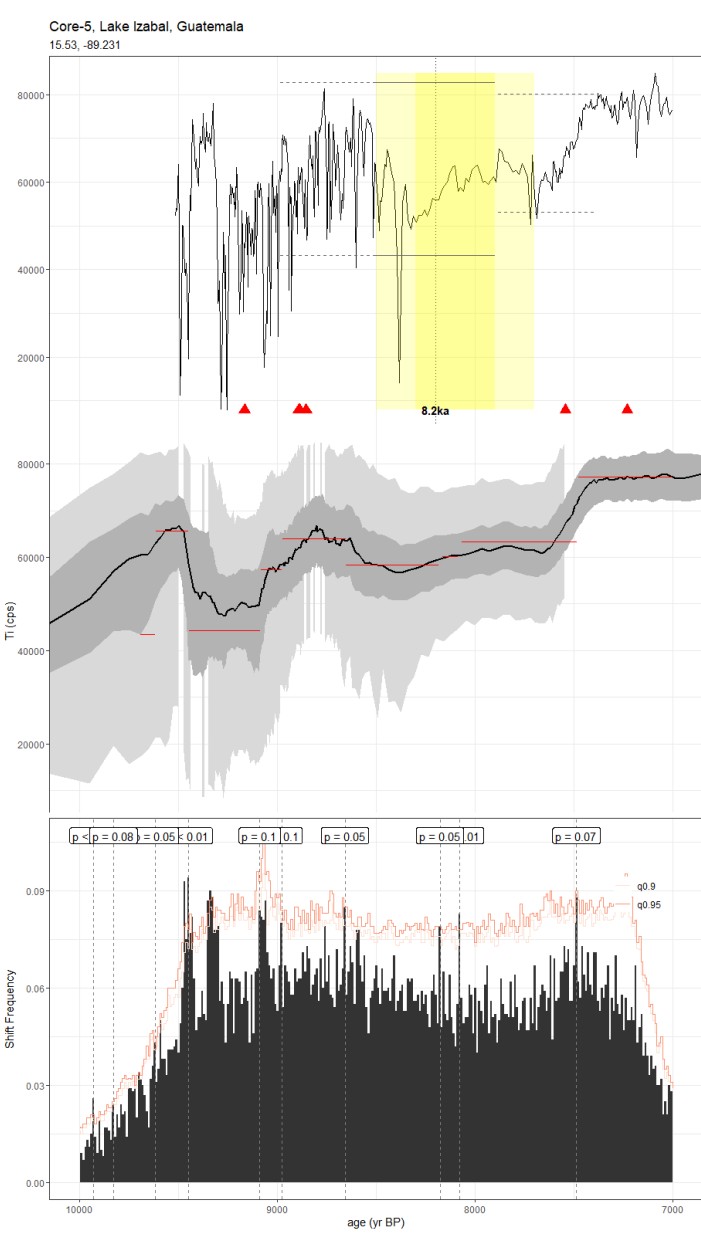

**Figure A10.** As in Fig. A3, but for the lacustrine titanium content record of Duarte et al., 2021 (Core5LI). The published age model was constructed using BACON using the IntCal20 calibration curve, and here, we construct our age ensemble using the BACON algorithm included with geoChronR.




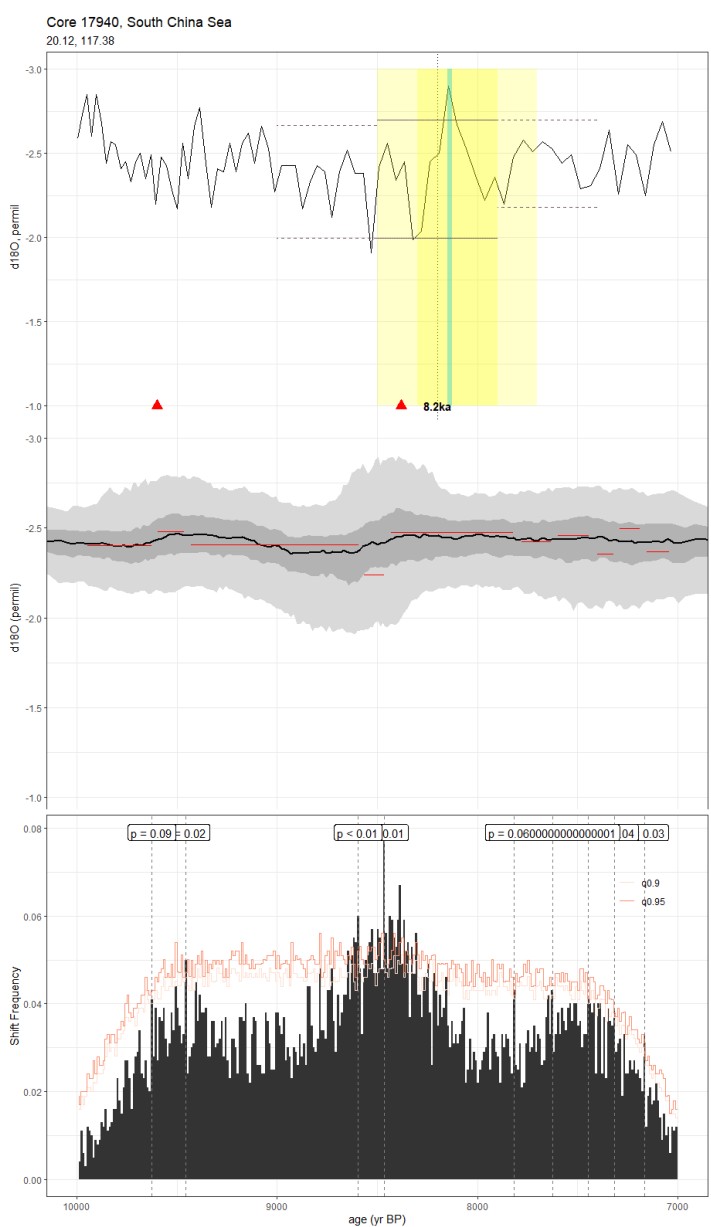

**Figure A11.** As in Fig. A3, but for the foraminifera record of Wang et al., 1999 (Core17940). The published age model was constructed using CALIB 3.0.3, corrected for a 400-year reservoir age and unspecified calibration curve. We constructed our age ensemble using the BACON algorithm included in geoChronR using the IntCal20 calibration curve.





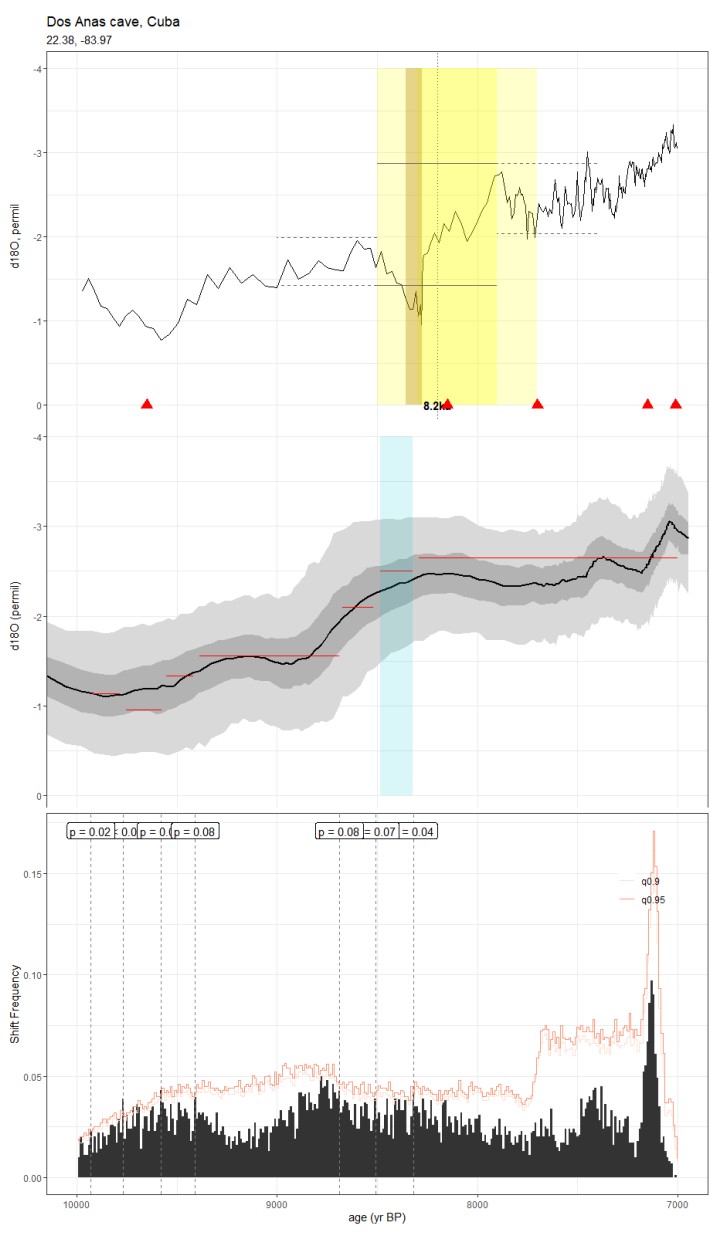

**Figure A12.** As in Fig. A3, but for the speleothem record of Fensterer et al., 2013 (CP). The published age model was constructed using the StalAge algorithm. We use the Bchron age ensemble constructed in SISALv2 here.





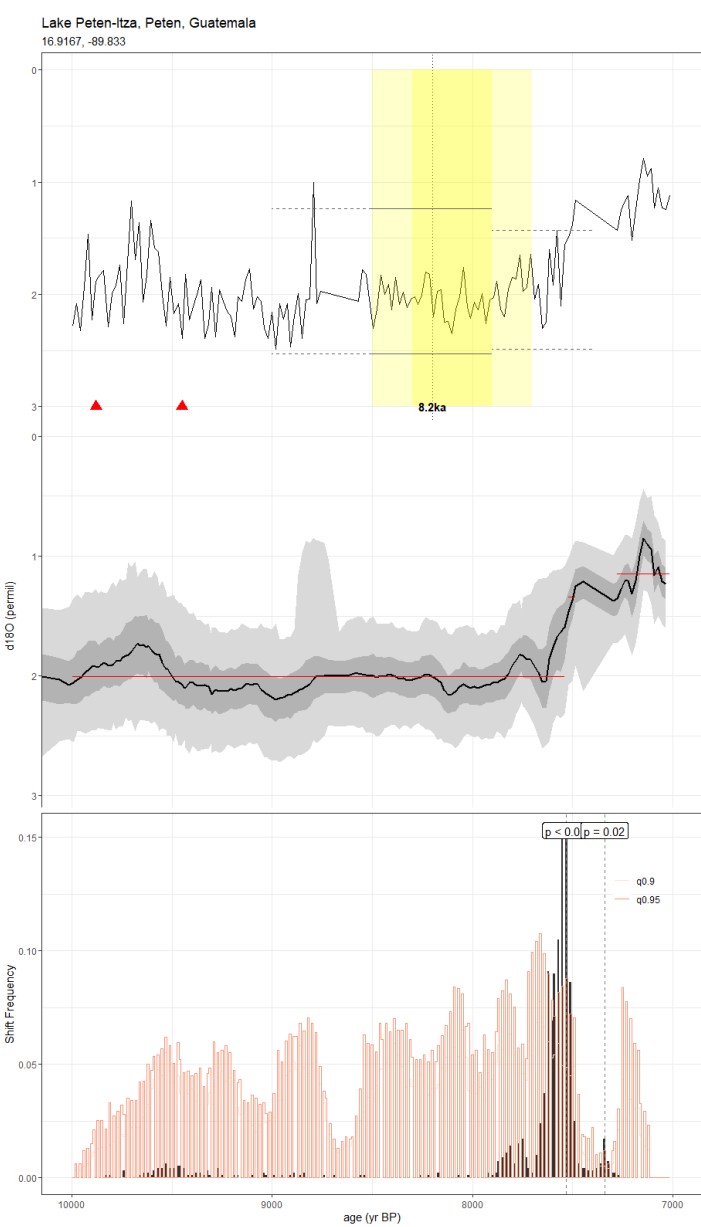

**Figure A13.** As in Fig. A3, but for the lacustrine gastropod $\delta^{18}$O record of Curtis et al., 1998 (Curtis6VII93). The published age model was constructed by linearly interpolating between 14C dates derived from terrestrial wood and charcoal samples. Here, we construct the age ensemble using the BACON algorithm included in geoChronR.



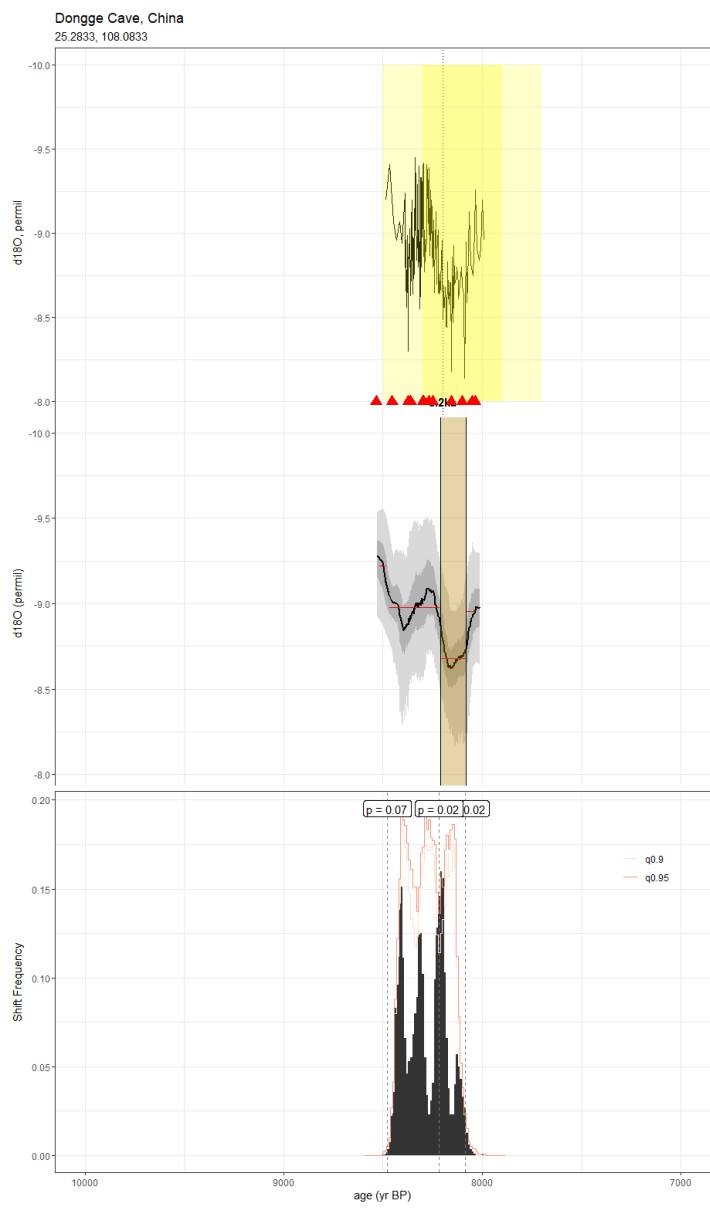

**Figure A14.** As in Fig. A3, but for the speleothem record of Cheng et al., 2009 (D4Cheng). The age model used in the original publication was unreported. Here, we use the BACON algorithm included in geoChronR to produce our age ensemble.



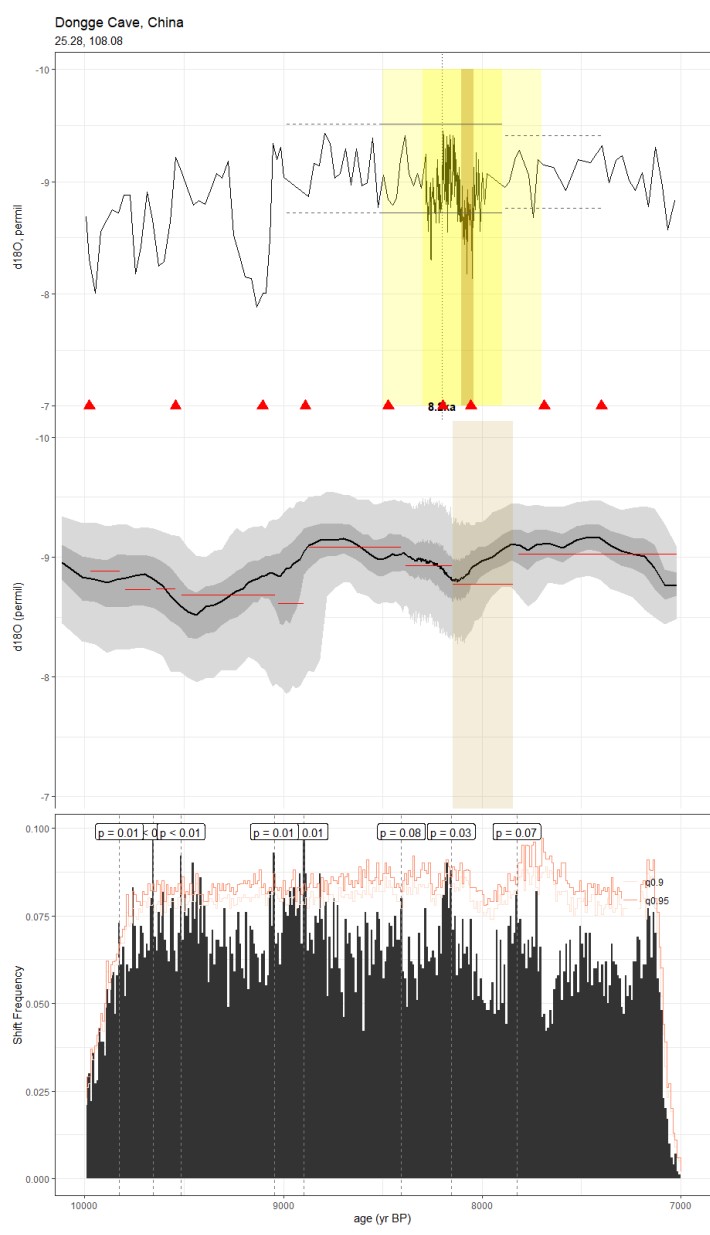

**Figure A15.** As in Fig. A3, but for the speleothem record of Dykoski et al., 2005 (D4Dykoski). The published age model was constructed by linearly interpolating between U/Th dates. Here, we reconstruct the age model using the BACON algorithm in geoChronR.



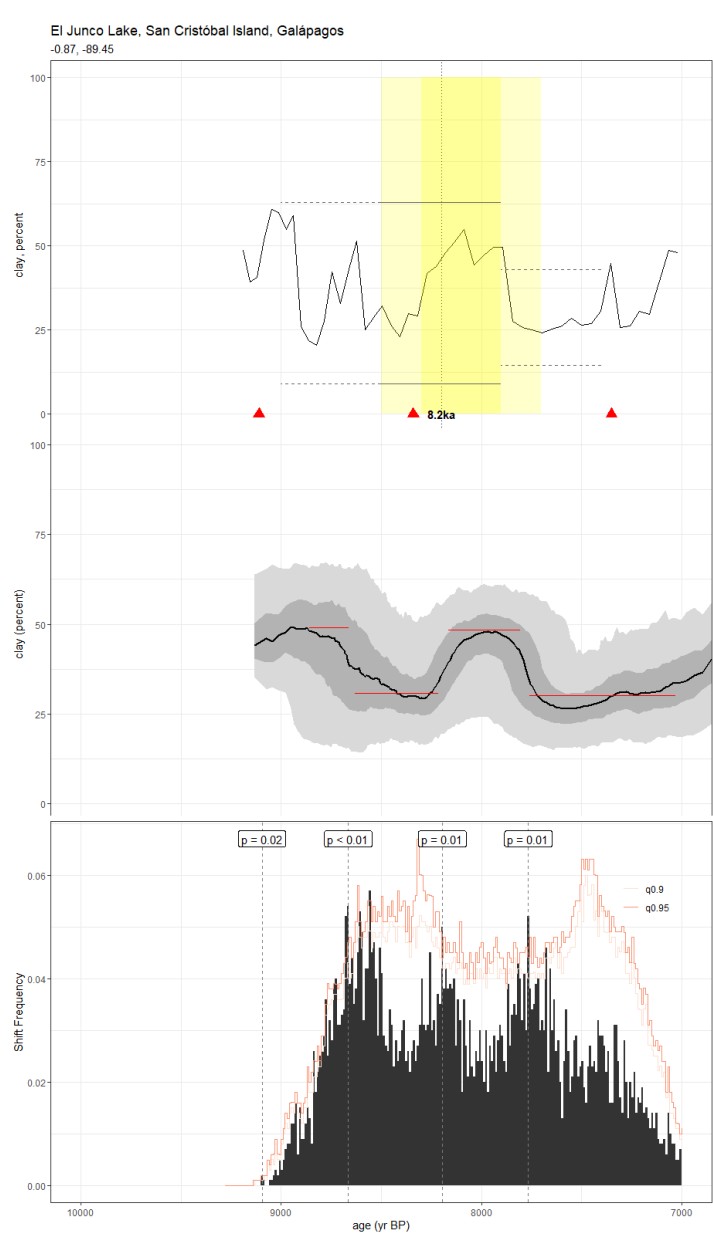

**Figure A16.** As in Fig. A3, but for the lacustrine clay content record of Conroy et al., 2008 (EJConroy). The published age model was constructed using CALIB 5.0 with the Southern Hemisphere dataset. The age ensemble presented here was created using the BACON algorithm with the SHCal20 calibration curve in geoChronR.



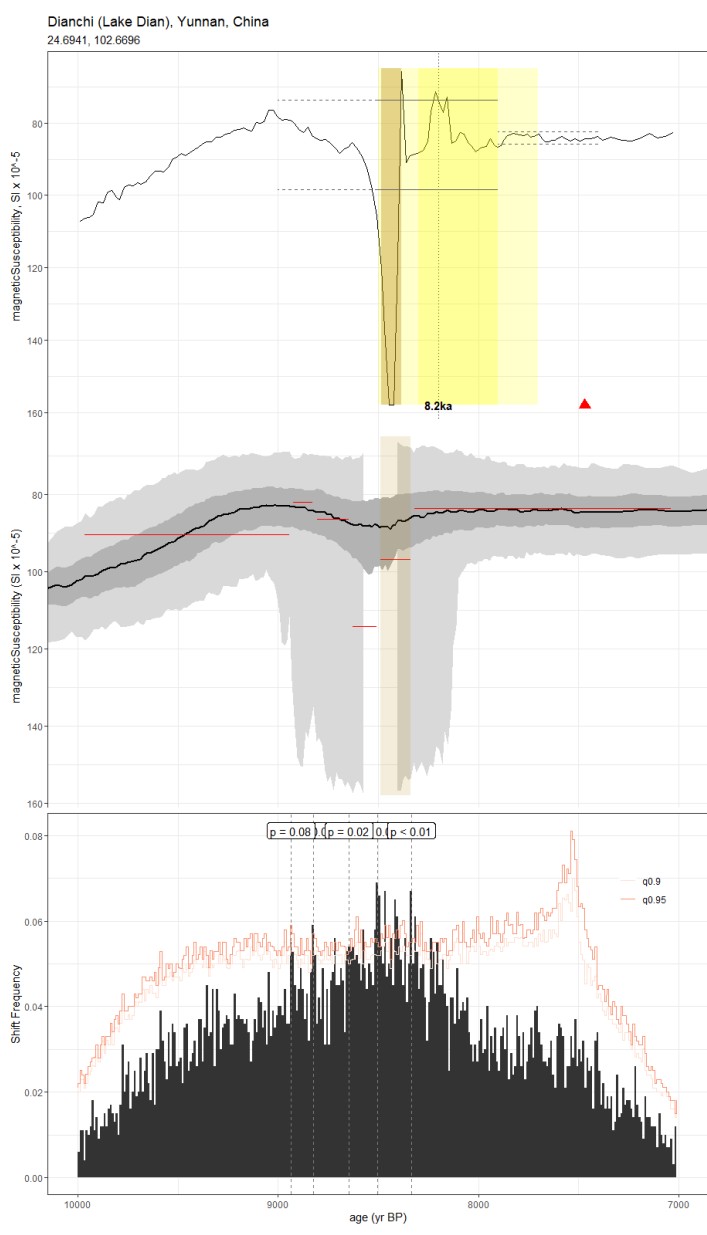

**Figure A17.** As in Fig. A3, but for the lake sediment magnetic susceptibility record of Hillman et al., 2021 (F14). The original age model was constructed using BACON with the IntCal20 calibration curve. Here, we have reconstructed it using the BACON algorithm in geoChronR.





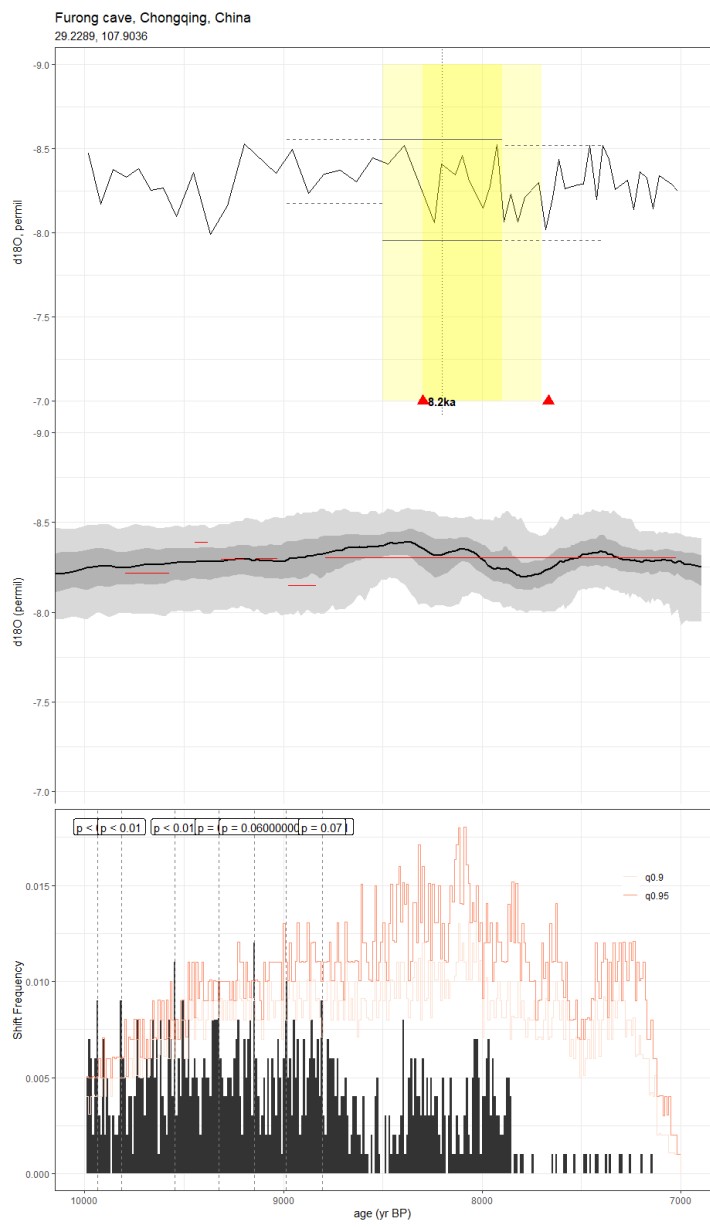

**Figure A18.** As in Fig. A3, but for the speleothem record of Li et al., 2011 (FR5). The age modeling algorithm used to construct the original age model was unreported, but leveraged the IntCal09 calibration curve. Here, we use the copRa age ensemble included in SISALv2.



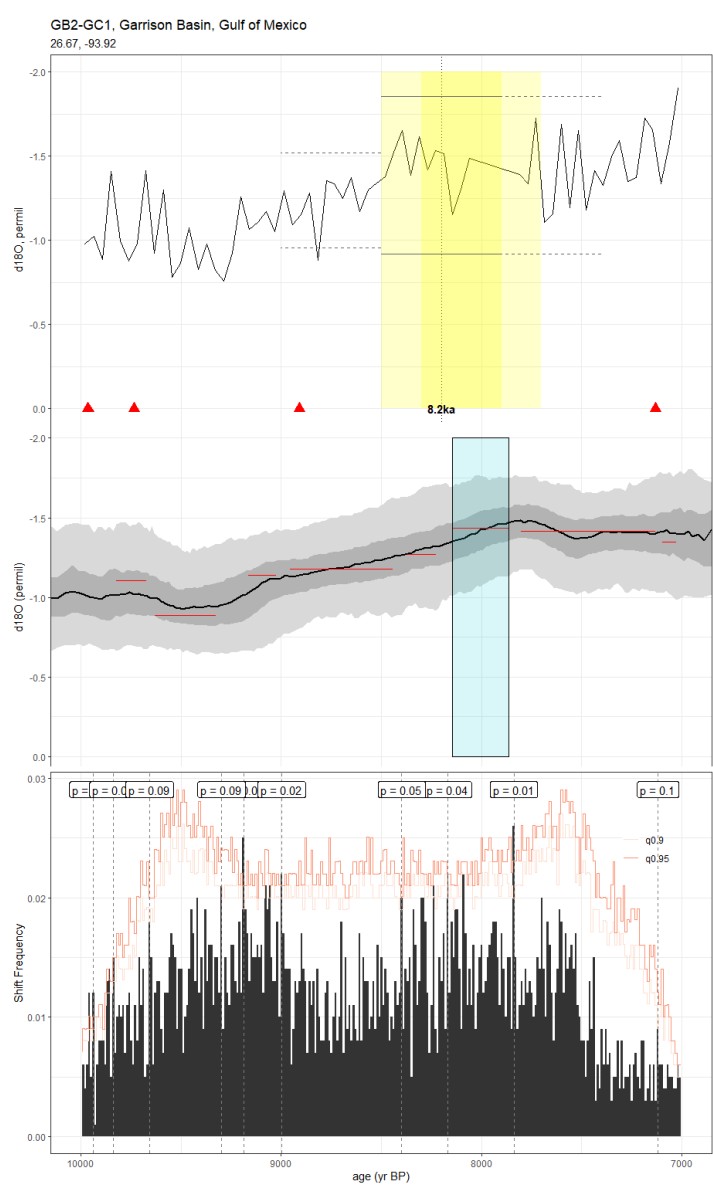

**Figure A19.** As in Fig. A3, but for the foraminifera record of Thirumulai et al., 2021 (GB2GC1). The published age model was developed using the BACON algorithm and Marine13 calibration curve, which we reconstructed using geoChronR.



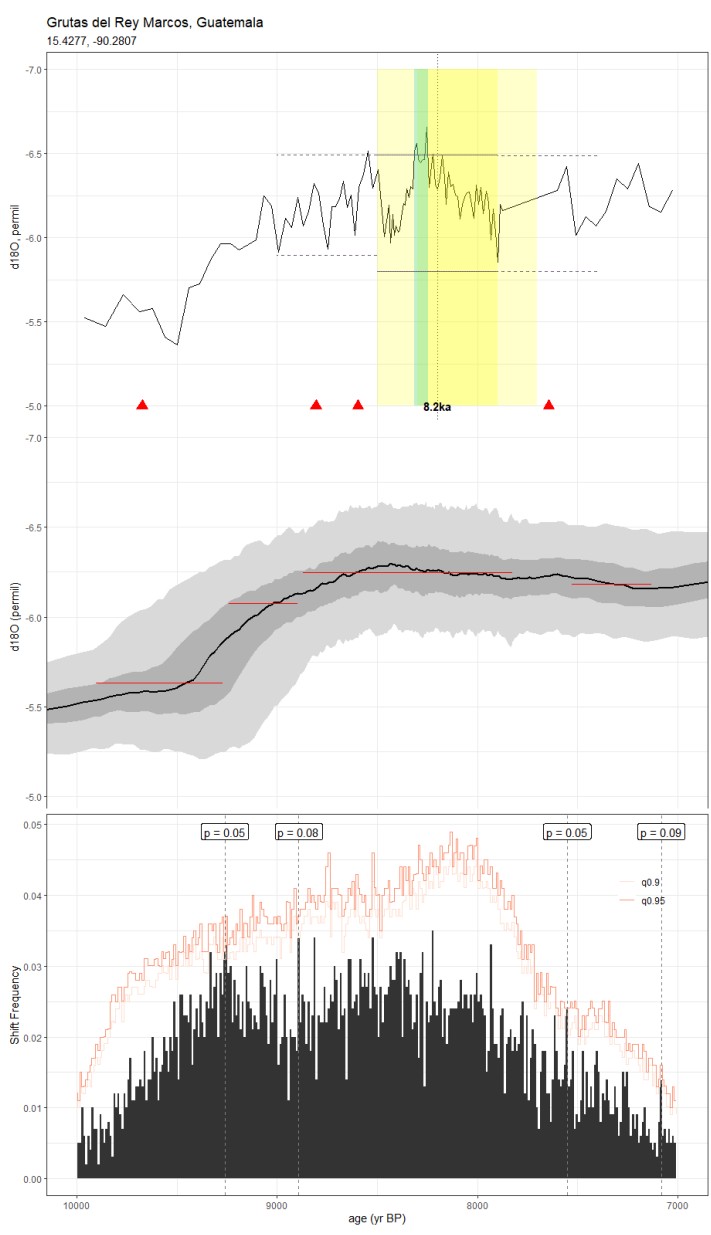

**Figure A20.** As in Fig. A3, but for the speleothem record of Winter et al., 2020 (GURM1). The published age model was constructed using the copRa algorithm, while we use the BACON ensemble produced for SISALv2 here.





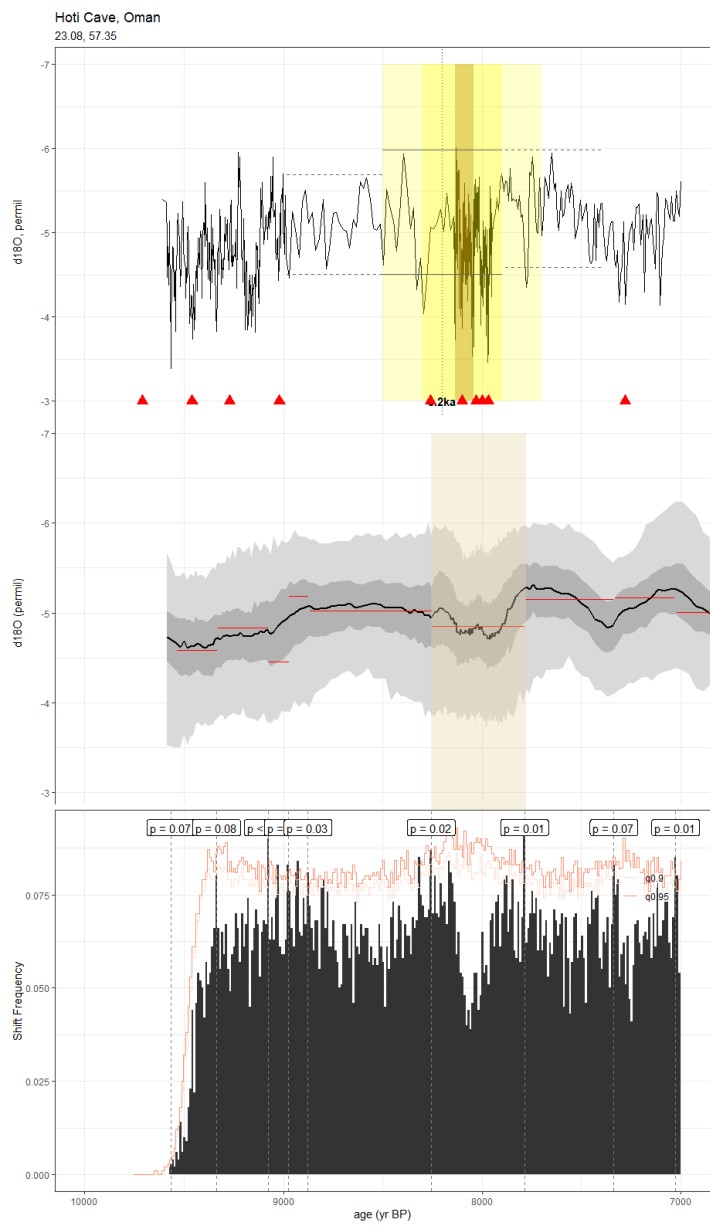

**Figure A21.** As in Fig. A3, but for the speleothem record of Neff et al., 2001 (H5). While the method used in the construction of the published time series was unreported, we leveraged the SISALv2 BACON ensemble for our analyses.





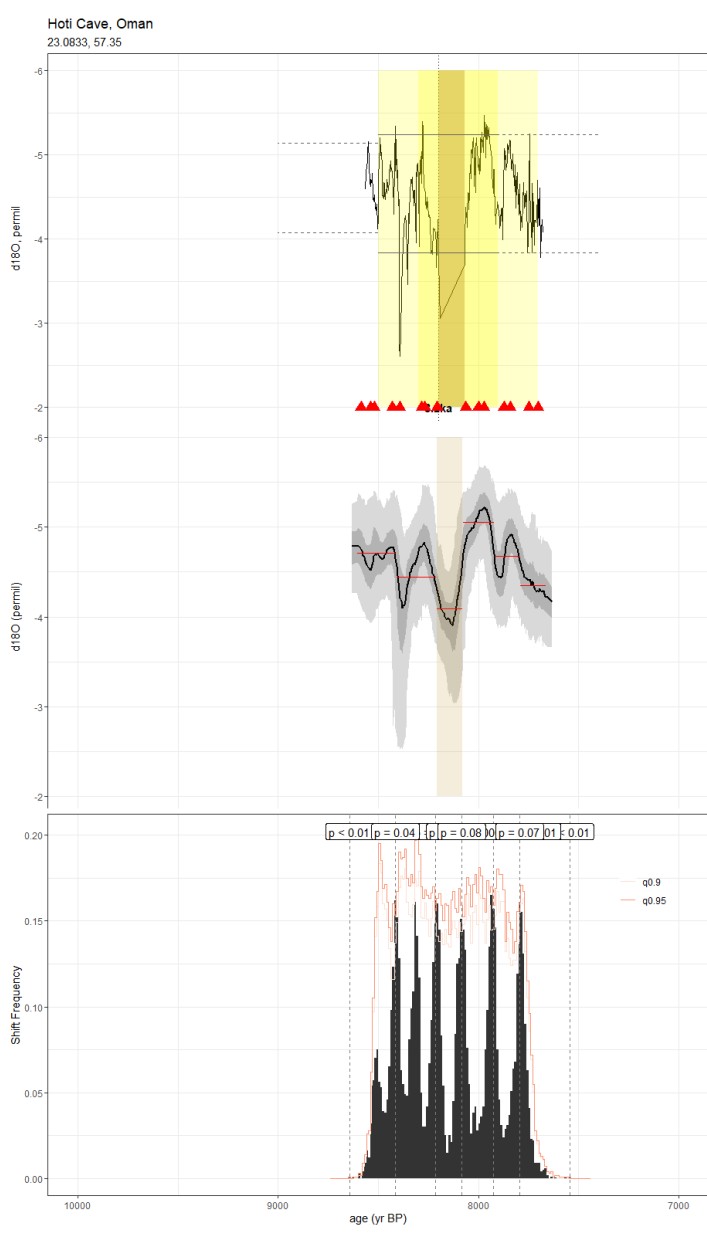

**Figure A22.** As in Fig. A3, but for the speleothem record of Cheng et al., 2009 (H14). The age modeling algorithm used to construct the original age model was unreported. Here, we constructed our age ensemble using BACON in geoChronR.




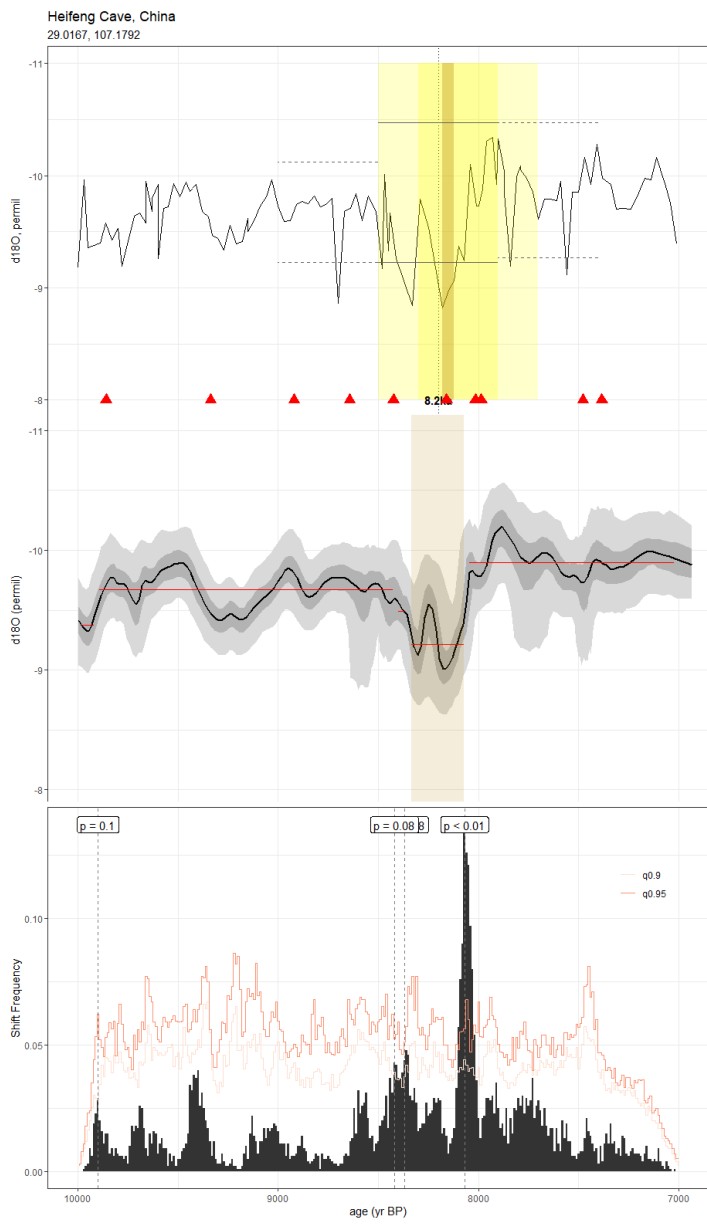

**Figure A23.** As in Fig. A3, but for the speleothem record of Yang et al., 2019 (HF01). The published age model was constructed via polynomial regression between radiometric dates. For our analyses, we leveraged the copRa age ensemble included in version 2 of the SISAL database.



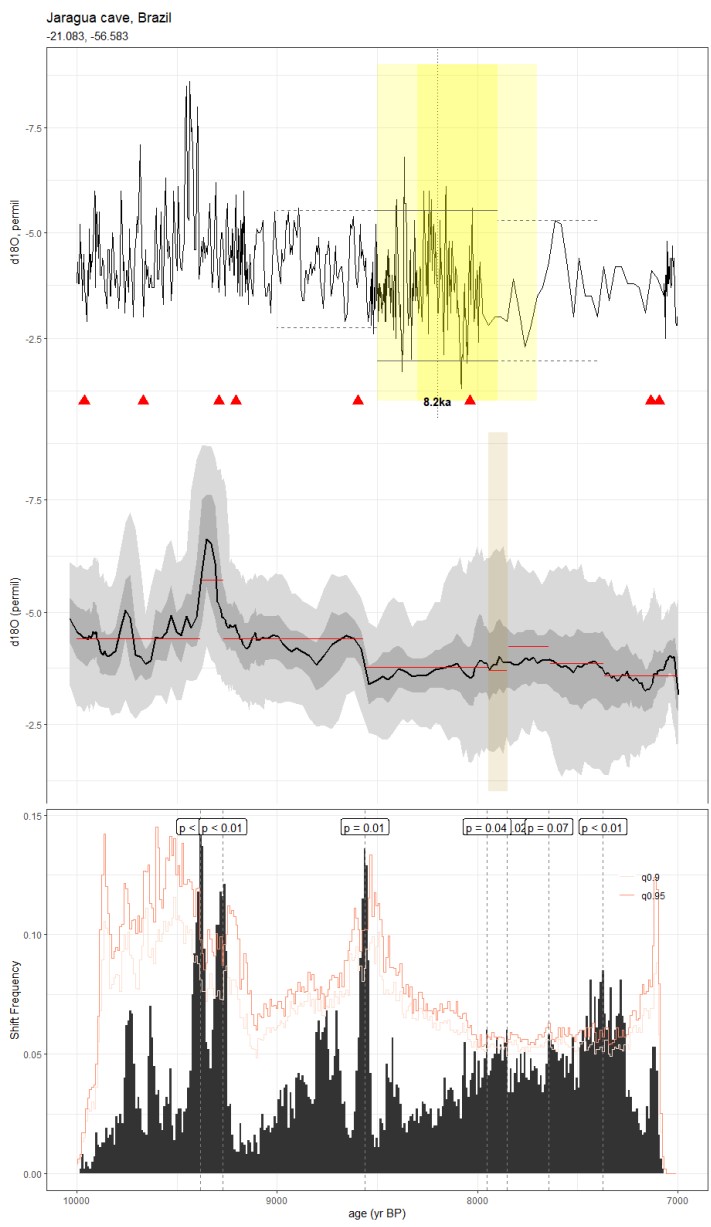

**Figure A24.** As in Fig. A3, but for the speleothem record of Novello et al., 2017 (JAR7). The published age model was constructed via linear interpolation between radiometric dates. For our analyses, we leveraged the BACON age ensemble produced for SISALv2.




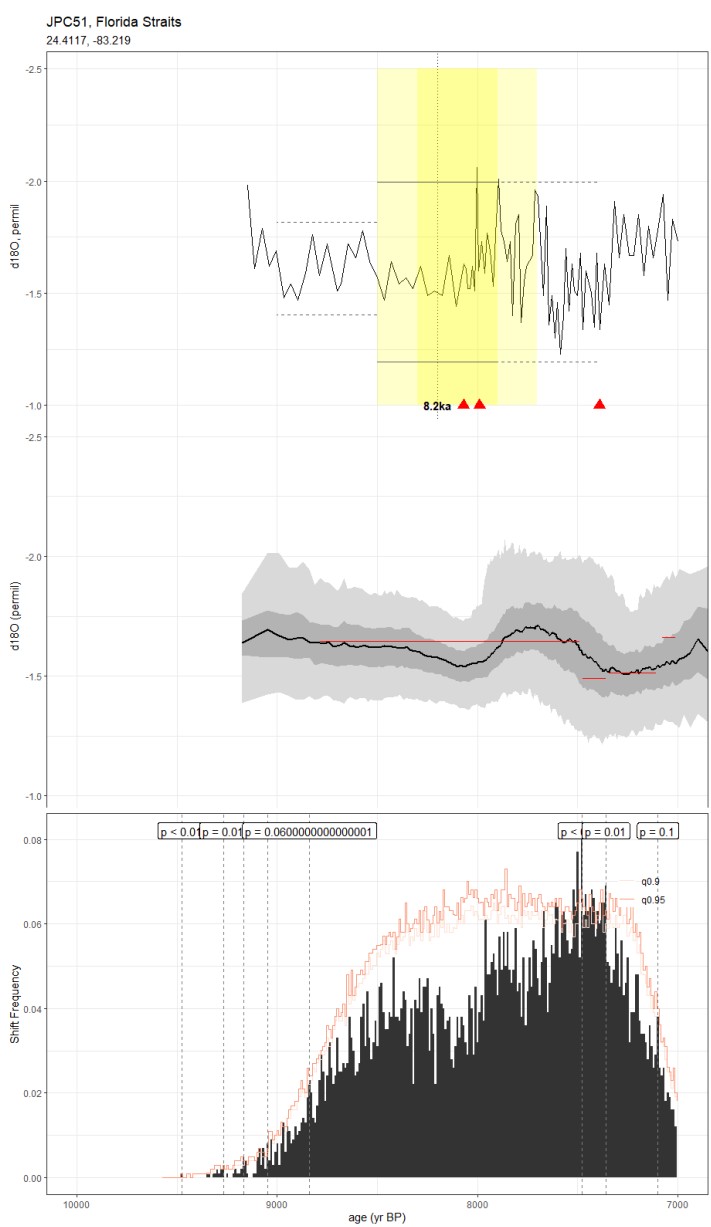

**Figure A25.** As in Fig. A3, but for the foraminifera record of Schmidt et al., 2012 (JPC51). The published age model was created using CALIB 6.0, with a standard -400 year reservoir age correction for surface waters. Here, we use the BACON algorithm included in geoChronR to produce our age ensemble.




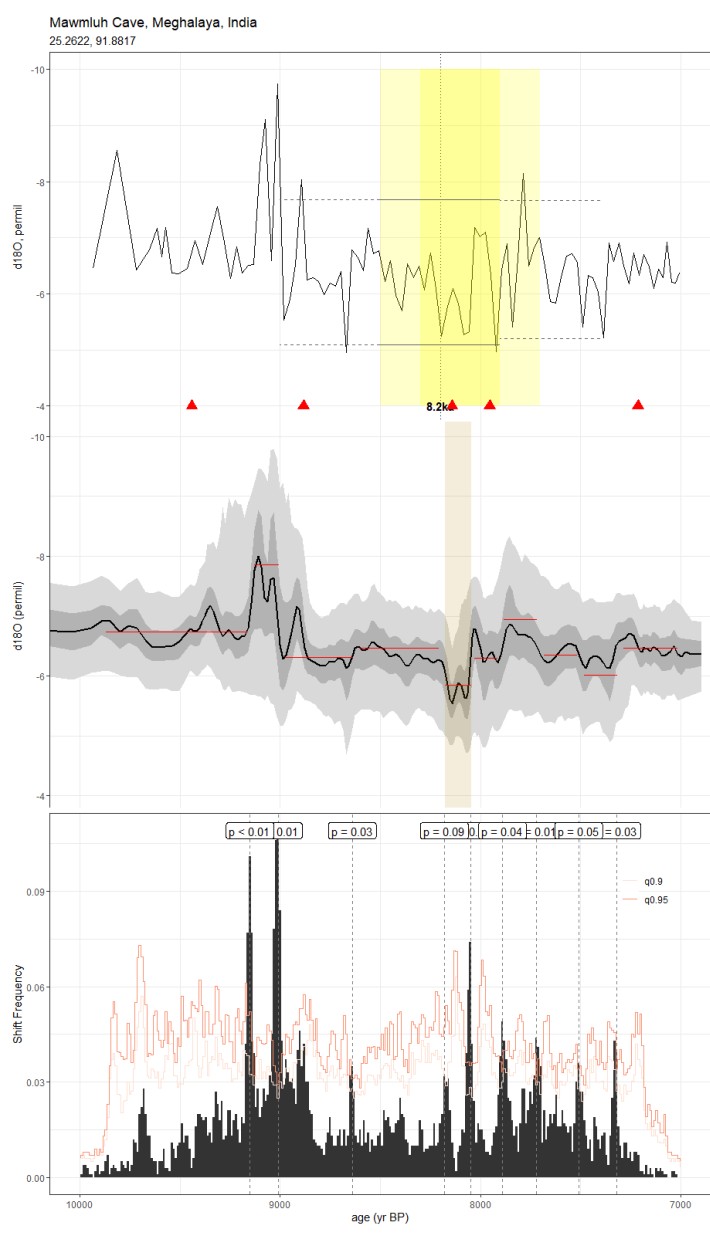

**Figure A26.** As in Fig. A3, but for the speleothem record of Huguet et al., 2018 (KM1). While the published age model was constructed using the StalAge algorithm, we leverage the Bchron age ensemble included in SISALv2 here.





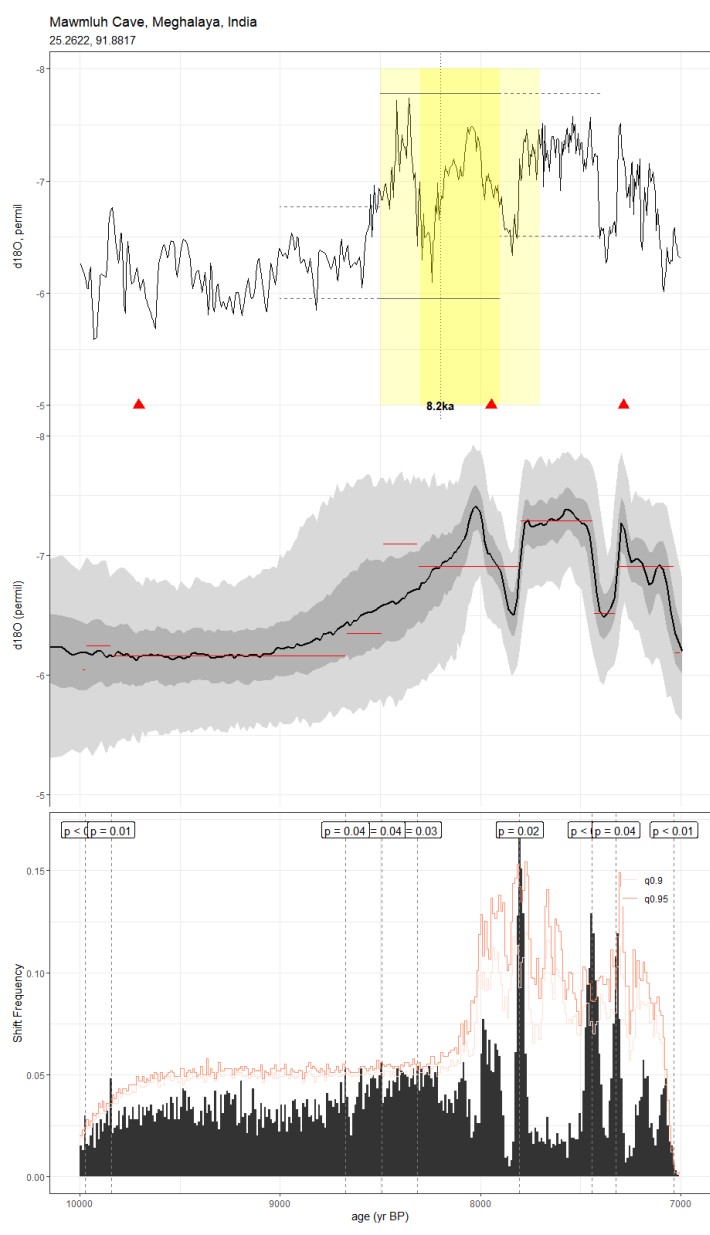

**Figure A27.** As in Fig. A3, but for the speleothem record of Berkelhammer et al., 2012 (KMA). The published age model was created using the StalAge algorithm. Here, we used the BACON age ensemble included in SISALv2.

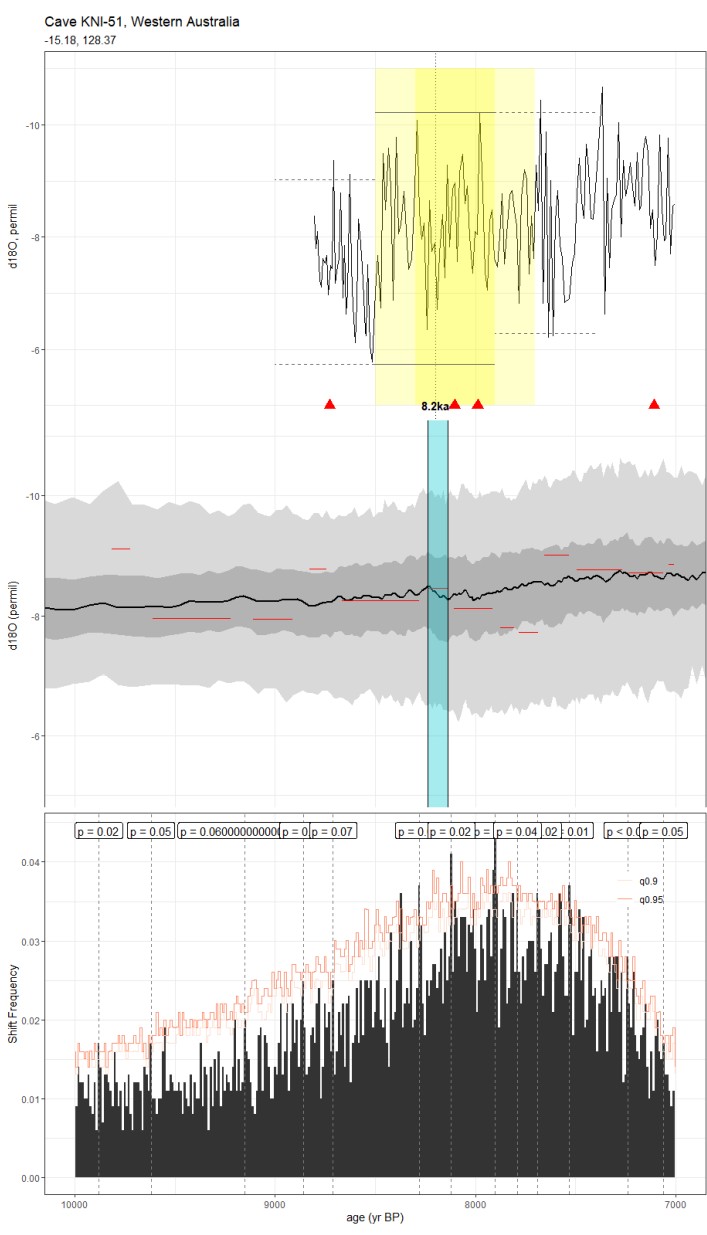

**Figure A28.** As in Fig. A3, but for the speleothem record of Denniston et al., 2013 (KN51). The method used in the construction of the published age model is unknown, but we use the copRa ensemble generated for SISALv2 here.



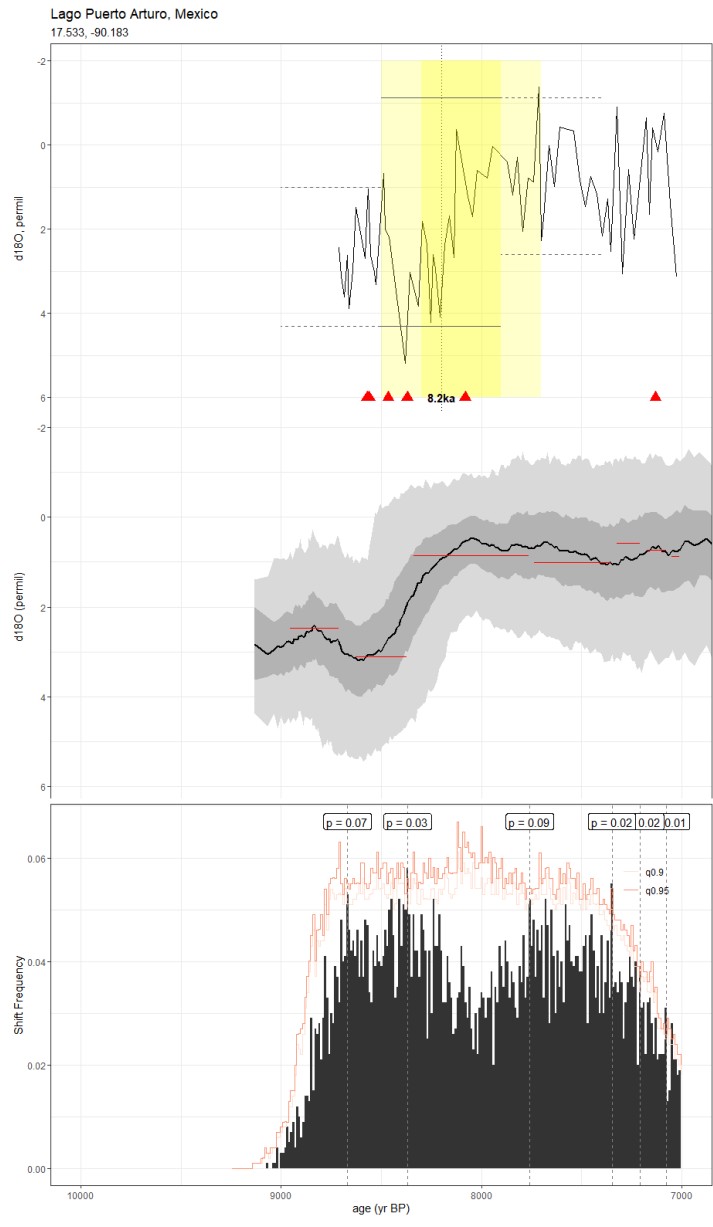

**Figure A29.** As in Fig. A3, but for the lake sediment $\delta^{18}$O record of Wahl et al., 2014 (LagoPuertoArturo). The published age model was constructed using CLAM 2.2 and the IntCal13 calibration curve. For our analyses, we reconstructed the age model using BACON and IntCal20 in geoChronR.

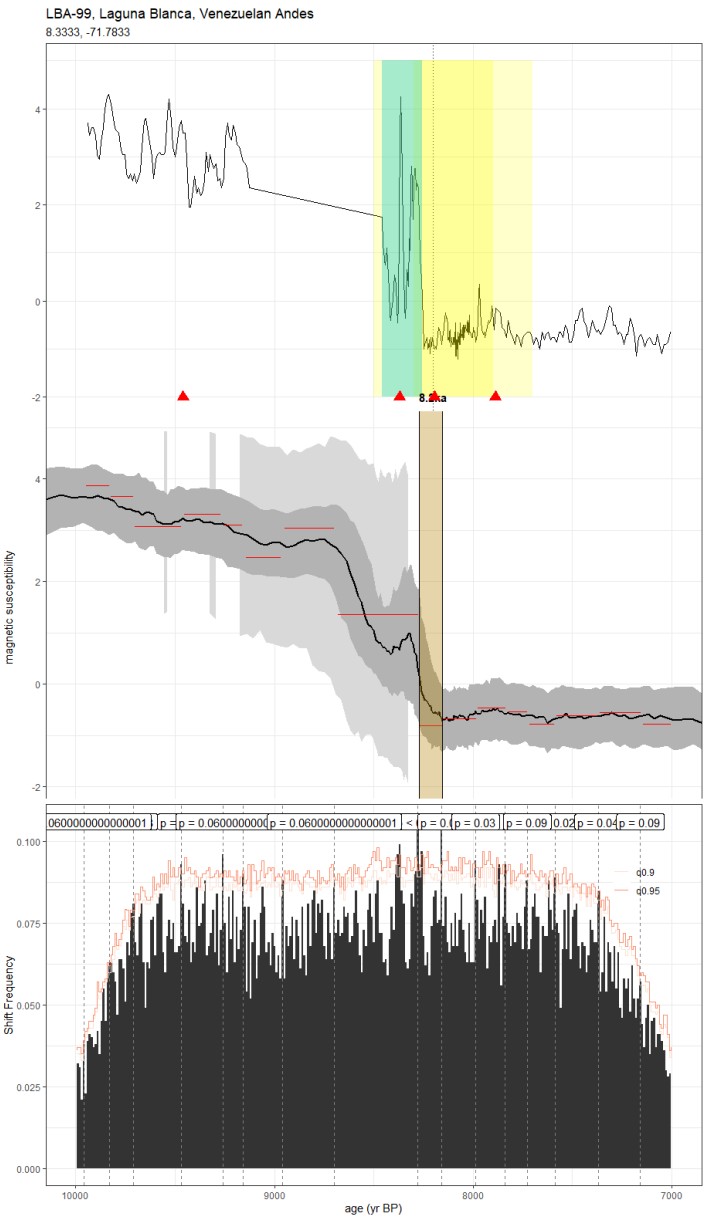

**Figure A30.** As in Fig. A3, but for the lake sediment magnetic susceptibility record of Pollisar et al., 2013 (LBA99). The published age model was constructed by linearly interpolating between radiometric dates with the IntCal04 calibration curve. Here, we constructed our ensemble using the BACON algorithm and IntCal20 curve included in geoChronR.





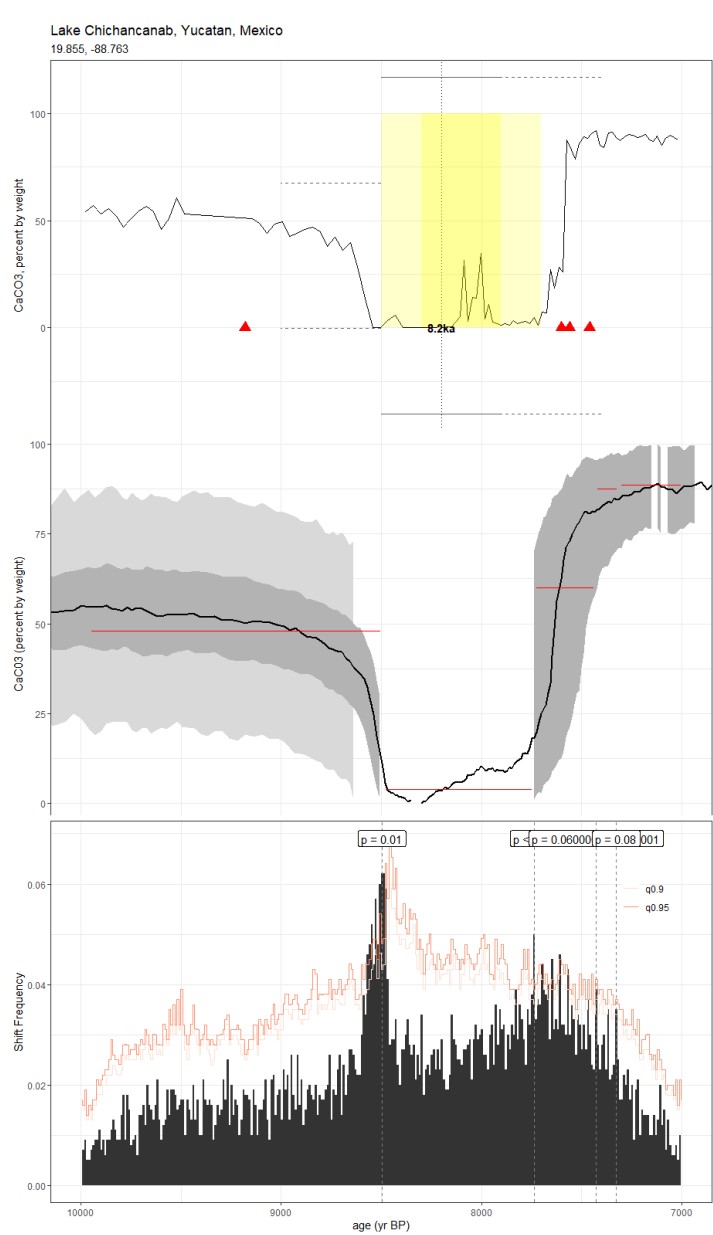

**Figure A31.** As in Fig. A3, but for the lake sediment record of Hodell et al., 1995 (LC1). The published age model was created using the decadal tree ring dataset in CALIB. Here, we use BACON with the IntCal20 calibration curve supplied by geoChronR.



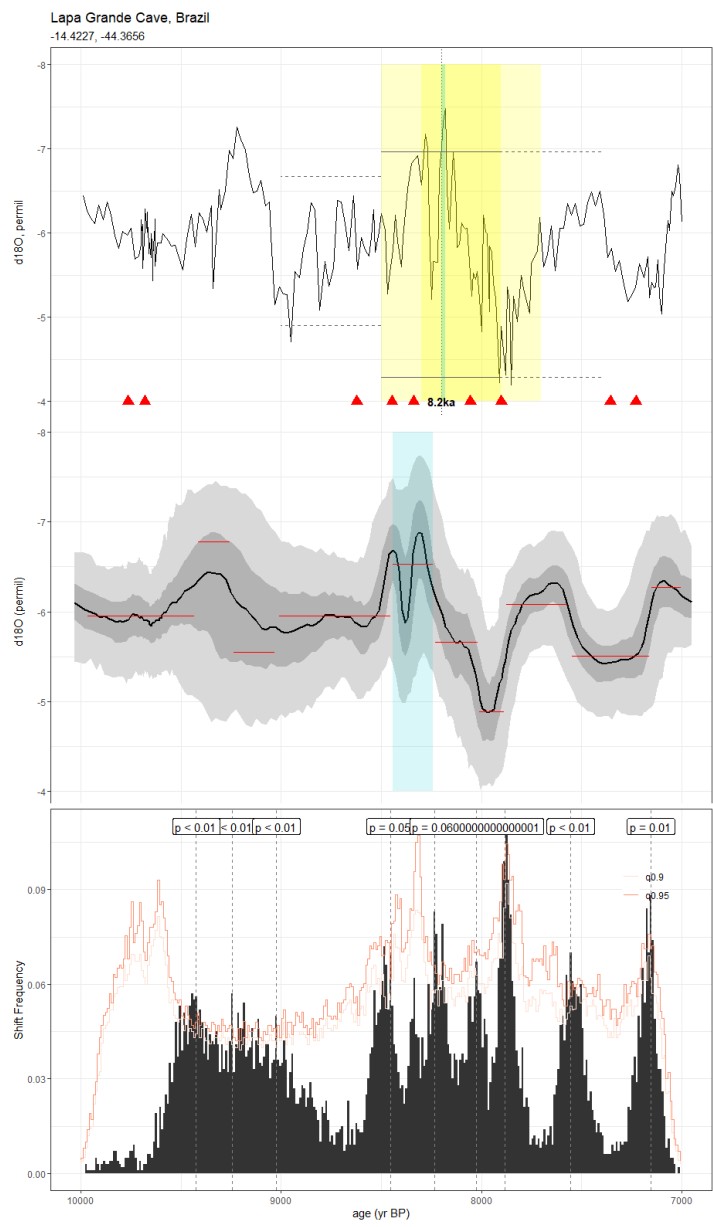

**Figure A32.** As in Fig. A3, but for the speleothem record of Strikis et al., 2011 (LG11). The original method used in construction of the published age model was unreported, but we leverage the BACON age ensemble published in SISALv2 for our analyses.




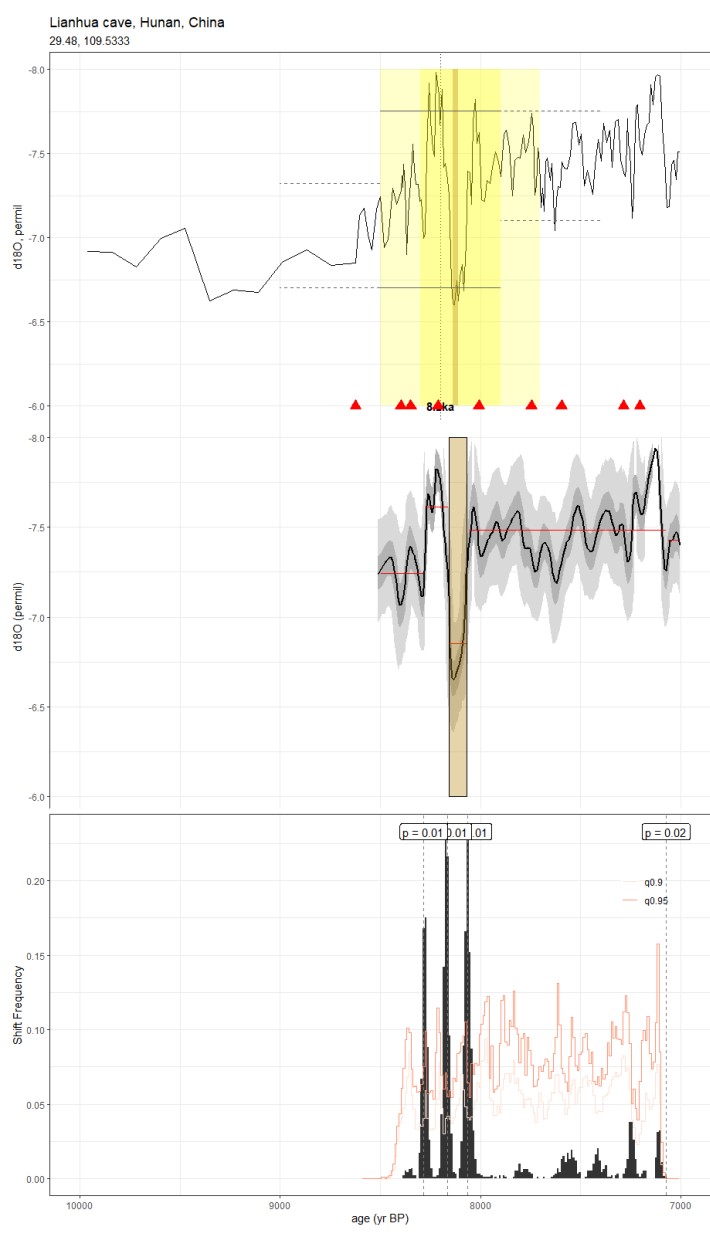

**Figure A33.** As in Fig. A3, but for the speleothem record of Zhang et al., 2013 (LH2). The published age model was generated by linearly interpolating between radiometric dates. Here, we employ the BACON age ensemble included in version 2 of the SISAL database.



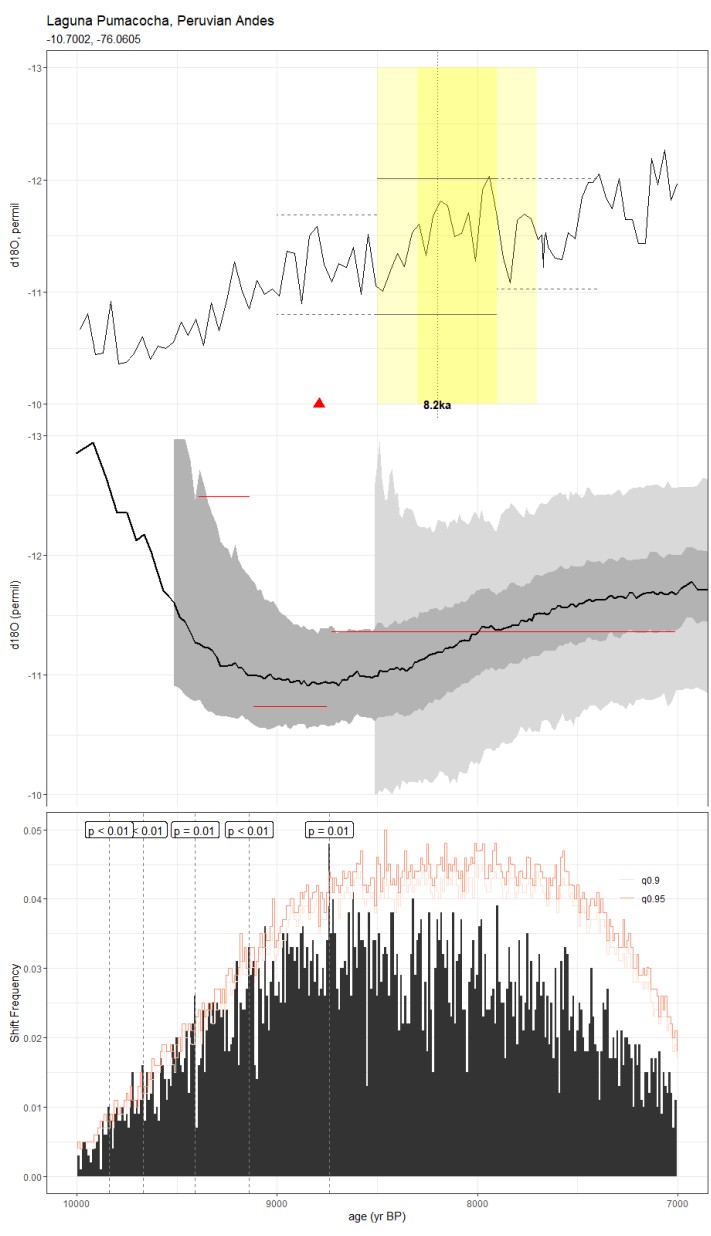

**Figure A34.** As in Fig. A3, but for the lacustrine sediment record of Bird et al., 2011 (LP). The published age model was created using CALIB 5.0 with an unreported calibration curve. Here, we construct our age ensemble in geoChronR using the BACON algorithm and SHCal20.

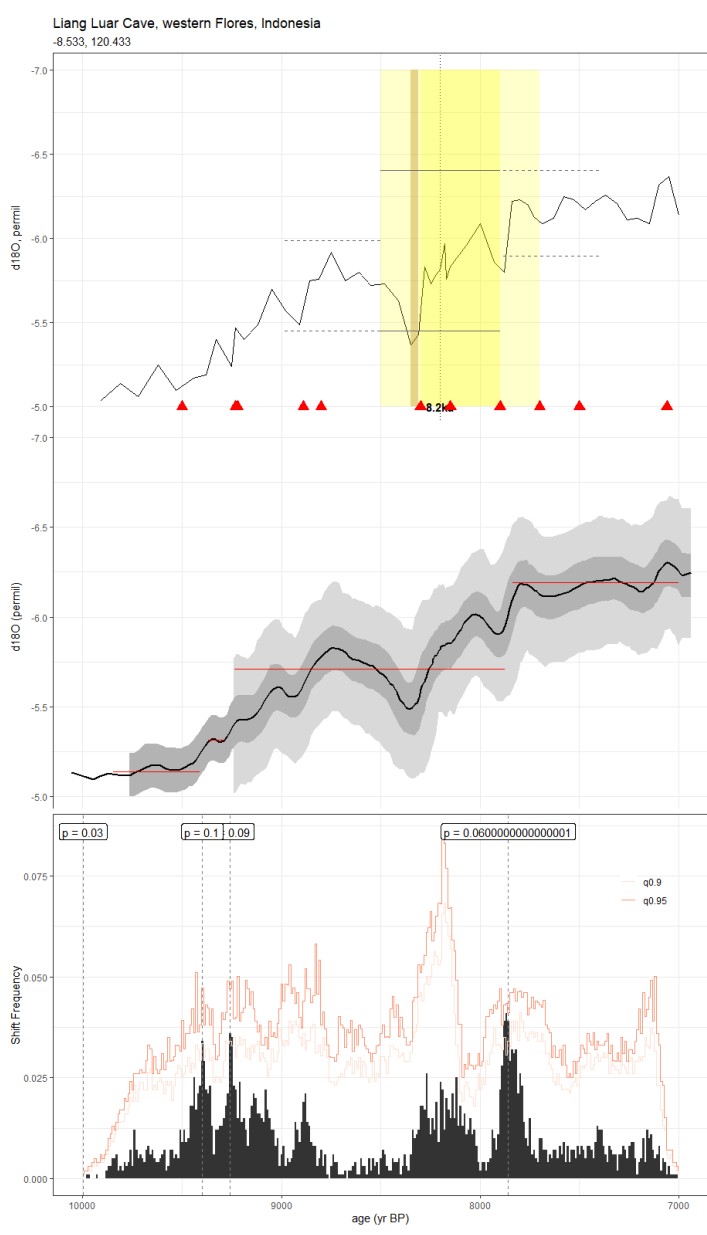

**Figure A35.** As in Fig. A3, but for the speleothem record of Ayliffe et al., 2013 (LR06_B3_2013). The published age model was constructed by linearly interpolating between radiometric dates. For our analyses, we leveraged the Bchron ensemble published in the SISALv2 dataset.





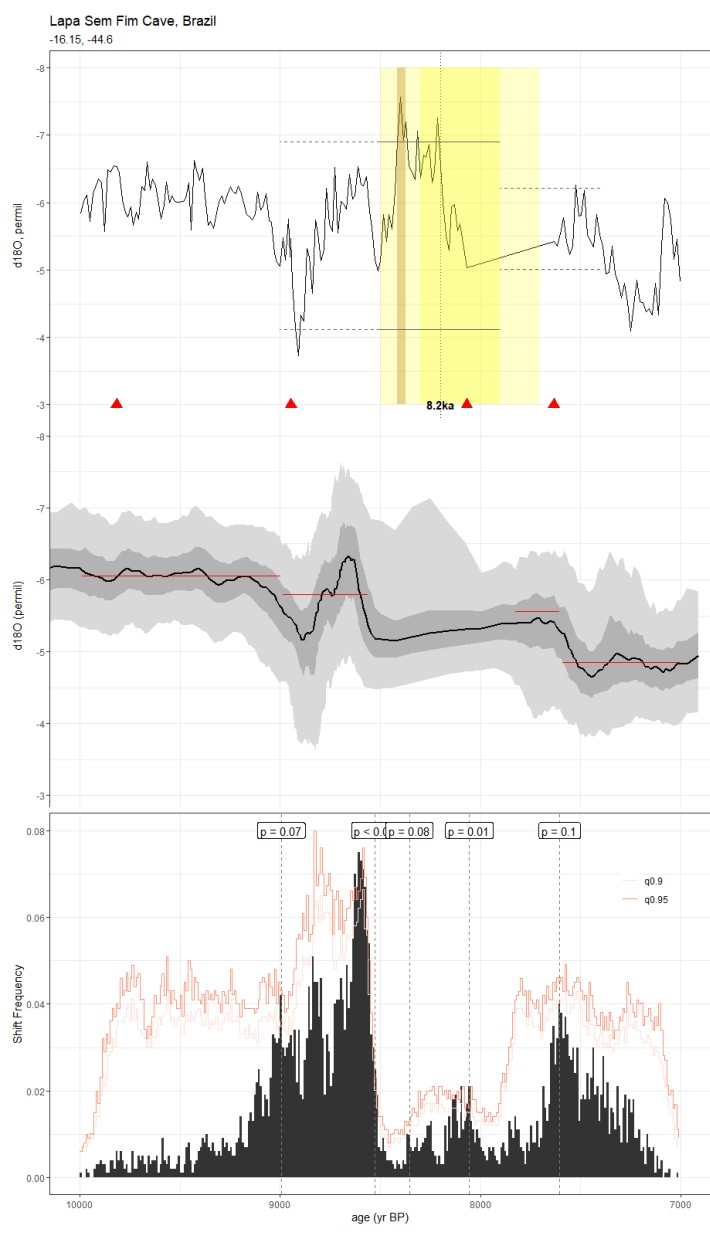

**Figure A36.** As in Fig. A3, but for the speleothem record of Azevedo et al., 2021 (LSF19). The original method used in the construction of the published age model was unreported, but we use the BACON ensemble supplied in version 2 of the SISAL database for our analyses.



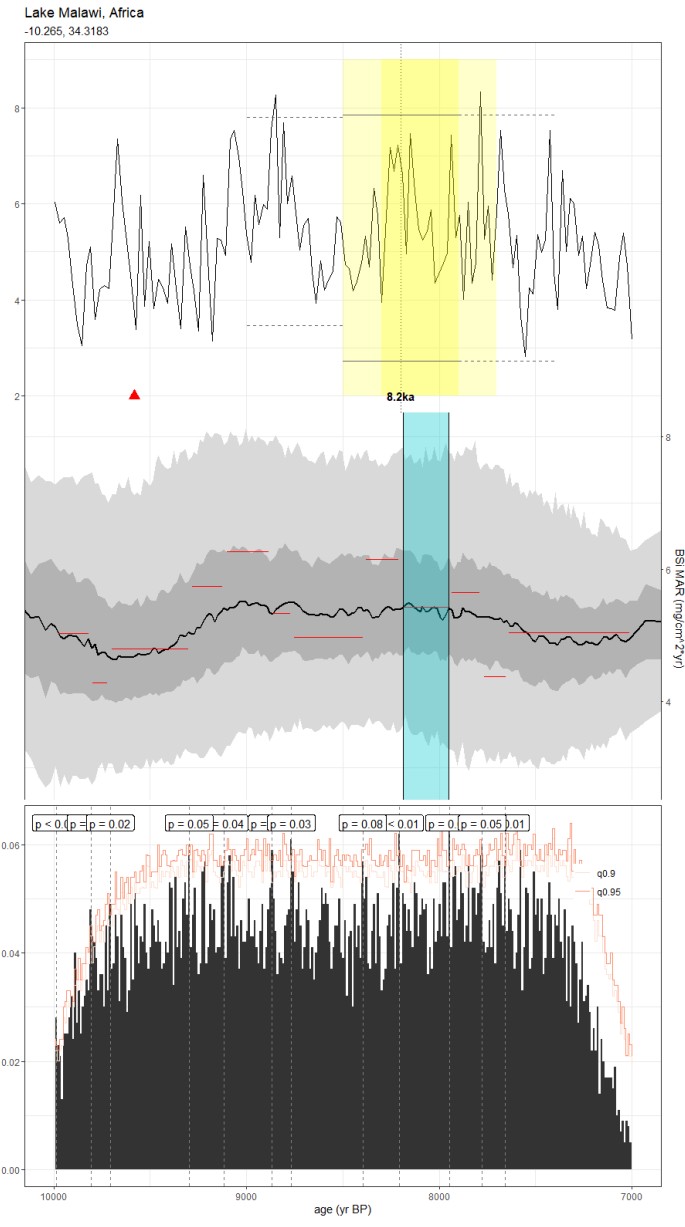

**Figure A37.** As in Fig. A3, but for the lake sediment BSi MAR record of Johnson et al., 2003 (M981P). CALIB 4.3 was used in the construction of the published age model, with a reservoir age correction of -450 years applied to the radiometric dates. Here, we constructed the age ensemble using the BACON algorithm in geoChronR.



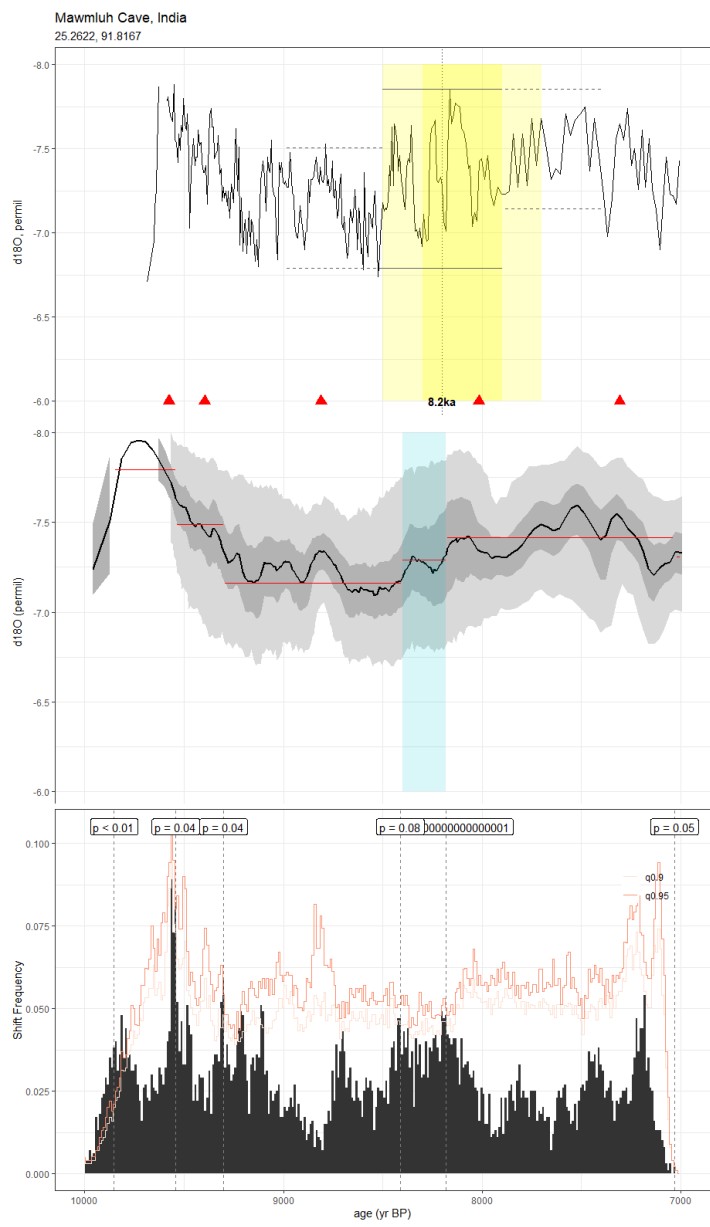

**Figure A38.** As in Fig. A3, but for the speleothem record of Lechleitner et al., 2017 (MAW6). The published age model was constructed using copRa. Here, we employed the Bchron ensemble included in SISALv2.



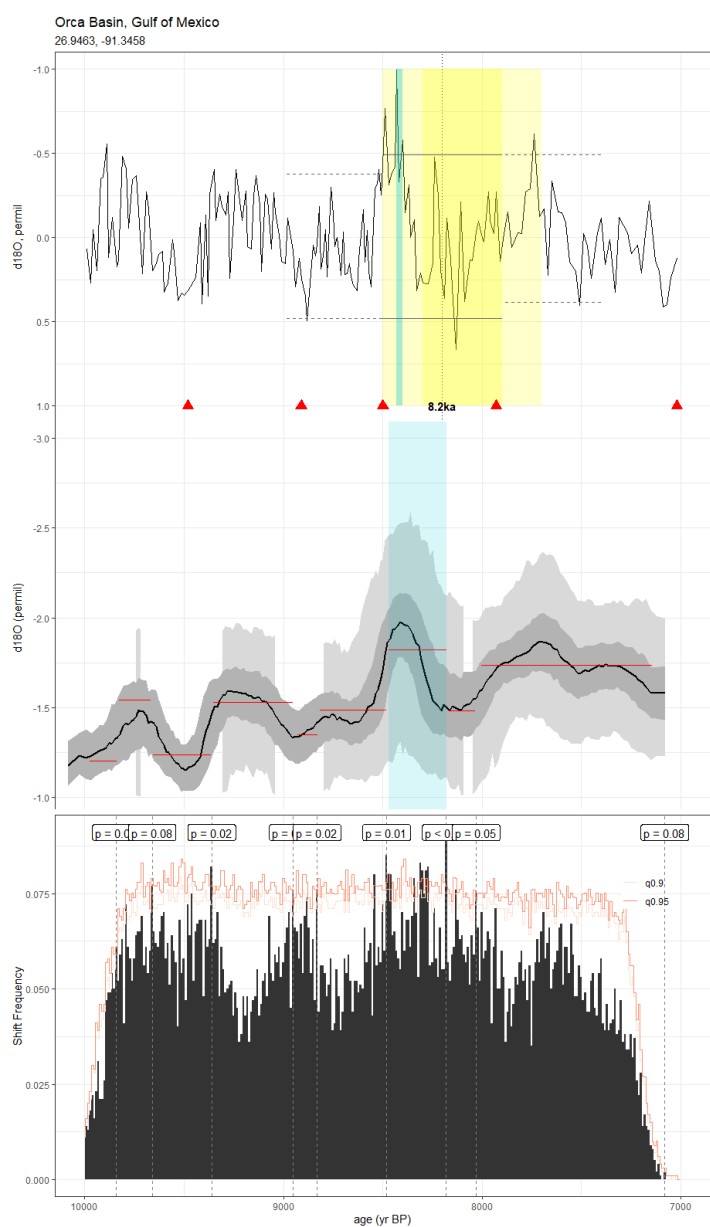

**Figure A39.** As in Fig. A3, but for the foraminifera record of LoDico et al., 2006 (MD022550). The published age model was constructed using CALIB 5.0, with a 400 year reservoir age correction applied. Here, we used the BACON algorithm included in geoChronR to create the age ensemble.




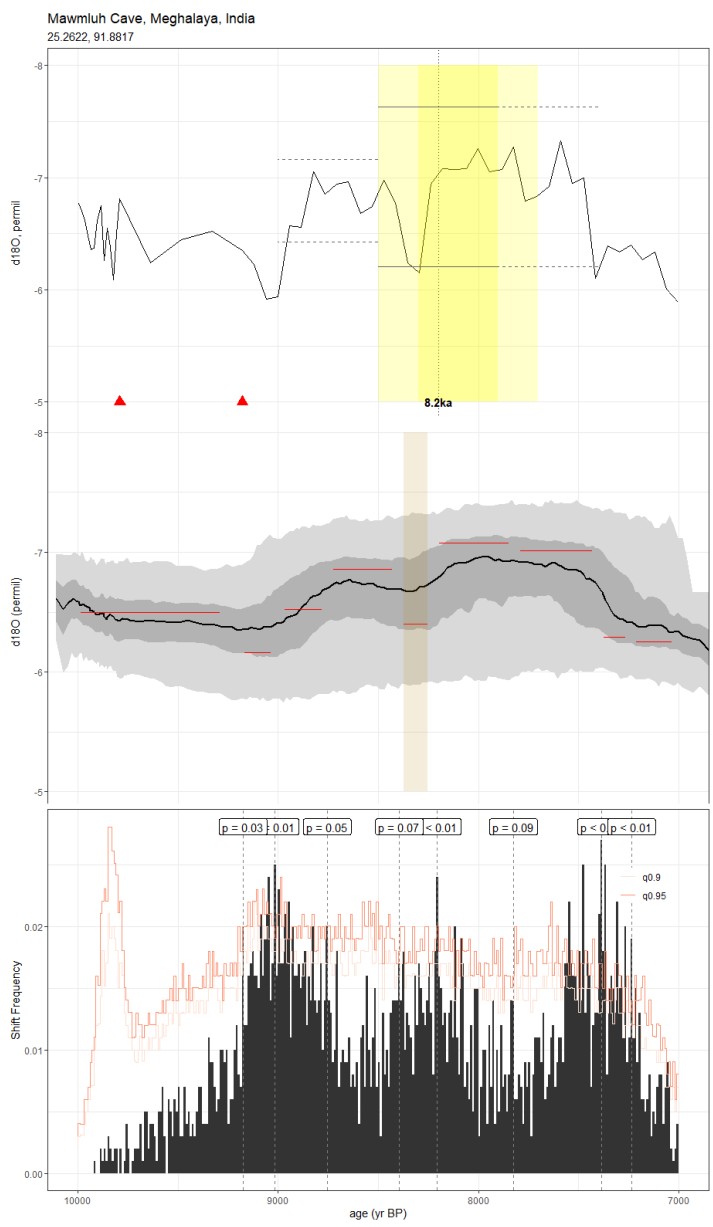

**Figure A40.** As in Fig. A3, but for the speleothem record of Dutt et al., 2015 (MWS1). The published age model was created using the StalAge algorithm. Here, we use the Bchron ensemble from SISALv2.



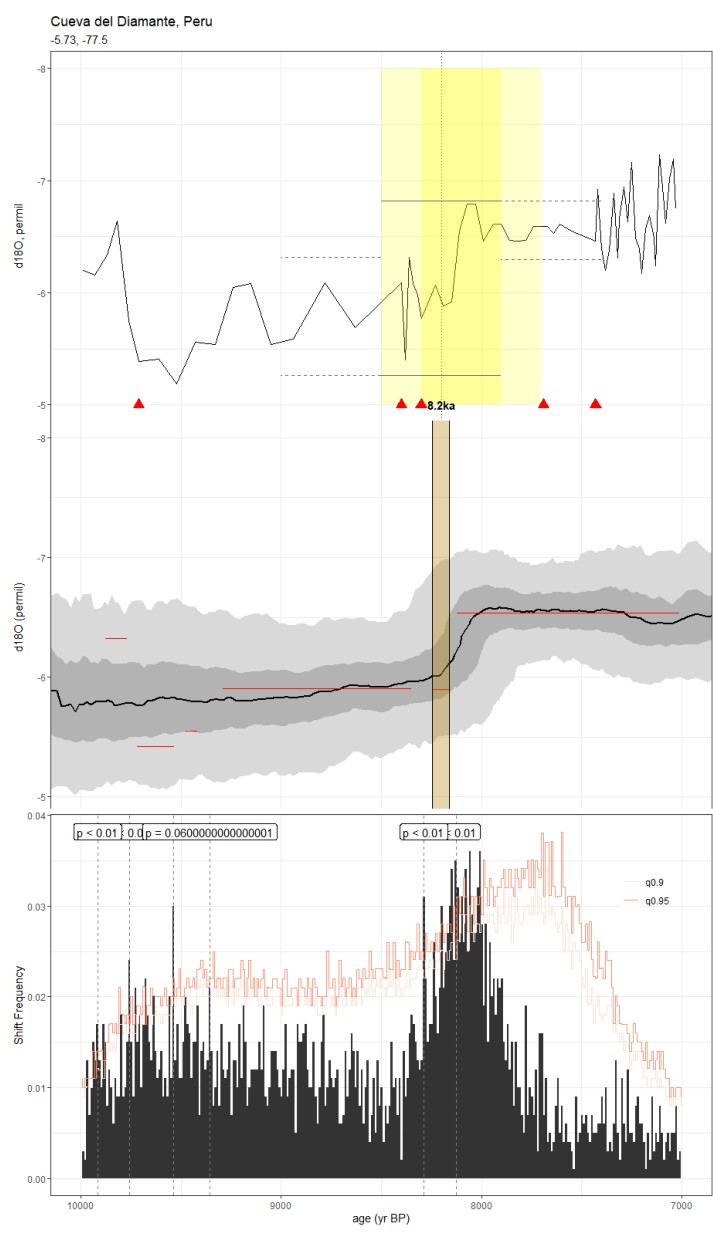

**Figure A41.** As in Fig. A3, but for the speleothem record of Cheng et al., 2013 (NARC). The published age model was constructed by linearly interpolating between radiometric dates, but here, we leverage the copRa ensemble from the SISALv2 dataset.



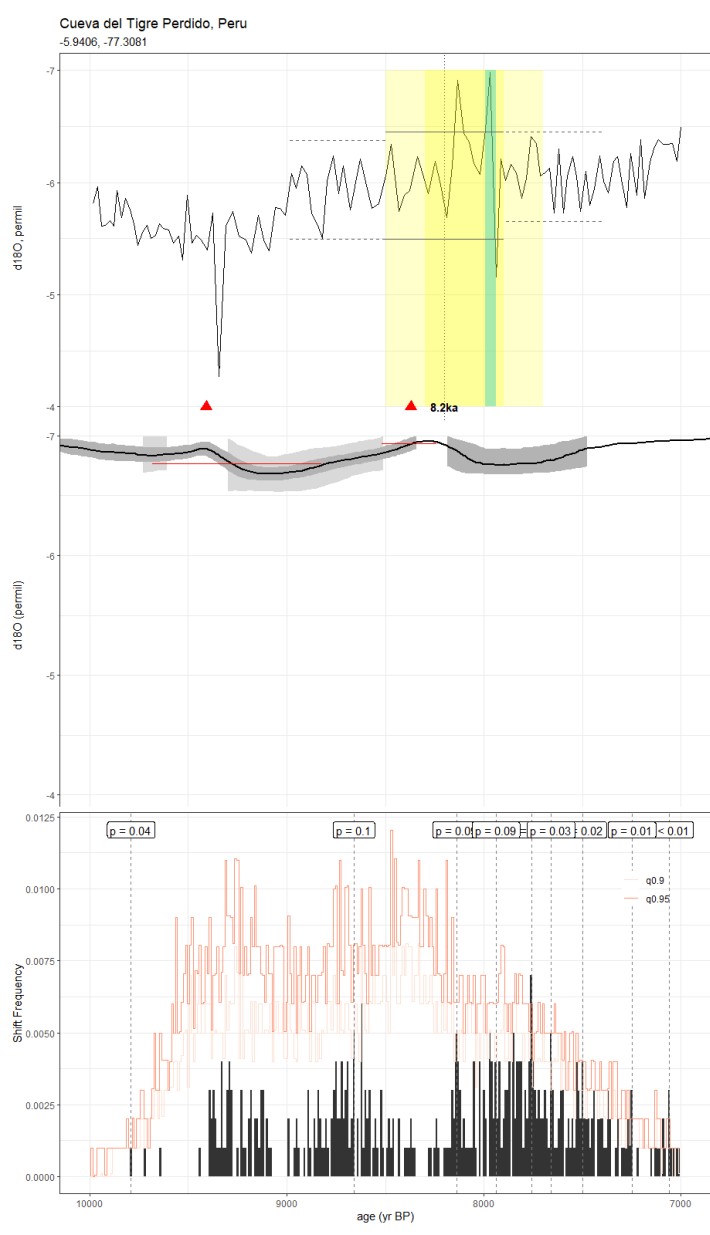

**Figure A42.** As in Fig. A3, but for the speleothem record of van Breukelen et al., 2008 (NCB). Isoplot 3 was used to construct the published age model, however, we use the SISALv2 BACON age ensemble for our analyses.




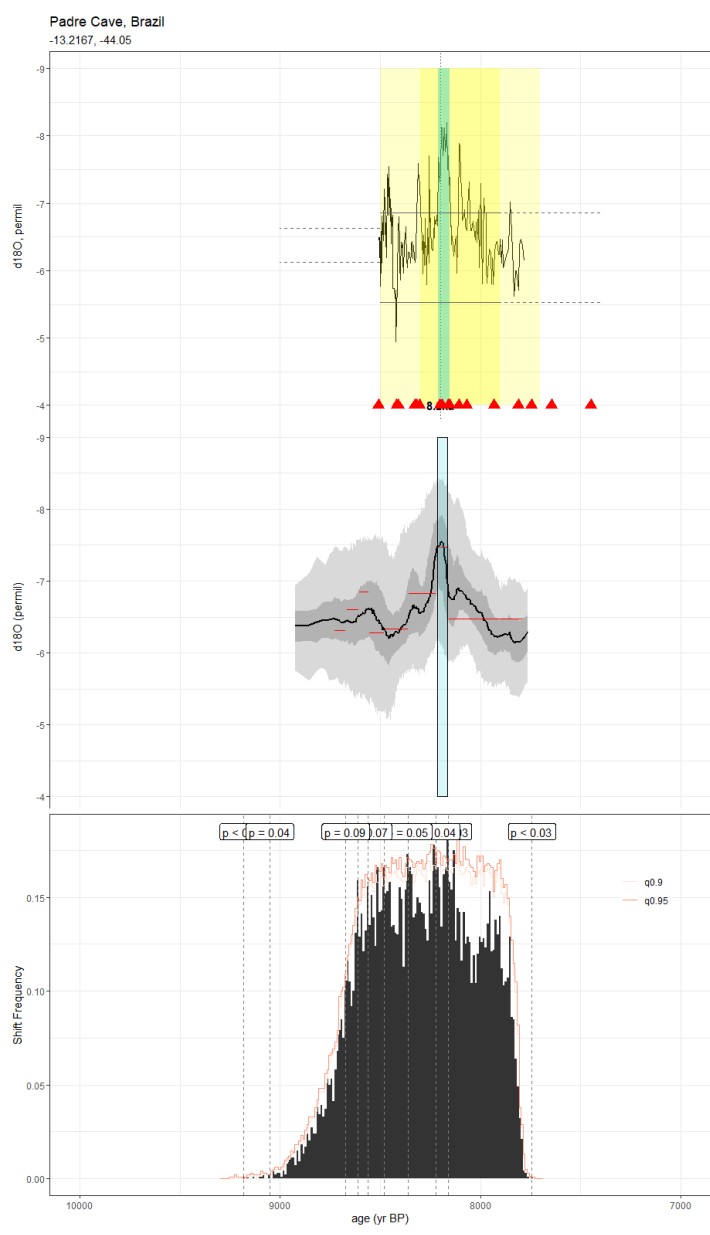

**Figure A43.** As in Fig. A3, but for the speleothem record of Cheng et al., 2009 (PAD07). The original age modeling method used in the construction of the published time series is unknown. Here, we present an age ensemble using the BACON algorithm provided by geoChronR.




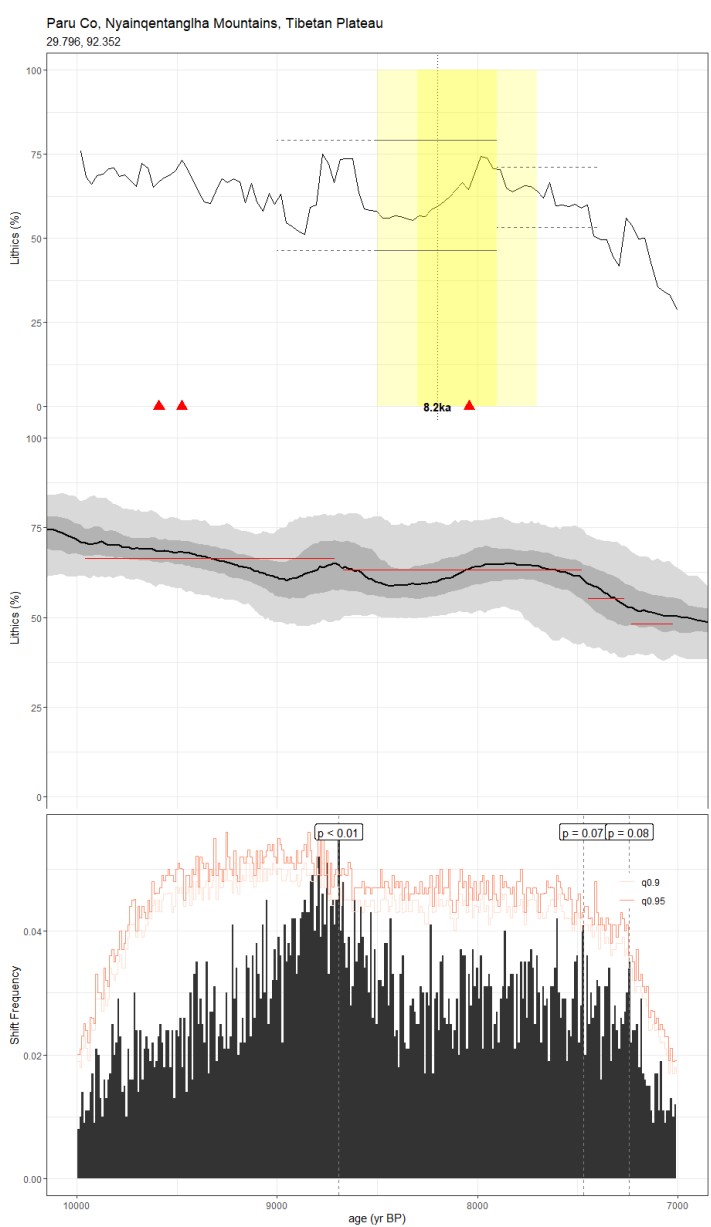

**Figure A44.** As in Fig. A3, but for the lake sediment (percent lithics) record of Bird et al., 2014 (ParuCo). CALIB 6.0 and the IntCal09 calibration curve were used in the construction of the published age model. We construct our age ensemble using BACON and IntCal20 via geoChronR.



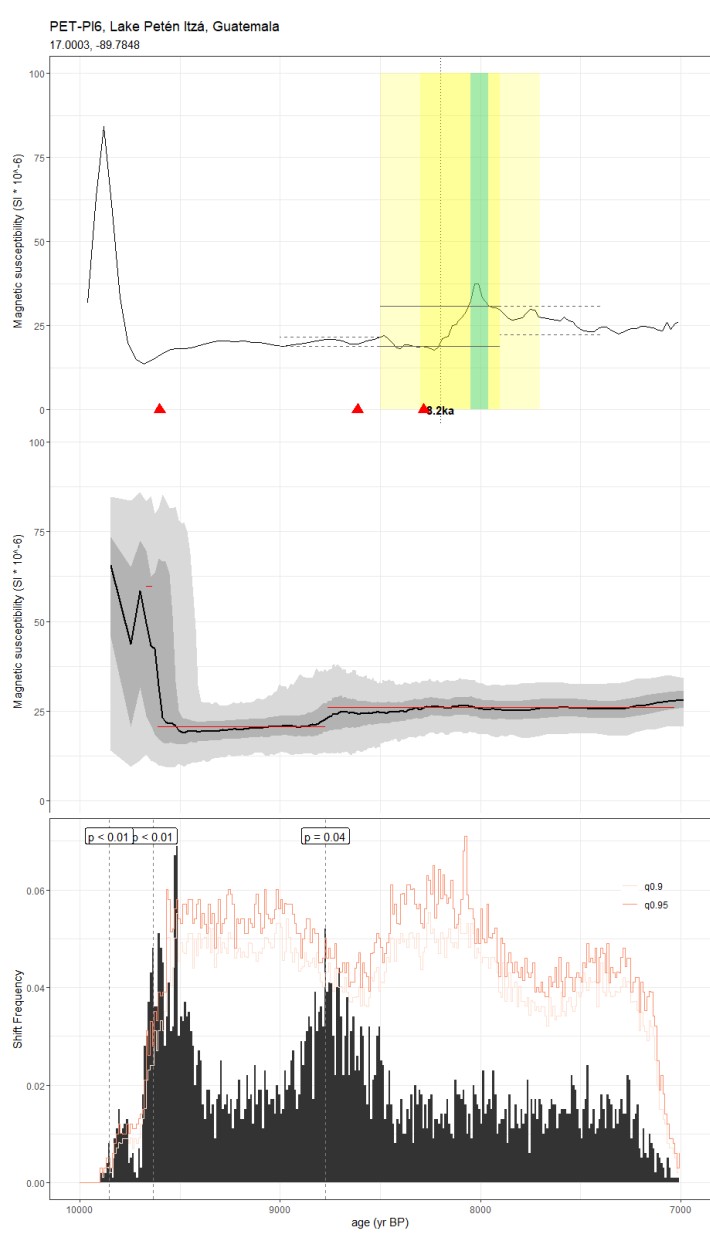

**Figure A45.** As in Fig. A3, but for the lake sediment magnetic susceptibility record of Escobar et al., 2012 (PETPI6). The published age model was generated using the OxCal algorithm with IntCal09 calibration curve. Here, we show an age ensemble created using the BACON algorithm with IntCal20 calibration curve generated by geoChronR.




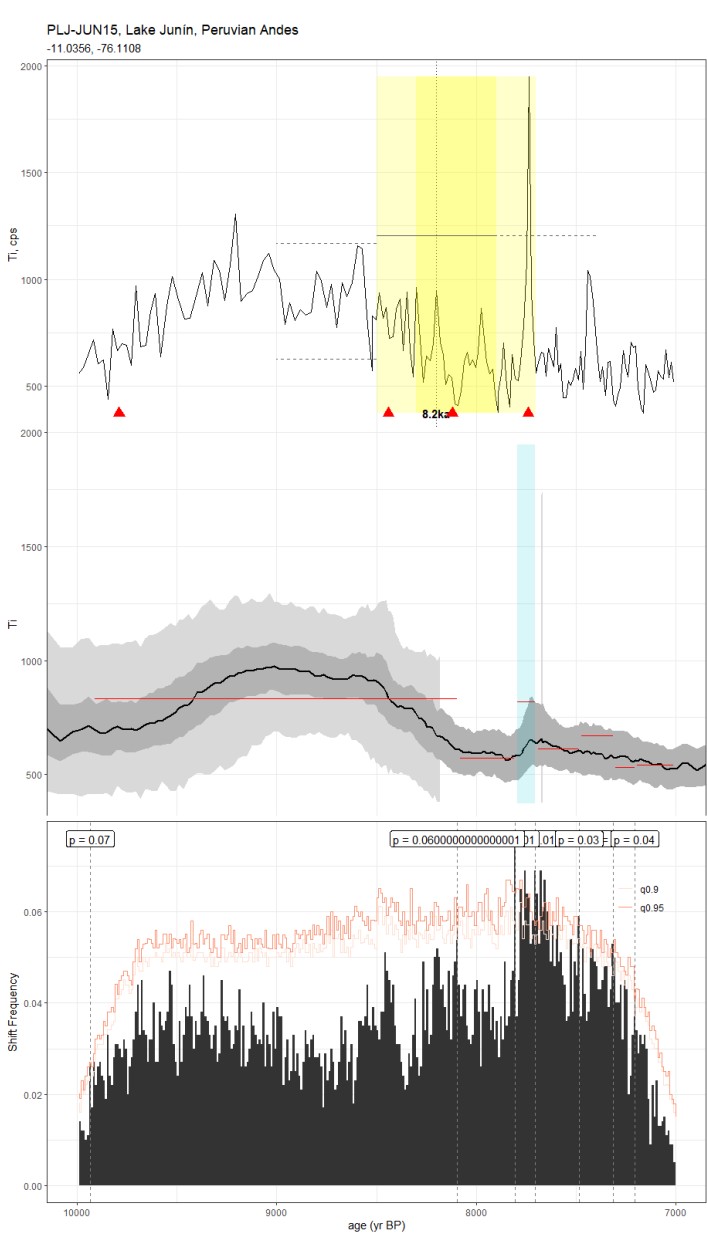

**Figure A46.** As in Fig. A3, but for the lake sediment titanium content record of Woods et al., 2020 (PLJJUN15). The published age model was created using the BACON algorithm with IntCal13 calibration curve. Here, we reconstruct a BACON ensemble using the IntCal20 curve in geoChronR.





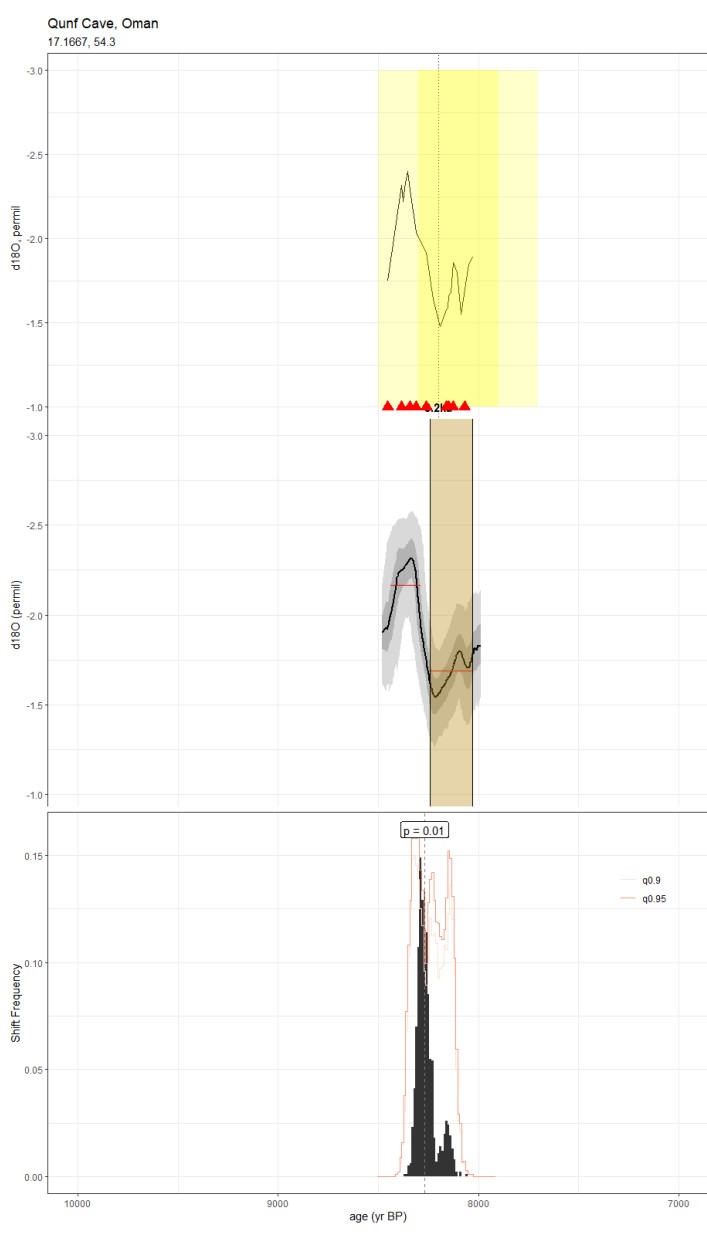

**Figure A47.** As in Fig. A3, but for the speleothem record of Cheng et al., 2009 (Q5Cheng). The method used in the construction of the published age model is unreported; here, we use BACON in geoChronR to generate our age ensemble.



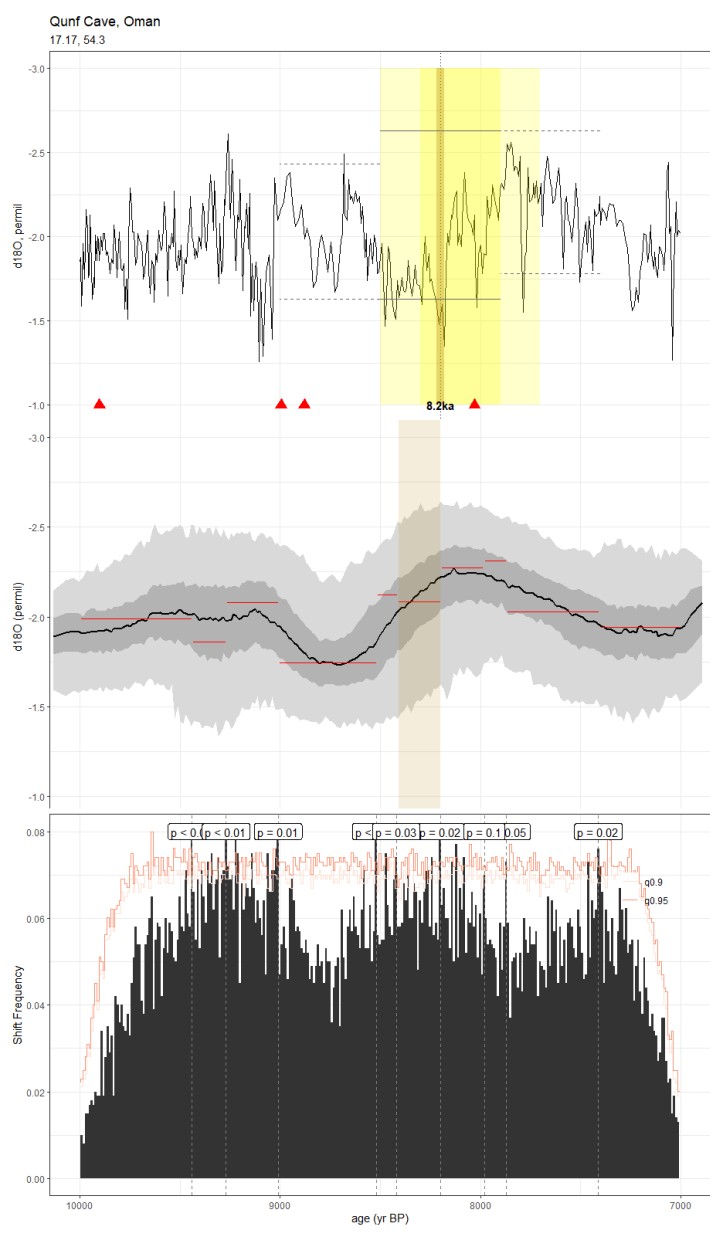

**Figure A48.** As in Fig. A3, but for the speleothem record of Fleitmann et al., 2007 (Q52007). The published age model was created via a polynomial fit to the age-depth curve of the Th–U data. Our age ensemble leverages the BACON algorithm included in geoChronR.



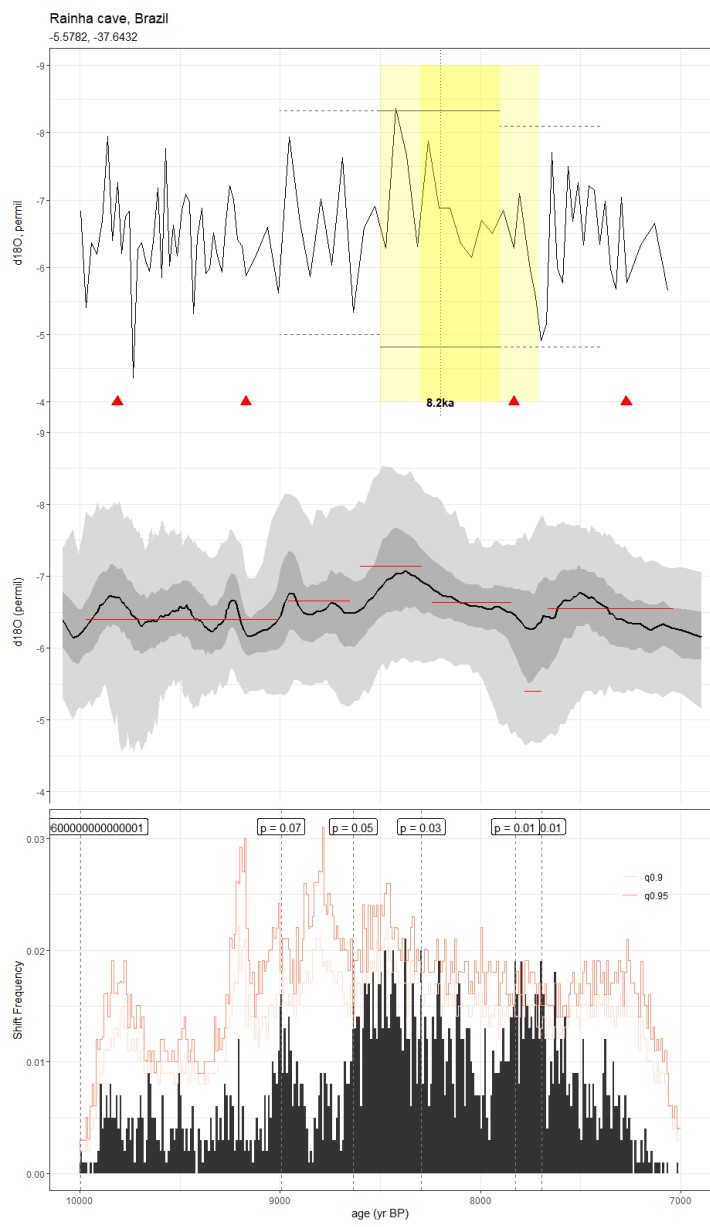

**Figure A49.** As in Fig. A3, but for the speleothem record of Cruz et al., 2009 (RN1). The method used in the construction of the published age model was unreported, but we leverage the Bchron ensemble supplied in version 2 of the SISAL database for our analyses.




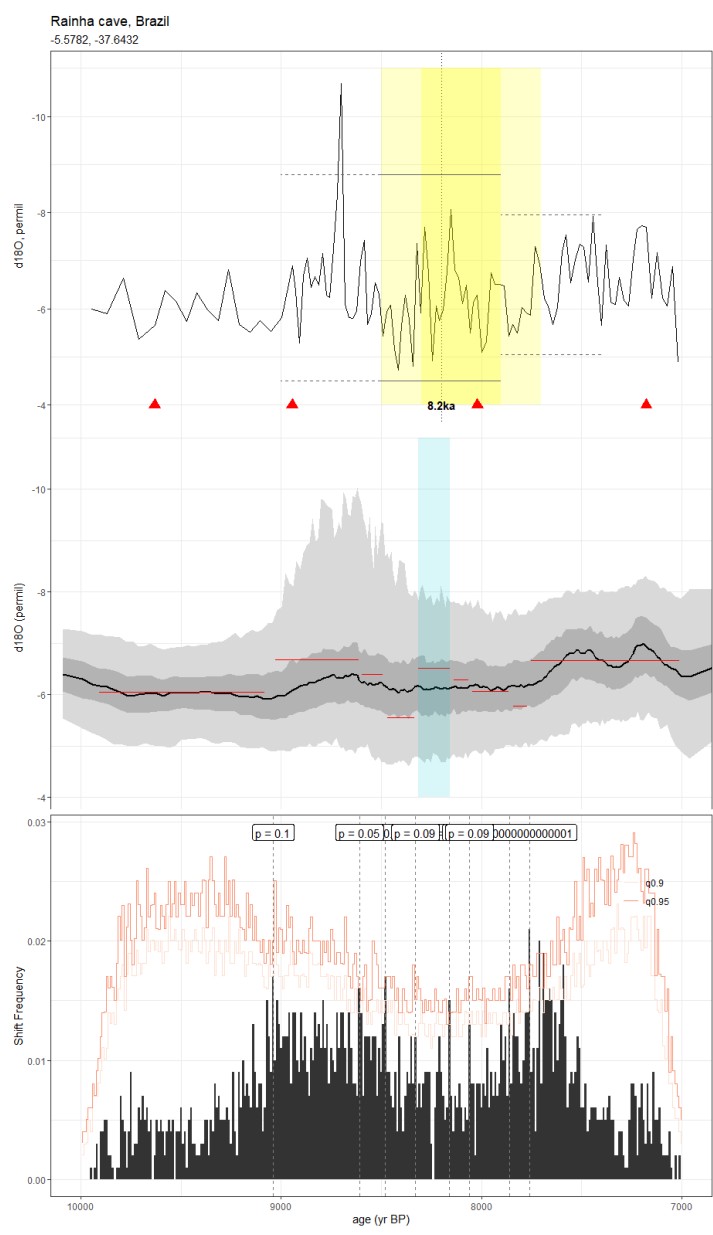

**Figure A50.** As in Fig. A3, but for the speleothem record of Cruz et al., 2009 (RN4). The method used in the construction of the published age model was unreported, but we leverage the Bchron ensemble supplied in version 2 of the SISAL database for our analyses.

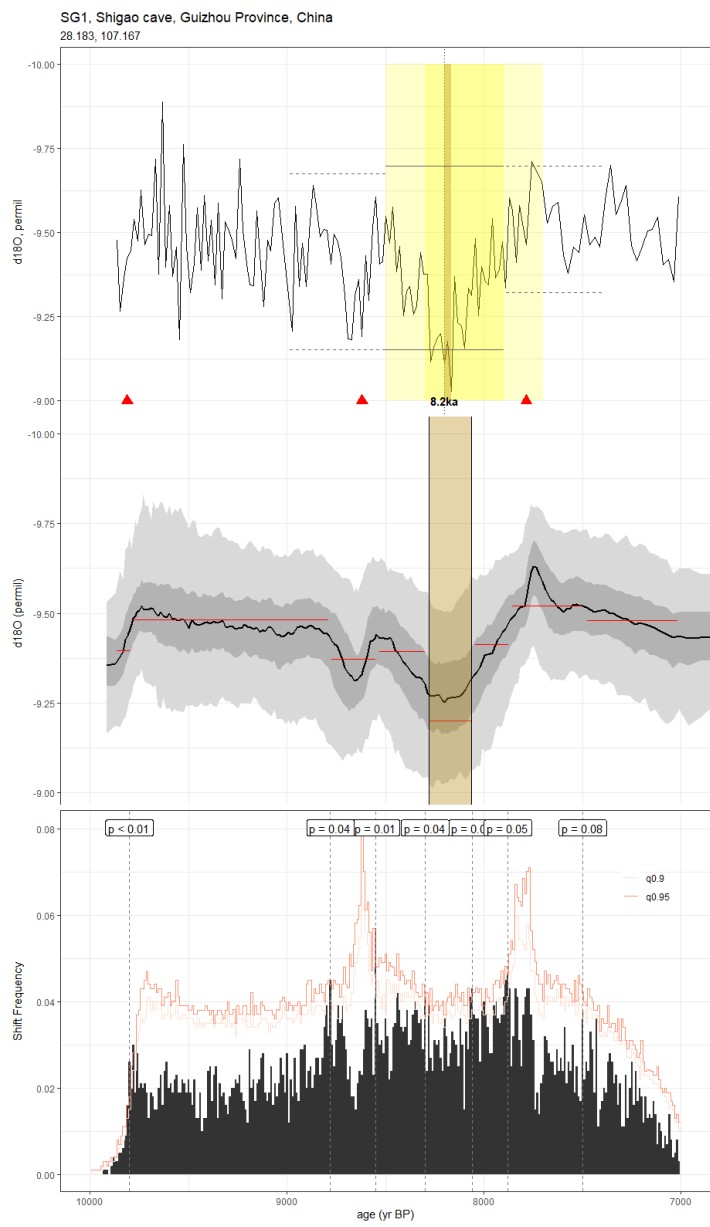

**Figure A51.** As in Fig. A3, but for the speleothem record of Jiang et al., 2012 (SG1). The published age model was constructed by linear interpolation between U/Th dates. Here, we leverage the BACON ensemble from SISALv2 for our analyses.





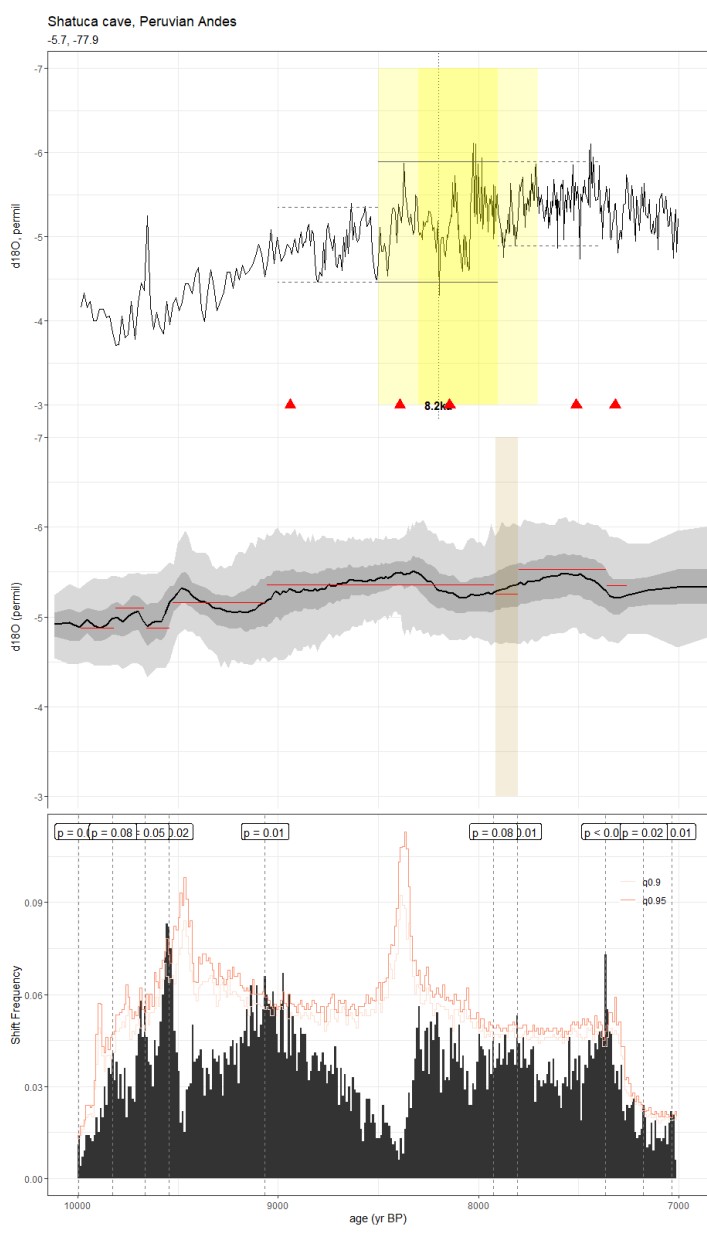

**Figure A52.** As in Fig. A3, but for the speleothem record of Bustamante et al., 2016 (Sha3). The published age model was developed using copRa. Here, we present the Bchron age ensemble generated for SISALv2.




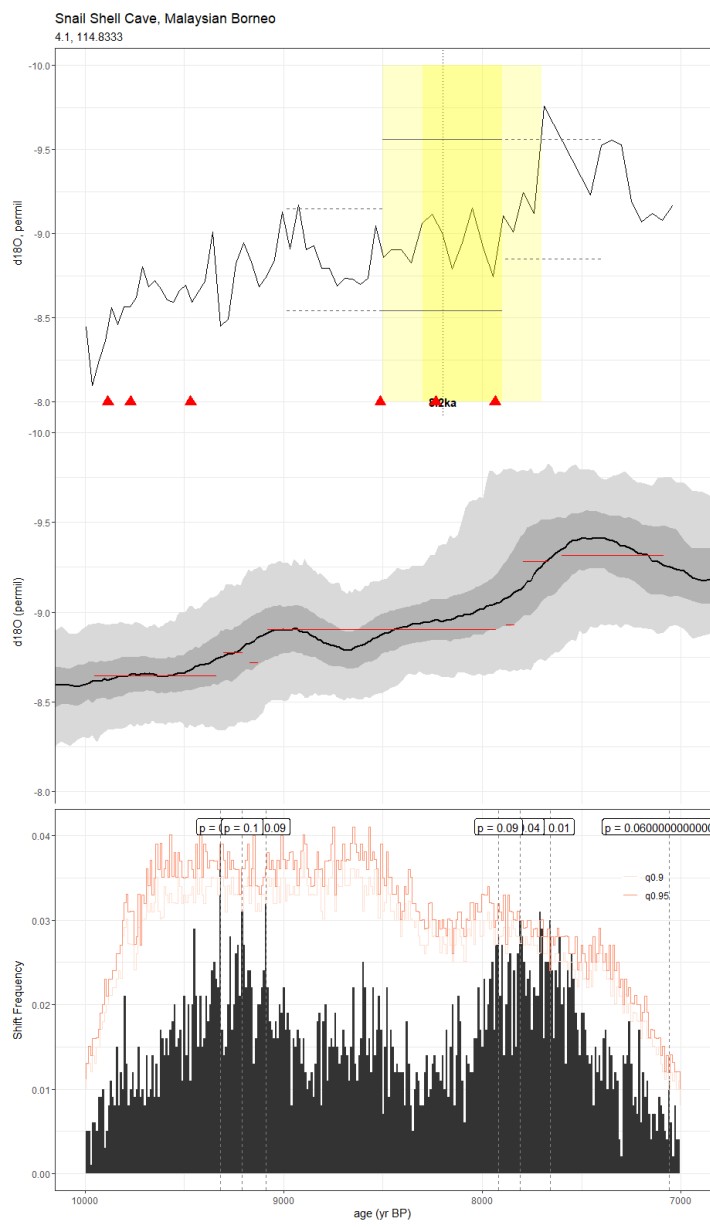

**Figure A53.** As in Fig. A3, but for the speleothem record of Carolin et al., 2016 (SSC01). StalAge was used to construct the published age model. We used the BACON algorithm included in geoChronR to generate our age ensemble.



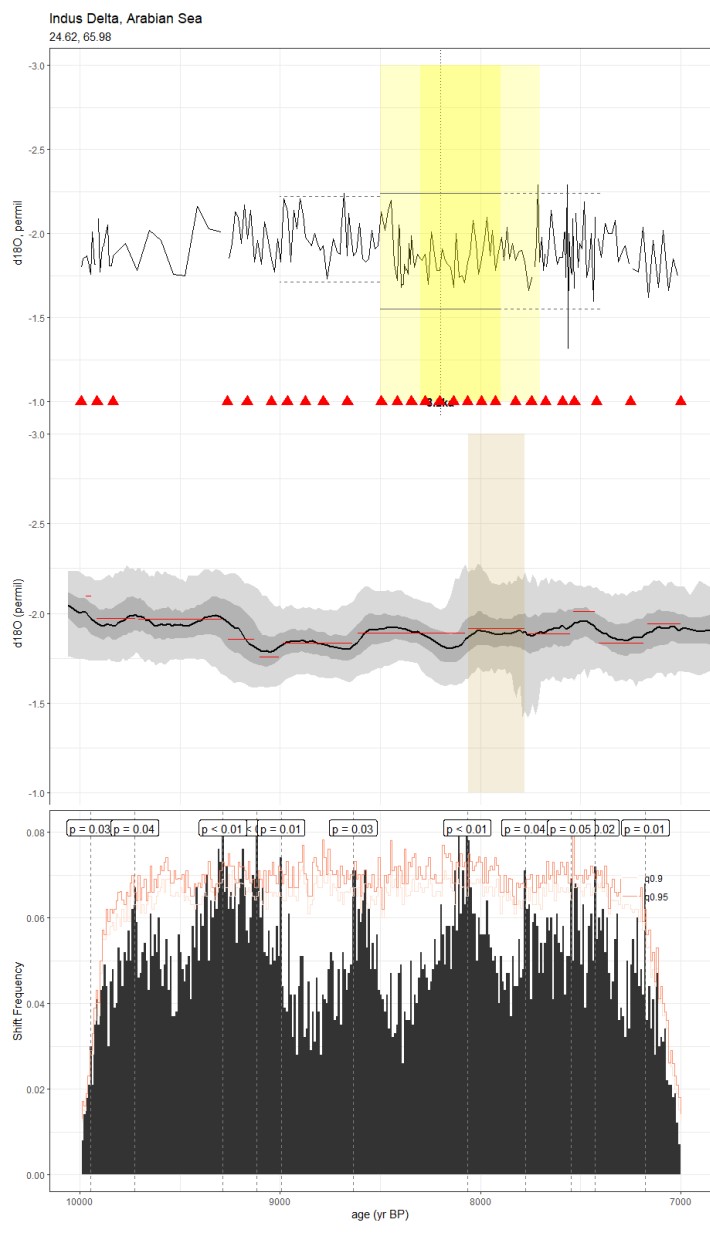

**Figure A54.** As in Fig. A3, but for the foraminifera record of Staubwasser et al., 2003 (Staubwasser63KA). The published age model was generated via a least-squares regression between 14C dates using the IntCal98 calibration curve. Here, we constructed our age ensemble using the BACON algorithm and IntCal20 calibration curve in geoChronR.





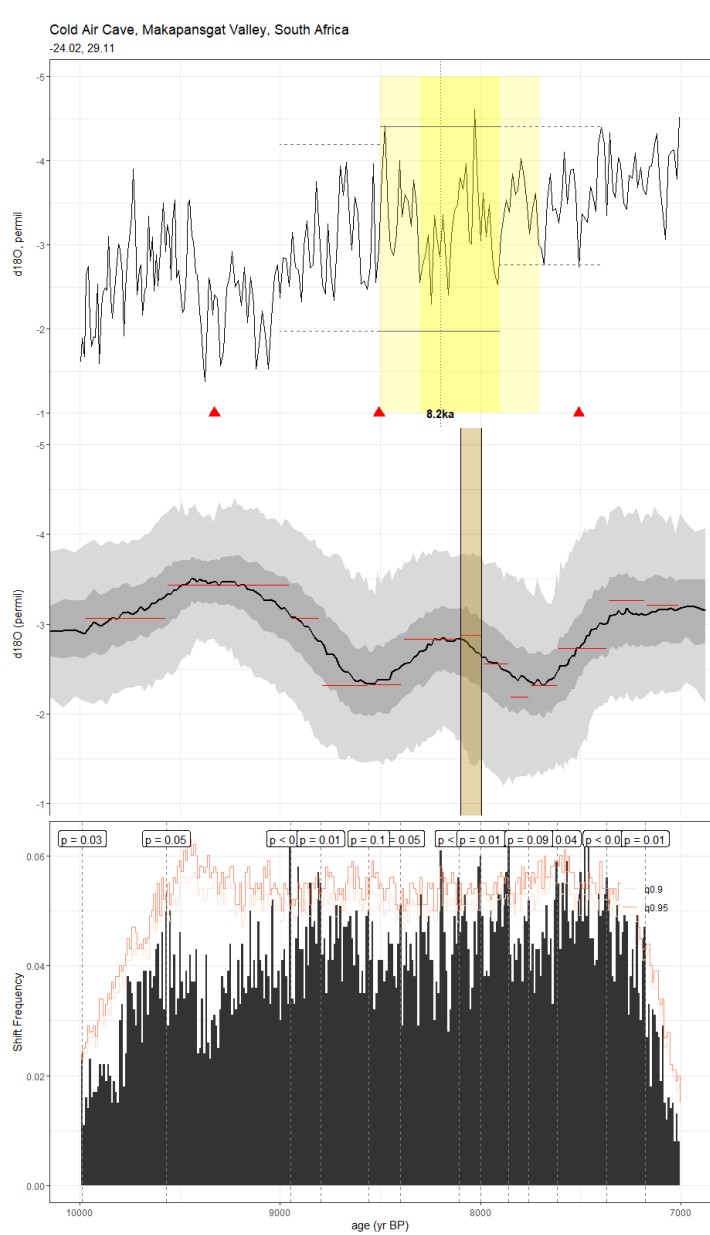

**Figure A55.** As in Fig. A3, but for the speleothem record of Holmgren et al., 2003 (T8). The published age model was constructed via linear interpolation between dates. Here, we construct our ensemble using the BACON age model algorithm in geoChronR.




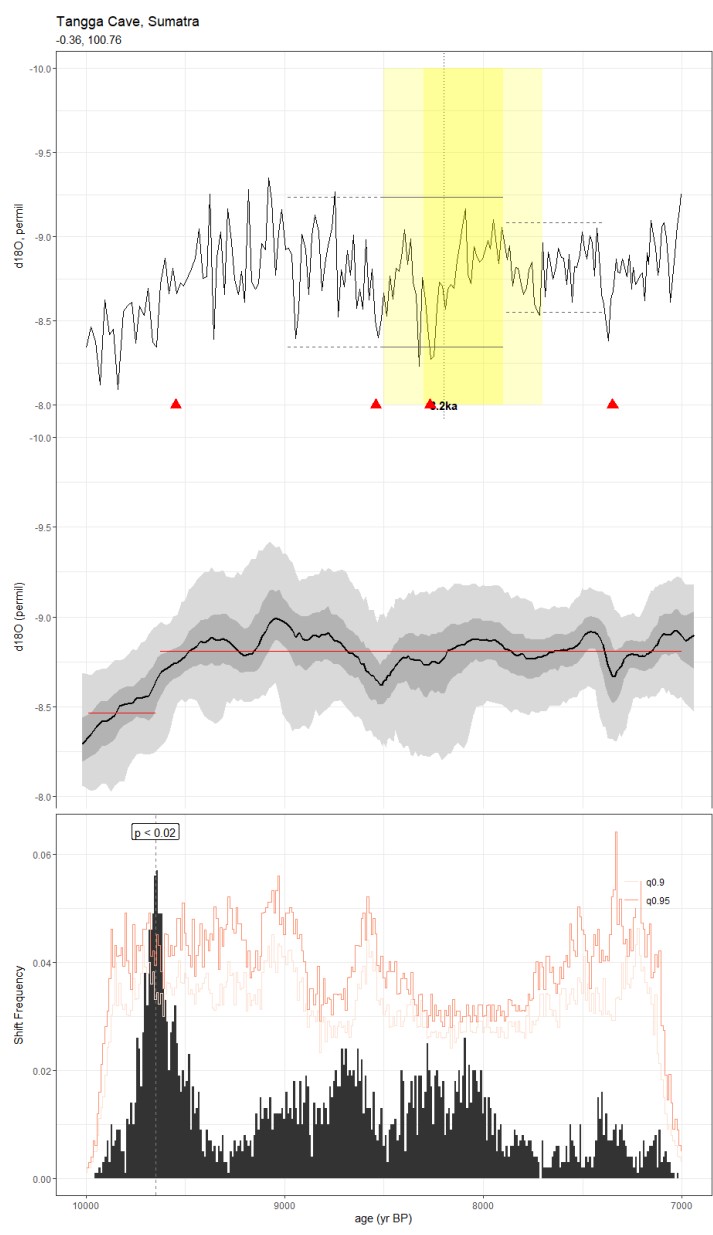

**Figure A56.** As in Fig. A3, but for the speleothem record of Wurtzel et al., 2018 (TA122). The published age model was constructed using the BACON algorithm. Here, we used the copRa ensemble generated for SISALv2.



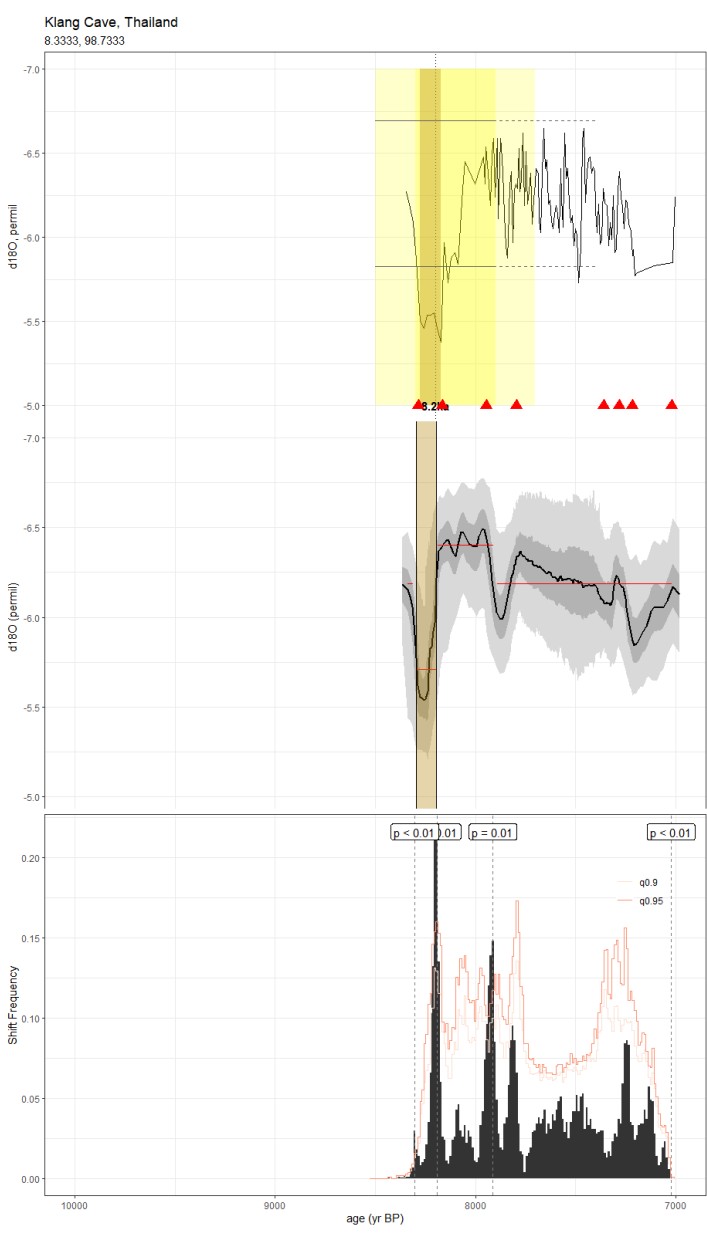

**Figure A57.** As in Fig. A3, but for the speleothem record of Chawchai et al., 2021 (TK07). The published age model was constructed using the BACON algorithm. Here, we used the BACON age ensemble supplied in the SISALv2 database.



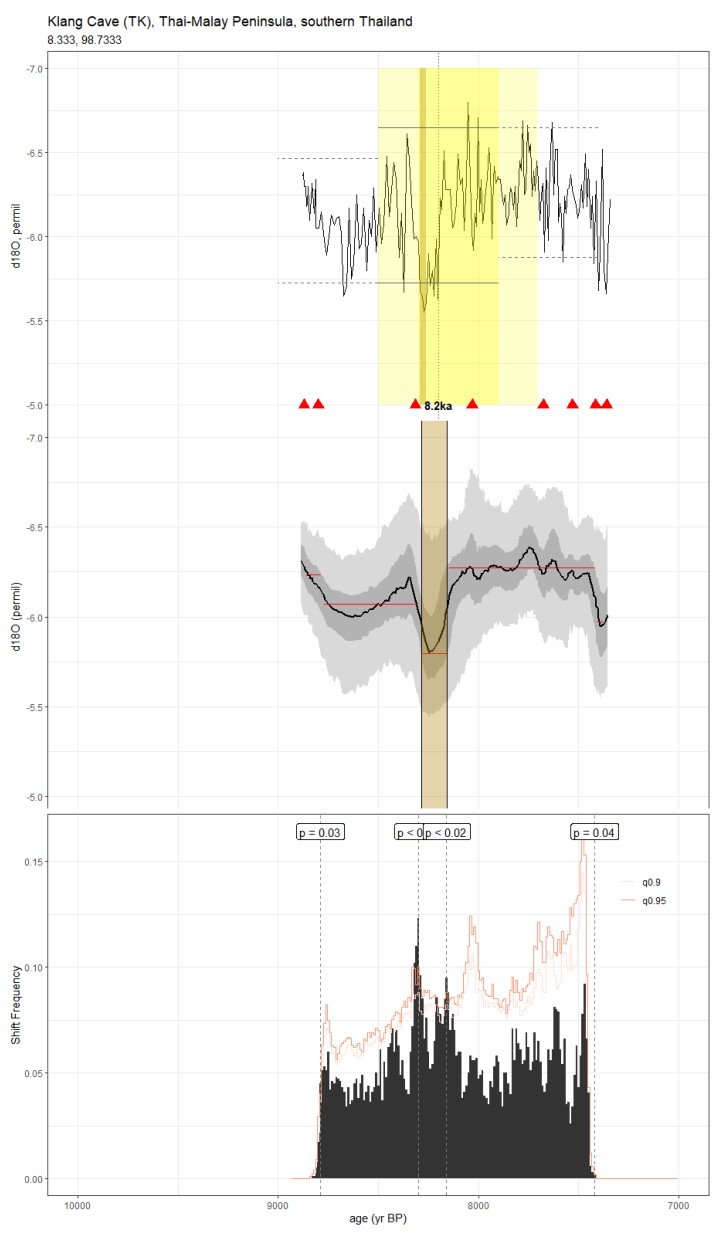

**Figure A58.** As in Fig. A3, but for the speleothem record of Chawchai et al., 2021 (TK20). The published age model was constructed using the BACON algorithm. Here, we used the BACON age ensemble supplied in the SISALv2 database.



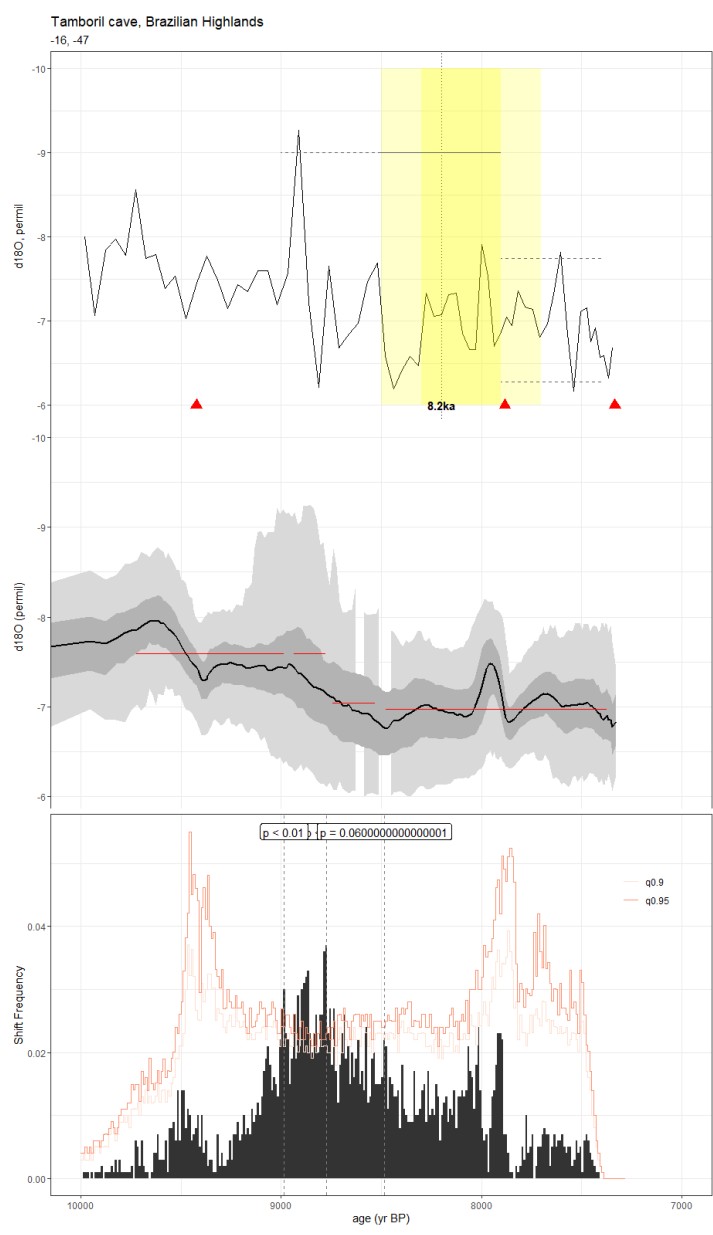

**Figure A59.** As in Fig. A3, but for the speleothem record of Ward et al., 2019 (TM6). The published age model was constructed using the copRa algorithm, though we use the BACON age ensemble supplied in the SISALv2 database for our analyses.





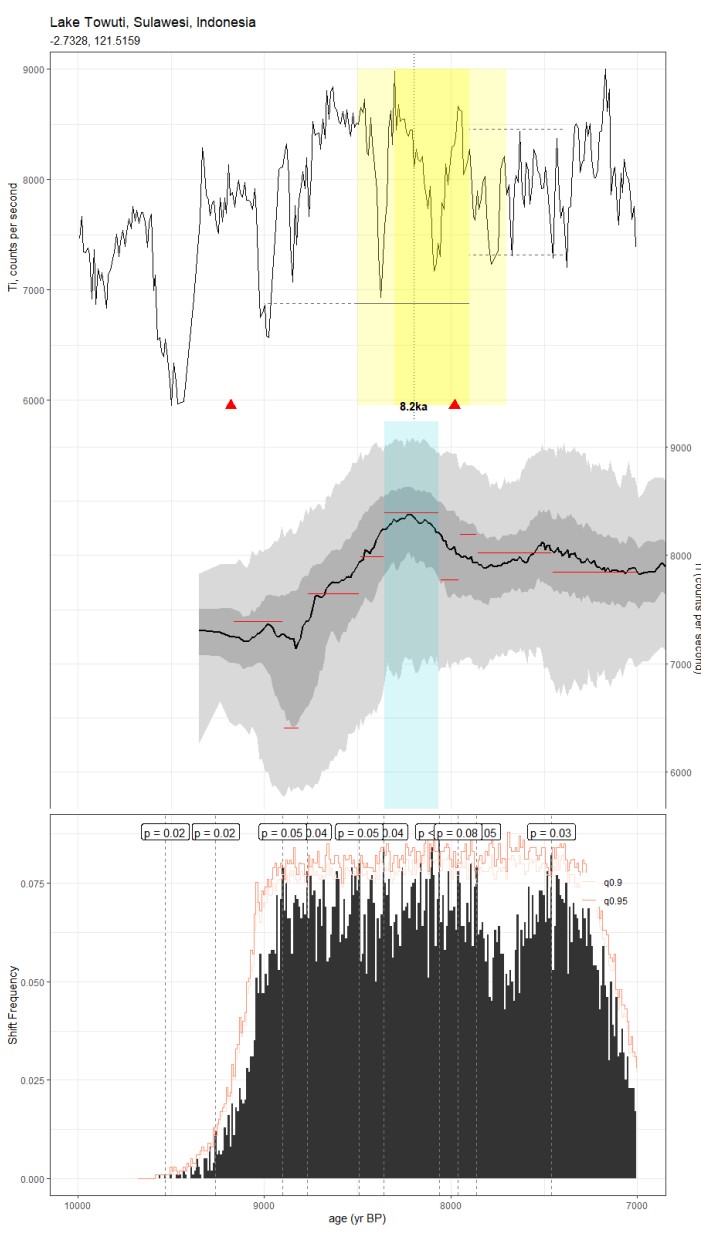

**Figure A60.** As in Fig. A3, but for the lacustrine sediment record of Russell et al., 2014 (TOW109B). CALIB 6.0 was used to construct the published age model, though we leveraged the BACON algorithm included in geoChronR to generate our age ensemble.



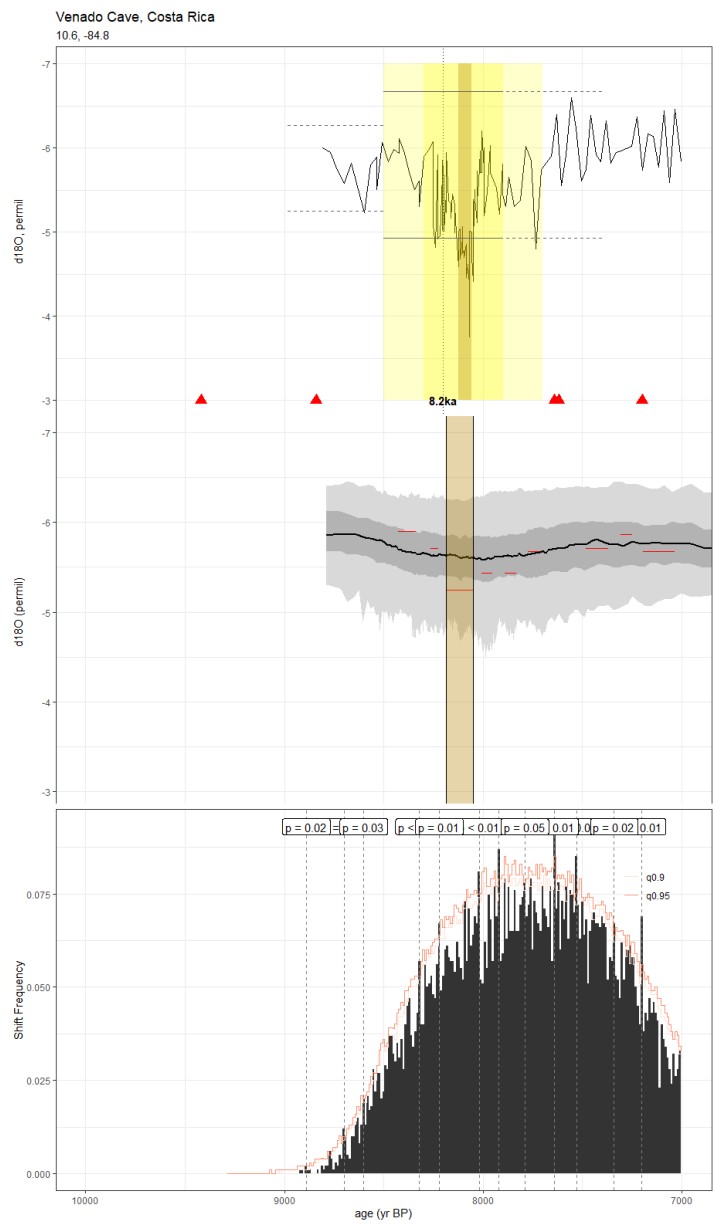

**Figure A61.** As in Fig. A3, but for the speleothem record of Lachniet et al., 2004 (V1). The published time series was aligned to a fifth-order polynomial best-fit age model between isochron dates. We employ the BACON ensemble provided by SISALv2 for our analyses.



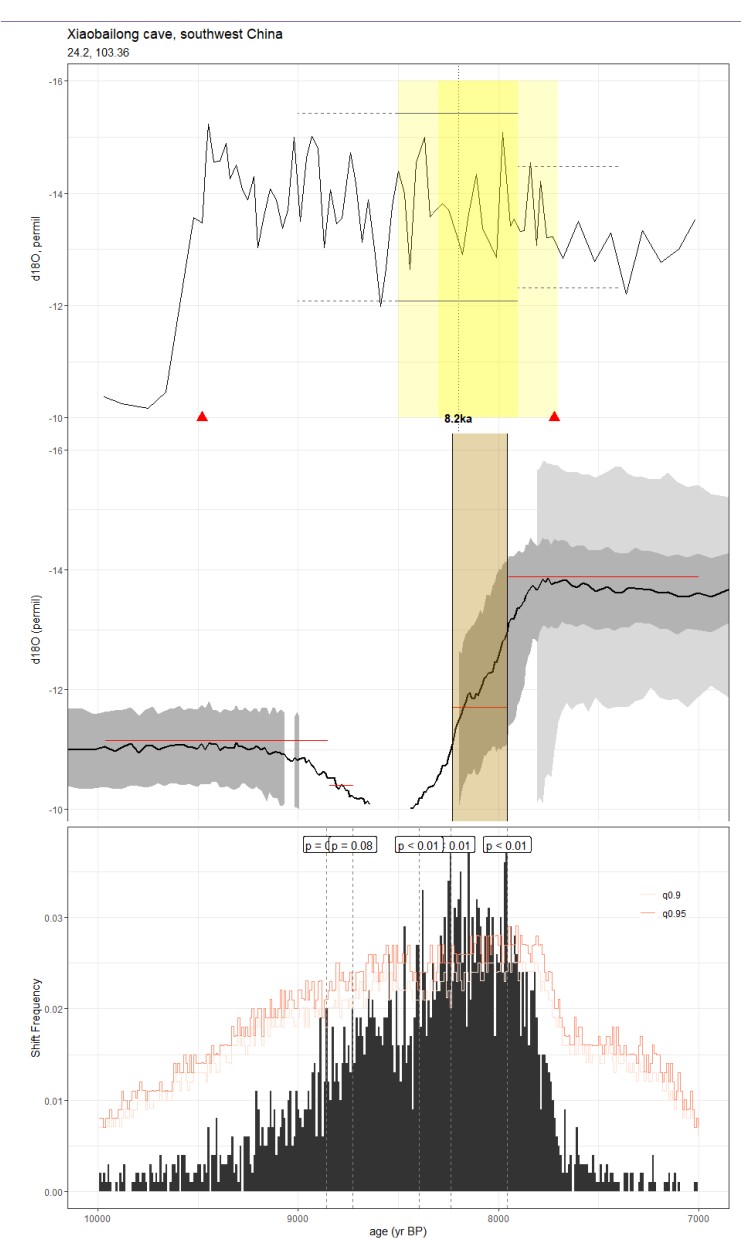

**Figure A62.** As in Fig. A3, but for the speleothem record of Cai et al., 2015 (XBL29). The published age model was derived from linear interpolation between radiometric dates. For our analyses, we leveraged the SISALv2 Bchron age ensemble.





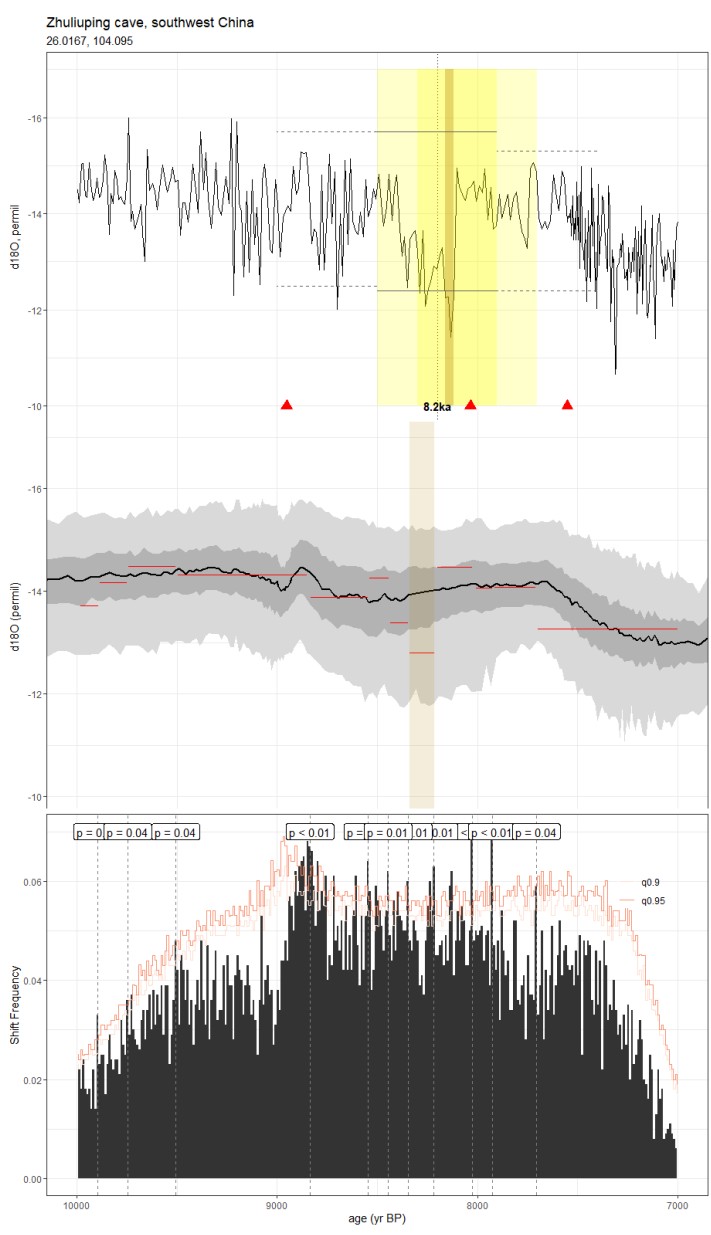

**Figure A63.** As in Fig. A3, but for the speleothem record of Huang et al., 2016 (ZLP1). The published age model was derived from linear interpolation between radiometric dates. For our analyses, we leveraged the SISALv2 BACON age ensemble.