# Peer review of "Data-model comparisons of the tropical hydroclimate response to the 8.2 ka Event with an isotope-enabled climate model"

_EGUsphere, 2024_

## Author Comment (AC1)

**RC1: 'Comment on egusphere-2024-3483', Anonymous Referee #1, 12 Dec 2024**

General comments

In this paper, Moore and colleagues have thoroughly analyzed the signal of the Holocene 8.2ka event in hydroclimatic proxy records from the tropics. The proxy-based signal is compared to simulation results obtained with the isotope-enabled iCESM global climate model. In the compilation, proxy records from different environments are considered: marine sediment records, lake sediment records and speleothems. As such this work extends the compilation recently published by Parker and Harrison (2022), who focused on the 8.2ka signal in speleothem records. Innovative aspects of this manuscript are the extensive analysis of the impact of age uncertainties and the comparison of the proxy-based hydroclimatic anomaly with a state-of-the-art isotope-enabled model simulation of the 8.2ka event. The paper is well-written, with a clear structure, and includes high-quality figures. The main result is a convincing proxy-based overview of the hydroclimatic response to the 8.2ka event in the tropics, showing clear regional variability. In my view, the manuscript requires some minor revisions, mostly associated with the detection methods used and the climate model experiment. I provide details below.

Main comments

Introduction, Line 59. "This period occurred during the otherwise stable Holocene epoch (11,700 years ago to present) and was driven by the discharge of around 163,000 km3 of meltwater from proglacial Lakes Ojibway and Agassiz (remnants of the Laurentide Ice Sheet) into the North Atlantic". The volume and source of the freshwater perturbation are actually still under discussion. For instance, the volume of the proglacial lakes is under discussion (Törnqvist and Hijma, 2012). In addition, Gregoire et al. (2012) argue that a collapse of the "ice saddle" over Hudson Bay could have played an important role in forcing the 8.2ka event. It is important to make the reader aware of this uncertainty. See also the recent discussion by Aguiar et al. (2021). Besides, it is confusing to refer to Lakes Ojibway and Agassiz as "remnants of the Laurentide Ice Sheet", so I suggest removing this part of the sentence.

Thank you for this comment. We have updated the wording to better reflect this nuance as follows:

*"This period occurred during the otherwise stable Holocene epoch (11,700 years ago to present) and is thought to have been driven by the discharge of around $1.63 \times 10^5$ $km^3$ of meltwater from proglacial Lakes Ojibway and Agassiz into the North Atlantic, triggering a large-scale salinity anomaly and resultant reduction in the strength of the Atlantic Meridional Overturning Circulation (AMOC; e.g., Barber et al., 1999; Ellison et al., 2006). However, the precise source, routing, and strength of the freshwater perturbation are still under discussion (e.g. Törnqvist and Hijma, 2012), ranging from an upper limit of $27.1 \times 10^5$ $km^3$ of freshwater released from the retreating Laurentide Ice Sheet (LIS) between 9 ka and 8 ka (Peltier 2004) to a smaller but more abrupt discharge of $5.3 \times 10^5$ $km^3$ between 8.31 ka and 8.18 ka (Li et al., 2012). Recent*

*data-model comparisons from Aguiar et al. (2021) suggest that an additional 8.2 × 10⁵ km³ of freshwater flowed into the Labrador Sea after the collapse of the Hudson Bay due to the routing of river discharge over the western Canadian Plains (Carlson et al., 2009)."*

Section 2.3.1. MM method. I propose including a schematic figure explaining how the MM method of detection works in practice, using two examples: one example with multiple events of the same sign and one example with multiple events of different signs. I am thinking of an extended version of Morrill's Figure 2. It is not clear to me what the actual modification is to the Morrill method. Please specify. If feasible, including such a schematic for the actR method would also be helpful for readers not familiar with this type of changepoint analysis.

Thank you for this suggestion. We have annotated two such event cases below (Figs. 1a and 1b), which will be included as supplemental figures in the revised manuscript.

Our MM method differs from the method presented in Morrill et al., (2013) in several ways as it serves primarily as a supplement to our actR results. We have updated the wording of that section to clarify that we did not perform the "leave one out" calculation of the standard deviation that Morrill et al. (2013) employed to account for noisy data and outliers in each reference window. Additionally, Morrill et al. (2013) define anomalous events as excursions with at least two adjacent anomalous values of the same sign (three for records with sub-decadal resolution). We modify this by requiring that events "must last at least 10 years and if multiple events are detected within the 7.9-8.5ka window, they are combined into a single event if there are no more than three data points or thirty years separating the different excursions". Finally, we clarify that Morrill et al., 2013 use a moving two-tailed z-test method to define the duration of their detected events, while we only consider the time between the initial and final anomalous data point in our duration calculation.

Per your suggestion, we have also included a schematic summarizing the actR analysis process below (Fig. 2), which will also be added to the Supplementary Material of the manuscript.

[Figure]

Figure 1a. A schematic illustrating the application of our modified Morrill method to the speleothem record of Lachniet et al., 2004 (V1), corresponding to Fig. A61 in our manuscript. The green and purple rectangles represent the range of values in each reference window, bracketed by the upper and lower bounds of x̄±2σ over each window. The brown rectangle highlights an anomalous isotopic enrichment event (8.124-8.058 ka) which is composed of three separate "events" (separated by 9-19 years of data which do not exceed the event threshold), consolidated into a single 8.2ka Event. The maximum absolute z-score over the event period (+3.4σ) is taken to represent the magnitude of the event.

[Figure]

Figure 1b. As for Fig. 1a, but for the record of Zhang et al., 2013 (LH2), corresponding to Fig. A33 in our manuscript. This record shows multiple events of opposing signs within the detection window: an anomalous isotopic depletion (-1.4σ, 8.221-8.208 ka) and an anomalous enrichment (+2.1σ, 8.138-8.129 ka). The event with the larger absolute value z-score is taken to represent the 8.2ka Event.

[Figure]

Figure 2. A schematic illustrating the actR analysis process. Step 1: Relevant records are identified and collated into our compilation based on the criteria outlined in the Methods (see Tables 1 and 2 in our manuscript). Records are then converted to the LiPD file format for analysis. Step 2: A 1000-member age model ensemble is developed using the geoChronR library in R, or, where available for our speleothem records, drawn from the ensembles presented in version 2 of the SISAL database. This allows us to propagate age uncertainties through each successive analysis step. Step 3: The age ensemble is appended to the proxy data to generate a 1000-member ensemble time series, including both age and simulated proxy data (σ/2) uncertainties (represented by the ribbons, where the outermost [lighter] bands represent extreme quantile values and the innermost [darker] bands the central quantile values).

The data are fit to a Gaussian distribution, and the change point analyses are conducted across this ensemble to determine the timing of changes in the proxy data. Step 4: The significance of the detected change points is tested by performing the same change point analyses against 100 isospectral surrogate time series, and the frequency of shifts is plotted as a histogram summarized in 10-year-long bins. The p-value is shown for shifts exceeding the α = 0.05 and α = 0.10 level (95% and 90% confidence intervals, respectively).

Section 2.4. Please make clear that the experimental setup for the iCESM simulation of the 8.2ka event does only partly follow the PMIP4 design reported by Otto-Bliesner et al. (2017). In the PMIP4 design, the 8.2ka simulation starts from an experiment with fixed forcing for 9.5 ka instead of 9 ka as is done here. In addition, in the PMIP4 design the freshwater hosing is applied in the Labrador Sea instead of across the entire northern North Atlantic.
We agree that it is important to specify how our experimental setup otherwise differs. We have updated the wording of that section to clarify that we followed the PMIP4 protocol for the timing and duration of the meltwater events (2.5 Sv for 1 year, followed by 0.13 Sv for 99 years), while the location of the meltwater forcing is notably different. We have also clarified that our hosing experiment branches from 9 ka boundary conditions (instead of 9.5 ka), and thus uses slightly different orbital and GHG configurations.

Discussion. In some regions, for example the Caribbean and SE Asia, there are records without any detected change located close to records with a clear signal for the 8.2ka event (see for instance Figure 2). How to interpret the absence of a signal in these records without significant change? It would be insightful to discuss the possible reasons for the absence of a signal.
You raise an important point that we have addressed more explicitly in our revision by adding the text below:

*"In central America and Brazil, records with no detected change are located close to records with a clear signal of the event (Fig. 5, Fig. 6). This could be due to several factors, including localized hydroclimate signals and variable age uncertainty and temporal resolution in the proxy records."*

Section 4.2. Simulated 8.2ka event. The results of the last 50 years of the "hose" experiment are taken to represent the modelled 8.2ka event. The mapped difference between "hose" and "ctrl" is used, for instance in Figure A2 and in the model-data comparison. It would be informative to know how the climate evolved through time in "hose" relative to "ctrl" (for instance related to detectability in the model result, see next point), so I propose including additional figures with modelled time-series of hydroclimate for the key regions shown in Figures 4 to 7. These additional figures could show 100-year time-series of both "ctrl" and "hose".
Thank you for the suggestion. We will include the following time series of precipitation δ$^{18}$O for these regions as supplemental figures (see Figs. 3-5 below).

[Figure]

[Figure]

Figure 3. The amount-weighted, area-weighted mean annual precipitation δ¹⁸O for the "ctrl" (red) and "hose" (blue) simulations over all of East Asia (manuscript Fig. 4; top panel), the western subregion defined by the +Δδ¹⁸Op anomaly in manuscript Fig. 4a (bottom left panel), and the eastern subregion defined by the -Δδ¹⁸Op anomaly in Fig. 4a (bottom right panel).

[Figure]

Figure 4. As in Fig. 3, but for NE South America (manuscript Fig. 5).

[Figure]

[Figure]

[Figure]

Figure 5. As in Fig. 3, but averaged over all of Southern Central America (top panel), the southern subregion defined by the +Δδ18Op anomaly in manuscript Fig. 6a (bottom left panel), and the northern subregion defined by the -Δδ18Op anomaly in Fig. 6a (bottom right panel).

Detectability of the simulated 8.2ka event. The question of detectability of the 8.2ka event is not only relevant for proxy records, but also for the model results. The difference between "hose" and "ctrl" includes anomalies produced by both internal variability and forced variability (i.e. by hosing). Ideally, one should not perform just one simulation for the 8.2ka event as is done here, but rather an ensemble experiment in which the members differ only in initial conditions and have identical freshwater forcing (hosing). The ensemble mean allows then to analyze the forced response and to separate it from the internal variability. I refer to Wiersma et al. (2011),

who performed this analysis for the temperature response associated with the 8.2ka event. I realize that it is probably not feasible within the framework of this study to perform several additional ensemble members with iCESM, but in my view the aspect of detectability in the model results should also be discussed in the paper. In addition, I suggest performing a statistical test to determine what results of "hose" are significantly different from the "ctrl" climate, and to include only those significantly different results in the model-data comparison.

Thank you for this comment. We agree that it is important to highlight the statistically significant changes in the model simulations. While running additional ensemble members to separate the forced response from internal variability is outside the scope of this study, to address the issue of significance, we have performed an unpaired two sample Student's t-test on the precipitation, effective moisture, and amount-weighted precipitation $\delta^{18}O$ fields from the hosed and control model runs. In our revised figures showing the changes in precipitation and precipitation $\delta^{18}O$ under hosing (see below), we now only plot those changes that exceed the 95% confidence level ($p < 0.05$).

[Figure]

Figure 6. The difference in amount-weighted $\delta^{18}O_p$ between the last 50 model years of the "hose" and "ctrl" simulations. Shading indicates anomalous changes that exceed the 95% confidence level ($p < 0.05$).

Other comments

Line 87. Why is it a critical tool? Please elaborate.

This sentence was removed and the paragraph was modified to more explicitly indicate how model data can complement proxy data to provide a more comprehensive understanding of the response of the climate system to the 8.2ka Event:

*"We further assess how well the proxy reconstructions compare to a new isotope-enabled model simulation of the 8.2ka Event. Such model simulations provide dynamical context to the sparse*

*proxy data and, by tracking water isotopes through the hydrologic cycle, facilitate more direct comparisons between proxy and model data than conventional climate models."*

Line 104. How is this sensitivity to hydroclimate determined?
The passage in question: *"To assess the tropical hydroclimate response to the 8.2ka Event, we developed an updated compilation of published, high-resolution, continuous, well-dated proxy datasets, collating records that span the period 7ka-10ka, cover latitudes from 30°N to 30°S, and which are sensitive to some aspect of hydroclimate variability."*

We address this in point (ii) of the subsequent paragraph: *"To constrain the timing and duration of the abrupt hydroclimate anomaly associated with the 8.2ka Event, the datasets in this compilation were screened to meet the following criteria: (i) data resolution of 50 years or better over the period of 7ka-10ka; **(ii) based on hydroclimate-sensitive proxy data interpreted by authors as reflecting precipitation amount or intensity, the isotopic compositions of environmental water (including precipitation, lake water, and seawater), effective moisture, lake level, fluvial discharge, or sea surface salinity (SSS)**; and (iii) contain at least three radiometric dates over the 7ka-10ka interval. Emphasis was placed on collecting water isotope-based records to enable more direct comparison with isotope-enabled climate model simulations."*

Line 151. Why 10 years? On what is this based?
The actR parameter in question (summary.bin.step) controls how the shift detection information is summarized in the output, and has no bearing on the change point calculation itself. Because the majority of our records are decadally resolved, and because we wanted to constrain the timing of significant shifts as precisely as possible, this value was chosen for our data processing.

Line 166. Why 50 or 100 years? Why not taking just 50 years?
Thank you for this question. Our parameter choices were based on the following considerations: The large dataset required parameters that would effectively accommodate most time series in our compilation while minimizing spurious shift detections. For all but one record in our compilation, the 100-year minimum segment length optimally captured abrupt changes without detecting broader climatic trends or less significant (spurious) shifts. For the PAD07 speleothem record of Cheng et al. (2009; Fig. A43 in our manuscript), it was necessary to reduce the minimum segment length to 50 years to capture the clear isotopic depletion near 8.2 ka that was otherwise smoothed over when employing our longer 100-year minimum segment length (see Fig. 7 below). This was the only record in our compilation that necessitated such an adjustment, and we have clarified this in the manuscript. A comprehensive sensitivity analysis of this parameter would be valuable to systematically determine the optimal minimum segment lengths across different types of paleoclimate records. However, this was outside the scope of our current study.

[Figure]

Figure 7. A comparison of shifts in mean (red horizontal lines, thickened for visibility) detected in the ensemble time series for PAD07 (Cheng et al., 2009) using a minimum segment length parameter of 100 (top panel) and 50 (bottom panel).

Line 183. In the actR method, the time window for detection (7.9 to 8.3 ka) is different from the window used in the MM method (7.9 to 8.5 ka). Why this difference and what is the consequence? In addition, is there also a minimum duration for detection in the actR method? The analysis window used in the MM method is the same as the window defined in Morrill et al. (2013), maintained here for some degree of parity with their original analysis, which they chose "to bracket the [8.2ka] event, while accommodating errors in the age models of several hundred years". The tighter ("significant") window that we employed in our actR method (7.9-8.3ka) is based on the duration of the 8.2ka Event in Greenland ice core data within age uncertainties

(8.25ka to 8.09ka; Thomas et al., 2007). Additionally, we define a second, broader actR window from 7.7-8.5ka to further account for age uncertainty across which we define our "tentative" events. The actR method allows us to leverage new, robust age modeling tools and the propagation of age uncertainties through our changepoint analyses to better estimate the timing and duration of abrupt events, like the 8.2ka Event, while the MM method adds supporting evidence to our interpretation of the sign and magnitude of the detected changes.

To your second question, the minimum event duration length using the actR method is 20 years. We have added this to our Methods section in our revision, and we thank you for bringing this omission to our attention.

Line 212. Are the forcings for the "ctrl" simulation identical to the 400-year-long 9ka simulation? If not, what are the differences? And is the climate stable in "ctrl" or is there still a trend in the surface conditions, signifying adjustment to the different forcings?
Yes, the preindustrial control simulation is branched from the 9 ka simulation and all forcings and boundary conditions are identical to the 400-yr 9 ka simulation. Quasi-equilibrium has indeed been reached in the last 100 years of the control run (analyzed in this study), based on the lack of a trend in the global average surface air temperature (see Fig. 8 below).

[Figure]

Figure 8. Time series of simulated global mean surface air temperature (SAT) for the last 100 years of the control run (red) and all 100 years of the hosed run (blue).

4.3 Data-Model comparisons. It is noteworthy that the regional structure of the 8.2ka event found in this study resembles the hydroclimatic anomaly in a simulation of the Younger Dryas cold event in which freshwater forcing applied in the North Atlantic also plays an important role (Renssen et al. 2018). The hydroclimatic response for the Younger Dryas shows wetter conditions in the Caribbean, SE South America, Southern Africa and Madagascar, but drier

conditions in S Central America, the Arabian Peninsula and SE Asia, broadly consistent with the proxy-based signal of the 8.2ka event provided in the present paper in Figure 2.

Thank you for pointing this out. There are indeed some interesting similarities between these two data sets and thus, we have added the following sentence to our Results in 3.5 Data-Model Comparisons:

*"Similar hydroclimate features also appear in simulations of the Younger Dryas cold event from Renssen et al (2018)."*

However, we don't emphasize the comparisons between our proxy compilation and this study beyond this for several reasons: (1) the Younger Dryas cold event had differing boundary conditions and a much larger amplitude than the 8.2ka event, (2) the majority of the proxy records in Fig. 2 are speleothem $\delta^{18}O$ records, a proxy for precipitation $\delta^{18}O$, while the modeling results of Renssen et al. (2018) show the change in soil moisture, and (3) the fact that the most robust signature in our proxy records is the drying/isotopic enrichment event in East Asia, which does not agree with the simulated increase in soil moisture in that region in Renssen et al. (2018).

Line 238. "… a lack of agreement of the sign or presence of an event". Do you mean a lack of agreement between the two detection methods? Please clarify.

Yes, thank you for the suggestion. The sentence has been updated to make this explicit: "The remaining 31 records in our compilation displayed a lack of agreement between the two detection methods in the sign or presence of an event and are thus excluded from further analysis."

Figures 2-7. The grey symbols are very hard to see, so I suggest improving their visibility.

Thank you for this suggestion. Because the size of the symbols scales with the magnitude of the Z-scores to emphasize the most prominent signals in the records, the grey symbols (indicating no change) are typically small. However, we agree that it is important to identify these records, and so we have added black outlines to the grey symbols in our revised figures, which we believe has made them more visible (e.g. see Fig. 9 below).

[Figure]

Figure 9. Summary of the 8.2ka Events detected using our actR method for the paleoclimate records included in this study. Records with drier and/or isotopically enriched events are shown

in brown, records with wetter and/or isotopically depleted events are shown in green, and records in which no event was detected are shown in white. Stippling indicates that a "significant" event was detected in a given record by actR with event "start" and "end" times within the 7.9-8.3ka interval at the $p < 0.05$ significance level. Slashed hatching indicates the presence of a "tentative" hydroclimate anomaly, with either a "significant" event detected outside of the 7.9-8.3ka window (between 7.7-8.5ka) or an event within that window where $0.1 > p \geq 0.05$. Symbol size is scaled by $250\ln(1+|z|)$, calculated from the per-record mean and standard deviation over the 7ka-10ka interval. Symbols are mapped over the simulated anomalous (a) amount-weighted oxygen isotopic composition of precipitation, (b) precipitation amount, (c) and effective moisture calculated from difference between the last 50 years of the iCESM "hose" and "ctrl" experiments at the 95% confidence level ($p < 0.05$).

Figures 3-7. What do the grey isolines represent? Please explain.
The grey isolines represent the average oxygen isotopic composition of precipitation from the control simulation, however, they are extraneous and so have now been removed from the revised figures.

Figures 4-7. In several cases, there are no proxy records of a specific type in a region (for example marine records in East Asia, Figures 4c and f). It is not very meaningful to show the simulation results in these cases, as there is no data-model comparison possible. So, I propose to only show the maps of key regions if there are proxy records available for the data-model comparison.
Thank you for the suggestion. We agree that the panels lacking proxy data are unnecessary. These extraneous panels have been removed from the revised figures.

Figure 5. Legend is missing.
Thank you for finding this. We have corrected it in Fig. 10 (see below), and have incorporated your suggestion from the previous comment as well. The MM results will be moved to the Supplemental Material.

[Figure]

Figure 10. Data-model comparison of IPCC region 11: northeastern South America. Symbols are scaled to 400ln(1+|z|) for visibility.

Figure A2a and b. According to the figure caption, the contours in these figures indicate the "range of temperatures in the "ctrl" simulation over the full 100 years". It is not clear what temperatures these isotherms represent, so I do not see how these contours are useful, and I suggest removing them from the figure.
Thank you for the suggestion. The unfilled contours representing the isotherms in the control simulation have been removed.

Additional references

Gregoire, L. J., et al. (2012) Deglacial rapid sea level rises caused by ice-sheet saddle collapses. Nature 487:597 219-222. https://doi.org/10.1038/nature11257.

Renssen, H., et al. (2018) The global hydroclimate response during the Younger Dryas event. Quaternary Science Reviews 193, 84-97. https://doi.org/10.1016/j.quascirev.2018.05.033

Törnqvist, T.E., Hijma, M.P. (2012) Links between early Holocene ice-sheet decay, sea-level rise and abrupt climate change. Nature Geoscience 5, 601-606. https://doi.org/10.1038/NGEO1536`.

Wiersma, A.P., et al. (2011) Fingerprinting the 8.2ka event climate response in a coupled climate model. Journal of Quaternary Science 26, 118-127, https://doi.org/10.1002/jqs.1439.

**References**

Aguiar, W., Meissner, K. J., Montenegro, A., Prado, L., Wainer, I., Carlson, A. E., and Mata, M. M.: Magnitude of the 8.2 ka event freshwater forcing based on stable isotope modelling and comparison to future Greenland melting, Scientific Reports, 11, 5473, https://doi.org/10.1038/s41598-021-84709-5, 2021.

Barber, D. C., Dyke, A., Hillaire-Marcel, C., Jennings, A. E., Andrews, J. T., Kerwin, M. W., Bilodeau, G., McNeely, R., Southon, J., Morehead, M. D., and Gagnon, J.-M.: Forcing of the cold event of 8,200 years ago by catastrophic drainage of Laurentide lakes, Nature, 400, 344–348, https://doi.org/10.1038/22504, 1999.

Carlson, A. E., Clark, P. U., Haley, B. A., and Klinkhammer, G. P.: Routing of western Canadian Plains runoff during the 8.2 ka cold event, Geophysical Research Letters, 36, 2009GL038778, https://doi.org/10.1029/2009GL038778, 2009.

Cheng, H., Fleitmann, D., Edwards, R. L., Wang, X., Cruz, F. W., Auler, A. S., Mangini, A., Wang, Y., Kong, X., Burns, S. J., and Matter, A.: Timing and structure of the 8.2 kyr B.P. event inferred from δ18O records of stalagmites from China, Oman, and Brazil, Geology, 37, 1007–1010, https://doi.org/10.1130/G30126A.1, 2009.

Ellison, C. R. W., Chapman, M. R., and Hall, I. R.: Surface and Deep Ocean Interactions During the Cold Climate Event 8200 Years Ago, Science, 312, 1929–1932, https://doi.org/10.1126/science.1127213, 2006.

Li, Y.-X., Törnqvist, T. E., Nevitt, J. M., and Kohl, B.: Synchronizing a sea-level jump, final Lake Agassiz drainage, and abrupt cooling 8200years ago, Earth and Planetary Science Letters, 315–316, 41–50, https://doi.org/10.1016/j.epsl.2011.05.034, 2012.

Morrill, C., Anderson, D. M., Bauer, B. A., Buckner, R., Gille, E. P., Gross, W. S., Hartman, M., and Shah, A.: Proxy benchmarks for intercomparison of 8.2 ka simulations, Clim. Past, 9, 423–432, https://doi.org/10.5194/cp-9-423-2013, 2013.

Peltier, W. R.: Global glacial isostasy and the surface of the ice-age Earth: The ICE-5G (VM2) model and GRACE, Annu. Rev. Earth Planet. Sci., 32, 111–149, https://doi.org/10.1146/annurev.earth.32.082503.144359, 2004.

Renssen, H., Goosse, H., Roche, D. M., and Seppä, H.: The global hydroclimate response during the Younger Dryas event, Quaternary Science Reviews, 193, 84–97, https://doi.org/10.1016/j.quascirev.2018.05.033, 2018.

Roberts, W. H. G. and Hopcroft, P. O.: Controls on the Tropical Response to Abrupt Climate Changes, Geophys. Res. Lett., 47, https://doi.org/10.1029/2020GL087518, 2020.

Thomas, E. R., Wolff, E. W., Mulvaney, R., Steffensen, J. P., Johnsen, S. J., Arrowsmith, C., White, J. W. C., Vaughn, B., and Popp, T.: The 8.2ka event from Greenland ice cores, Quaternary Science Reviews, 26, 70–81, https://doi.org/10.1016/j.quascirev.2006.07.017, 2007.

Törnqvist, T. E. and Hijma, M. P.: Links between early Holocene ice-sheet decay, sea-level rise and abrupt climate change, Nature Geosci, 5, 601–606, https://doi.org/10.1038/ngeo1536, 2012.

---

## Author Comment (AC2)

**RC2: 'Comment on egusphere-2024-3483', Anonymous Referee #2, 17 Jan 2025**

The study by Moore et al. compilated global hydroclimate proxies for the 8.2ka event in the tropics and subtropics and compared them with the hosing simulation by the state-of-the-art isotope enabled CESM.

Major novelties:

1. They developed an updated compilation of high-resolution, continuous, well-dated proxy datasets. This is important to the broad paleoclimate community.
2. They introduced the use of the Abrupt Change Toolkit in R (actR) for event detection, which better accounts for age model uncertainties in proxy records. As a result, they quantified the starting, ending, and duration of the 8.2ka event.
3. They revealed a more complex, regionally specific hydroclimate response pattern rather than a simple hemispheric dipole in the 8.2ka event.

However, the paper lacks depth in discussing the source of model-data differences, regional hydroclimate mechanisms, and the responses and of δ18Op, making it feel dry due to excessive qualitative descriptions of proxy and modeling results. In my opinion, it could be published in climate of the past, but a list of concerns should be addressed.

We appreciate your feedback. While a detailed analysis of the mechanisms of the hydroclimate changes in the model is outside the scope of this study, we have performed a decomposition of the isotopic signals in the model and included further discussion of the regional hydroclimate changes in the model, the possible underlying mechanisms, and how they compare with other climate model simulations and proxy data, as detailed below. We hope these additions add depth to our analysis and discussion.

Major concerns:

1. Condense the proxy and model results description while enhancing analysis of model-data differences. Focus on presenting actR results primarily, with MM results moved to supplementary materials for greater conciseness. This paper's unique contribution lies in the detailed regional data-model comparison, but thoughtful discussions lack. For instance, the δ18Op responses in East Asia during the 8.2ka event exhibit an east-west dipole pattern, contrasting with the uniform enriched isotopic signal seen in the Heinrich events. Despite similar hosing experiments, it is intriguing to explore why such discrepancies exist between these events.

Thank you for this helpful suggestion. We have moved many of the MM results to the Supplemental Material for conciseness. We have also enhanced the discussion of the regional δ$^{18}$Op responses in the model and elaborated on comparing these features to the proxy records. Regarding the east-west dipole pattern in East Asia, we now point out two other modeling studies that have demonstrated a large zonal asymmetry in the δ$^{18}$Op response in this region under meltwater forcing: Lewis et al. (2010) and Pausata et al. (2011). In Lewis et al. (2010), the

response of $\delta^{18}O_P$ to a simulated Heinrich event in GISS ModelE-R changes sign between inland China and the North Pacific, with a pattern very similar to our simulation of the 8.2 ka event, while in Pausata et al. (2011) the simulated response to a Heinrich event in CCSM3 includes a large enrichment signal over South and East Asia and no change over the North Atlantic. We have added a discussion comparing these model results to the text as follows:

*"Similarly large zonal asymmetry in the precipitation $\delta^{18}O$ response to meltwater forcing between China and the North Atlantic was identified in the Heinrich simulations of Lewis et al. (2010) and Pausata et al. (2011). In fact, the simulated pattern in precipitation $\delta^{18}O$ from iCESM under 8.2ka meltwater forcing is remarkably similar to the response in GISS ModelE-R under Heinrich forcing from Lewis et al. (2010), including in South and East Asia, NE Brazil, and the Caribbean/central America, indicating a robust model response in precipitation $\delta^{18}O$ to meltwater forcing. The only regions where this inter-model agreement breaks down is tropical/subtropical Africa and Antarctica."*

2. The data-model comparisons are necessarily quantitative rather than qualitative, particularly for speleothem δ18Op records. Otherwise, why use the isotope-enabled model? It would be beneficial to understand if changes in δ18Op are caused by the water isotope or precipitation seasonality.

Thank you very much for this suggestion. We have added a decomposition of the changes in amount-weighted precipitation $\delta^{18}O$ using the decomposition method performed in Liu and Battisti (2015) in order to assess whether the changes are due to changes in the monthly isotopic composition of precipitation or changes in the seasonality of precipitation (i.e., changes in the amount of monthly precipitation). To this end, we have added Figs. 11-14 to the manuscript, showing the decomposition in precipitation $\delta^{18}O$ for each of the target regions.

We have also added the following text to the Methods section:

*"We decomposed the changes in amount-weighted precipitation $\delta^{18}O$ following Liu and Battisti (2015) in order to assess whether the changes are due to changes in the monthly isotopic composition of precipitation or changes in the seasonality of precipitation (i.e., changes in the amount of monthly precipitation). Noting that the difference in amount-weighted $\delta^{18}O_P$ between the hosing and control simulation is:*

$$\delta 18O_{P,hose} - \delta 18O_{P,ctrl} = \frac{\sum_j \delta 18O_{j,hose} P_{j,hose}}{\sum_j P_{j,hose}} - \frac{\sum_j \delta 18O_{j,ctrl} P_{j,ctrl}}{\sum_j P_{j,ctrl}}, \ (1)$$

*the importance of changes in the seasonality of precipitation to the changes in $\delta^{18}O_P$ is then*

*given by:* $\frac{\sum_j \delta 18O_{j,ctrl} P_{j,hose}}{\sum_j P_{j,hose}} - \frac{\sum_j \delta 18O_{j,ctrl} P_{j,ctrl}}{\sum_j P_{j,ctrl}}, \ (2)$

*and the importance of changes in $\delta^{18}O$ of precipitation to the changes in total $\delta^{18}Op$ is given by:*

$$\frac{\sum_j \delta18O_{j,hose} P_{j,ctrl}}{\sum_j P_{j,ctrl}} - \frac{\sum_j \delta18O_{j,ctrl} P_{j,ctrl}}{\sum_j P_{j,ctrl}}. \quad (3)$$

*Note that Eqns (2) and (3) do not sum to the total changes in $\delta^{18}Op$ due to the nonlinearity in the definition of $\delta^{18}Op$."*

We also added the following text to the Discussion section:

*"In East Asia, the change in amount-weighted precipitation $\delta^{18}O$, including the east-west dipole pattern with isotopic depletion off the coast of China into the North Pacific and isotopic enrichment inland, is driven by the seasonal changes in precipitation $\delta^{18}O$. Under meltwater forcing, precipitation $\delta^{18}O$ inland is more enriched throughout the year, particularly in the dry season from December to April. While precipitation $\delta^{18}O$ off the coast is more depleted throughout the year, particularly during the wet season from June to November. Consistent with previous studies on Heinrich events, these results suggest that the meltwater-induced enrichment in Chinese speleothem $\delta^{18}O$ records is not driven by changes in local precipitation and/or the EASM strength, but rather driven by changes in moisture source, circulation, and/or upstream rainout (Chiang et al., 2020; Pausata et al., 2011, Lewis et al., 2010). That the largest changes in precipitation $\delta^{18}O$ over China occur during the winter season seems to align with the results from Lewis et al. (2010), indicating that increased moisture provenance in the Bay of Bengal during winter yields enriched precipitation $\delta^{18}O$ over China during Heinrich events.*

*In NE South America and central America, the change in amount-weighted precipitation $\delta^{18}O$ is also dominated by the seasonal changes in precipitation $\delta^{18}O$ and not the seasonality of precipitation, however the mechanisms of the response seem to differ from those in East Asia. In NE Brazil, precipitation increases under meltwater forcing and becomes more isotopically depleted during the wet season from December to July. These changes are consistent with a Type-1 control on precipitation $\delta^{18}O$ (Lewis et al., 2010), wherein the local amount effect dominates the precipitation $\delta^{18}O$ response. In central America, the change in amount-weighted precipitation $\delta^{18}O$ is characterized by a distinct SW-NE dipole with isotopic enrichment in the northern tropical Pacific and over Panama and isotopic depletion over the Caribbean and the remainder of central America. This pattern is also driven by the seasonal changes in precipitation $\delta^{18}O$ under meltwater forcing. In the northern tropical Pacific, wet season precipitation is substantially weakened and isotopically enriched, consistent with a Type-1 site (Lewis et al., 2010), wherein the local amount effect dominates the precipitation $\delta^{18}O$ response. Past studies on the response to Heinrich events have shown that regional precipitation changes in NE Brazil and eastern Pacific are associated with a southward shift of the Atlantic and eastern Pacific ITCZs (Lewis et al., 2010; Roberts et al., 2020; Atwood et al., 2020). The precipitation $\delta^{18}O$ response over the Caribbean and central America is distinct from any of the above sites. In this region, the wet season precipitation decreases under hosing, essentially eliminating the wet season, however precipitation becomes substantially more isotopically*

*depleted throughout the year. This response is expected in association with the large surface cooling of the tropical Atlantic Ocean and the addition of isotopically depleted meltwater into the North Atlantic. Thus, the precipitation δ¹⁸O response in this region would be classified as Type-5 according to the categorization of Lewis et al. (2010), with the mechanisms of precipitation δ¹⁸O governed by processes outside of the local or nonlocal amount effect, moisture source, or seasonality of precipitation."*

[Figure]

Figure 11. The contribution of (a) the changes in the seasonality of precipitation and (b) the monthly changes in $\delta^{18}O_p$ to the total change in mean annual amount-weighted $\delta^{18}O_p$ between the last 50 years of the "hose" and "ctrl" simulations. Stippling represents data plotted at the 95% confidence level ($p < 0.05$).

[Figure]

Figure 12. As in Fig. 11, but for East Asia. The unfilled black polygon represents the boundaries of the region defined by the IPCC.

[Figure]

Figure 13. As in Fig. 11, but for Northeast South America.

[Figure]

Figure 14. As in Fig. 11, but for Southern Central America.

**E Asia**

[Figure]

Figure 15. The area-weighted monthly average precipitation amount (top row) and precipitation δ¹⁸O (not amount-weighted; bottom row) for the "ctrl" (red) and "hose" (blue) simulations. Data from the western subregion defined by the +Δδ¹⁸Op anomaly in manuscript Fig. 4a are plotted in the left column. Data from the eastern subregion defined by the -Δδ¹⁸Op anomaly are plotted on the right.

[Figure]

Figure 16. As in Fig. 15, but averaged over all of Northeast South America.

[Figure]

Figure 17. As in Fig. 15, but for Southern Central America. Data from the southern subregion defined by the +Δδ$^{18}$Op anomaly in manuscript Fig. 6a are plotted in the left column, while data from the northern subregion defined by the -Δδ$^{18}$Op anomaly in Fig. 6a are plotted in the right column.

Minor:

Line 6: "event detention methods" should be "event detection methods"
Thank you for finding this. It has been corrected.

Line11: "decadal to multi-centennial timescales" should be: "decadal-to-multi-centennial timescales"
Thank you for the correction.

Line 33: "strong strong" should be "strong"
Thank you for finding this. It has been corrected.

Line 253: add "A total of 61"
The final compilation of records used in our analysis of the amplitude, timing, and duration of the 8.2ka event includes only 30 records. As stated in Lines 234-239:

*"The approximate start, end, and duration of hydroclimate anomalies associated with the 8.2ka event were calculated for all records in our compilation in which events of the same sign were detected in both our modified MM and actR event detection methods. This was done to provide a more robust reconstruction of the hydroclimate response to the 8.2ka Event than that which either method would achieve alone. **This final set of records comprises 30 of the 61 records (49%) in our compilation. The remaining 31 records in our compilation displayed a lack of agreement in the sign or presence of an event and are thus excluded from further analysis.**"*

To avoid confusion, we have reworded Lines 253-254 to state:

*"In the final set of 30 records (that agree on the sign of the event between the MM and ActR methods), drier and/or isotopically enriched events were detected in 13 of those 30 records, including six records from East Asia..."*

Fig. 3. The shading should be difference between control and hosing, right? The caption is confusing.
Thank you for bringing this to our attention. We have clarified this in the caption for Fig. 9 as follows:

*Summary of the 8.2ka Events detected using our actR method for the paleoclimate records included in this study. Records with drier and/or isotopically enriched events are shown in brown, records with wetter and/or isotopically depleted events are shown in green, and records in which no event was detected are shown in white. Stippling indicates that a "significant" event was detected in a given record by actR with event "start" and "end" times within the 7.9-8.3ka interval at the $p < 0.05$ significance level. Slashed hatching indicates the presence of a "tentative" hydroclimate anomaly, with either a "significant" event detected outside of the 7.9-8.3ka window (between 7.7-8.5ka) or an event within that window where $0.1 > p \geq 0.05$. Symbol size is scaled by $250\ln(1+|z|)$, calculated from the per-record mean and standard deviation over the 7ka-10ka interval. **Symbols are mapped over the simulated anomalous (a) amount-weighted oxygen isotopic composition of precipitation, (b) precipitation amount, (c) and effective moisture, calculated from difference between the last 50 years of the iCESM "hose" and "ctrl" experiments**, at the 95% confidence level ($p < 0.05$).*

Fig 5. No color bar
Thank you for finding this. It has been corrected (see Fig. 10 above).

**References**

Atwood, A. R., Donohoe, A., Battisti, D. S., Liu, X., and Pausata, F. S. R.: Robust Longitudinally Variable Responses of the ITCZ to a Myriad of Climate Forcings, Geophysical Research Letters, 47, e2020GL088833, https://doi.org/10.1029/2020GL088833, 2020.

Chiang, J. C. H., Herman, M. J., Yoshimura, K., and Fung, I. Y.: Enriched East Asian oxygen isotope of precipitation indicates reduced summer seasonality in regional climate and westerlies, Proc. Natl. Acad. Sci. U.S.A., 117, 14745–14750, https://doi.org/10.1073/pnas.1922602117, 2020.

Lewis, S. C., LeGrande, A. N., Kelley, M., and Schmidt, G. A.: Water vapour source impacts on oxygen isotope variability in tropical precipitation during Heinrich events, Clim. Past, 6, 325–343, https://doi.org/10.5194/cp-6-325-2010, 2010.

Pausata, F. S. R., Battisti, D. S., Nisancioglu, K. H., and Bitz, C. M.: Chinese stalagmite $\delta 18O$ controlled by changes in the Indian monsoon during a simulated Heinrich event, Nature Geosci, 4, 474–480, https://doi.org/10.1038/ngeo1169, 2011.

Roberts, W. H. G. and Hopcroft, P. O.: Controls on the Tropical Response to Abrupt Climate Changes, Geophys. Res. Lett., 47, https://doi.org/10.1029/2020GL087518, 2020.

---

## Author Response (AR1)

**Reply to Reviewers for egusphere-2024-3483**

**Summary of changes:**

We would like to thank the reviewers for their supportive and constructive feedback on our manuscript. We have substantially revised the paper to address their concerns, specifically in clarifying our methods, adding a discussion of potential hydroclimate mechanisms driving simulated precipitation isotope anomalies, and refining figures to better highlight our results. Detailed point-by-point responses to each comment follow below, with each reviewer's comment in standard typeface, our responses in **bold blue**, and quotes from the manuscript indented and *italicized*. Please note that we shifted from using the provided LaTeX template to the provided Word template during the revision process to facilitate collaborative editing, and while every effort has been made to retain the consistency of line numbering between both formats, there may be some disagreement. Nevertheless, the line numbering referenced in our responses refers to our current revision. We thank you again for your time and expertise.

Sincerely,
Andrea L. Moore, Alyssa R. Atwood, and Raquel E. Pauly
* * *
**RC1: 'Comment on egusphere-2024-3483', Anonymous Referee #1, 12 Dec 2024**

**RC1.1** In this paper, Moore and colleagues have thoroughly analyzed the signal of the Holocene 8.2ka event in hydroclimatic proxy records from the tropics. The proxy-based signal is compared to simulation results obtained with the isotope-enabled iCESM global climate model. In the compilation, proxy records from different environments are considered: marine sediment records, lake sediment records and speleothems. As such this work extends the compilation recently published by Parker and Harrison (2022), who focused on the 8.2ka signal in speleothem records. Innovative aspects of this manuscript are the extensive analysis of the impact of age uncertainties and the comparison of the proxy-based hydroclimatic anomaly with a state-of-the-art isotope-enabled model simulation of the 8.2ka event. The paper is well-written, with a clear structure, and includes high-quality figures. The main result is a convincing proxy-based overview of the hydroclimatic response to the 8.2ka event in the tropics, showing clear regional variability. In my view, the manuscript requires some minor revisions, mostly associated with the detection methods used and the climate model experiment. I provide details below.
**Thank you for the kind feedback.**

**RC1.2** Introduction, Line 59. "This period occurred during the otherwise stable Holocene epoch (11,700 years ago to present) and was driven by the discharge of around 163,000 km³ of meltwater from proglacial Lakes Ojibway and Agassiz (remnants of the Laurentide Ice Sheet) into the North Atlantic". The volume and source of the freshwater perturbation are actually still under discussion. For instance, the volume of the proglacial lakes is under discussion (Törnqvist and Hijma, 2012). In addition, Gregoire et al. (2012) argue that a collapse of the "ice saddle" over Hudson Bay could have played an important role in forcing the 8.2ka event. It is important to make the reader aware of this uncertainty. See also the recent discussion by Aguiar et al. (2021). Besides, it is confusing to refer to Lakes Ojibway and Agassiz as "remnants of the Laurentide Ice Sheet", so I suggest removing this part of the sentence.
**Thank you for this comment. We have changed the wording to better reflect this nuance as follows (lines 67-75):**
> *This event occurred during the otherwise stable Holocene epoch (11,700 years ago to present) and is thought to have been driven by the discharge of ~1.63×105 km³ of meltwater from proglacial Lakes Ojibway and Agassiz into the North Atlantic, triggering a large-scale salinity anomaly and resultant reduction in the strength of the Atlantic Meridional Overturning Circulation (AMOC; e.g., Barber et al., 1999; Ellison et al., 2006). The precise source, routing, and strength of the freshwater perturbation are still under discussion (e.g. Törnqvist and Hijma, 2012), ranging from an upper limit of 27.1×105 km³ of freshwater released from the retreating Laurentide Ice Sheet (LIS) between 9 ka and 8 ka (Peltier 2004), to a smaller but more abrupt discharge of 5.3×105 km³ between 8.31 ka and 8.18 ka (Li et al., 2012). Recent data-model comparisons from Aguiar et al. (2021) suggest that an additional 8.2×105 km³ of freshwater may have flowed into the Labrador Sea after the collapse of the Hudson Bay due to the routing of river discharge over the western Canadian Plains (Carlson et al., 2009).*

**RC1.3a** Section 2.3.1. MM method. I propose including a schematic figure explaining how the MM method of detection works in practice, using two examples: one example with multiple events of the same sign and one example with

multiple events of different signs. I am thinking of an extended version of Morrill's Figure 2. It is not clear to me what the actual modification is to the Morrill method. Please specify.

**Thank you for this suggestion. We have included a schematic illustrating our method as Fig. B1 in our revision (pg. 56). Our MM method differs from the method presented in Morrill et al., (2013) in several ways as it serves primarily as a supplement to our actR results. We have updated lines 166-177 to clarify how our MM method differs:**

> *For an excursion to be considered part of the 8.2 ka Event, the excursions must last at least 10 years. If multiple events are detected within the 7.9 ka-8.5 ka window, they are combined into a single event if there are no more than three data points, or thirty years, separating the different excursions. This modification is necessary to account for the varying sampling resolutions present within and between several of the records in our compilation. If multiple events of differing signs are detected within the 8.2 ka Event window, the event with the largest z-score is chosen as the representative hydroclimate response. The magnitude of the event is defined by the largest absolute value z-score within the event detection period.*

> *The MM method differs from the methodology presented in Morrill et al. (2013) in two additional aspects: (i) we do not perform the "leave one out" standard deviation calculation that Morrill et al. (2013) employed to account for noisy data and outliers in each reference window, and (ii), while Morrill et al. (2013) use a moving two-tailed z-test to define the duration of their detected events, we consider only the time between the initial and final anomalous data point in our calculation. We elected to simplify this method as it is primarily intended to supplement results using our actR methodology.*

**RC1.3b** If feasible, including such a schematic for the actR method would also be helpful for readers not familiar with this type of changepoint analysis.

**Thank you for this suggestion. We have included a schematic illustrating the actR method as Fig. 2 in our revision (pg. 33).**

**RC1.4** Section 2.4. Please make clear that the experimental setup for the iCESM simulation of the 8.2ka event does only partly follow the PMIP4 design reported by Otto-Bliesner et al. (2017). In the PMIP4 design, the 8.2ka simulation starts from an experiment with fixed forcing for 9.5 ka instead of 9 ka as is done here. In addition, in the PMIP4 design the freshwater hosing is applied in the Labrador Sea instead of across the entire northern North Atlantic.

**We agree that it is important to specify how our experimental setup otherwise differs. We have updated the wording of that section to clarify that we followed the PMIP4 protocol for the timing and duration of the meltwater events (2.5 Sv for 1 year, followed by 0.13 Sv for 99 years), while the location of the meltwater forcing is notably different. We have also clarified that our hosing experiment branches from 9 ka boundary conditions (instead of 9.5 ka), and thus uses slightly different orbital and GHG configurations. From lines 232-239:**

> *These simulations followed the Paleoclimate Modeling Intercomparison Project 4-Coupled Model Intercomparison Project 6 (PMIP4-CMIP6) 8.2 ka simulation parameters (Otto-Bliesner et al., 2017), with two exceptions: (1) the freshwater flux was applied across the entire northern North Atlantic in our simulations (instead of just in the Labrador Sea as in PMIP4) in order to limit the sensitivity of the subsequent AMOC and climate response to poorly resolved deepwater formation regions in the model, and (2) our hosing experiment branches from 9 ka boundary conditions (instead of 9.5 ka as in PMIP4), and thus uses slightly different orbital and GHG configurations from PMIP4. However, the impact of these marginally different boundary conditions is expected to be minimal.*

**RC1.5** Discussion. In some regions, for example the Caribbean and SE Asia, there are records without any detected change located close to records with a clear signal for the 8.2ka event (see for instance Figure 2). How to interpret the absence of a signal in these records without significant change? It would be insightful to discuss the possible reasons for the absence of a signal.

**You raise an important point that we have addressed more explicitly in our revision by adding the text below (lines 337-346):**

> *In several regions (including East Asia, Fig. 5; and northeastern South America, Fig. 6), records with no detected change are located near records with clear event signals. These regional differences could arise from several factors, including localized hydroclimate responses to the event, age uncertainty, and proxy interpretation uncertainties. For example, speleothem $\delta^{18}O$ records have been interpreted as representing a range of different climate processes, often within the same region, including changes in regional precipitation amount, monsoon strength, moisture source location, upstream rainout, seasonal frontal shifts, and temperature (e.g. Hu et al., 2019), reflecting the complexity of processes that impact $\delta^{18}O_p$ and speleothem $\delta^{18}O$. Because of the inherently regional nature of rainfall patterns and the uncertainties in the proxy records, we focus our interpretation on regional*

*hydroclimate signals that are supported by multiple records, often across different aspects of hydroclimate. In this way, we focus on the most robust aspects of the tropical hydroclimate response to the 8.2 ka Event.*

**RC1.6** Section 4.2. Simulated 8.2ka event. The results of the last 50 years of the "hose" experiment are taken to represent the modelled 8.2ka event. The mapped difference between "hose" and "ctrl" is used, for instance in Figure A2 and in the model-data comparison. It would be informative to know how the climate evolved through time in "hose" relative to "ctrl" (for instance related to detectability in the model result, see next point), so I propose including additional figures with modelled time-series of hydroclimate for the key regions shown in Figures 4 to 7. These additional figures could show 100-year time-series of both "ctrl" and "hose".

**Thank you for the suggestion. We have added Figures 11-13 to the revised manuscript to show the monthly changes in simulated precipitation amount and precipitation δ¹⁸O between the "hose" and "ctrl" runs for East Asia, northeast South America, and southern Central America (the three regions with the most robust responses and data-model agreement). We will consider including the following time series of precipitation δ¹⁸O for these regions as supplemental figures (see Figs. 1-3 below).**

[Figure]

**Figure 1.** The amount-weighted, area-weighted mean annual precipitation δ¹⁸O for the "ctrl" (red) and "hose" (blue) simulations over all of East Asia (manuscript Fig. 5; top panel), the western subregion defined by the +Δδ¹⁸O$_p$ anomaly in manuscript Fig. 5a , and the eastern subregion defined by the -Δδ¹⁸O$_p$ anomaly in Fig. 5a.

[Figure]

**Figure 2.** As in Fig. 1, but for NE South America (manuscript Fig. 6).

[Figure]

[Figure]

[Figure]

**Figure 3.** As in Fig. 1, but averaged over all of Southern Central America (top panel), the southern subregion defined by the $+\Delta\delta^{18}O_p$ anomaly in manuscript Fig. 7a , and the northern subregion defined by the $-\Delta\delta^{18}O_p$ anomaly in Fig. 7a.

**RC1.7** Detectability of the simulated 8.2ka event. The question of detectability of the 8.2ka event is not only relevant for proxy records, but also for the model results. The difference between "hose" and "ctrl" includes anomalies produced by both internal variability and forced variability (i.e. by hosing). Ideally, one should not perform just one simulation for the 8.2ka event as is done here, but rather an ensemble experiment in which the members differ only in initial conditions and have identical freshwater forcing (hosing). The ensemble mean allows then to analyze the forced response and to separate it from the internal variability. I refer to Wiersma et al. (2011), who performed this analysis for the temperature response associated with the 8.2ka event. I realize that it is probably not feasible within the framework of this study to perform several additional ensemble members with iCESM, but in my view the aspect of detectability in the model results should also be discussed in the paper. In addition, I suggest performing a statistical test to determine what results of "hose" are significantly different from the "ctrl" climate, and to include only those significantly different results in the model-data comparison.

**Thank you for this comment. We agree that it is important to highlight the statistically significant changes in the model simulations. While running additional ensemble members to separate the forced response from internal variability is outside the scope of this study, to address the issue of significance, we have performed an unpaired two sample Student's t-test on the precipitation, effective moisture, and amount-weighted precipitation $\delta^{18}O$ fields from the hosed and control model runs. In our revised figures showing the changes in precipitation and precipitation $\delta^{18}O$ under hosing (e.g., revised Figs. 4-8), we now only plot those changes that exceed the 95% confidence level (p < 0.05).**

**RC1.8** Line 87. Why is it a critical tool? Please elaborate.

**This sentence was removed and the paragraph was modified to more explicitly indicate how model data can complement proxy data to provide a more comprehensive understanding of the response of the climate system to the 8.2 ka Event (lines 110-115):**

*We further assess how well the proxy reconstructions compare to a new isotope-enabled model simulation of the 8.2 ka Event. Such model simulations provide dynamical context to the sparse proxy data and, by tracking water isotopes through the hydrologic cycle, enable more direct comparisons between proxy and model data than conventional climate models. Such data-model comparisons facilitate improved understanding of the tropical hydroclimate response to abrupt AMOC disruptions and provide a necessary benchmark for climate models that are used in projections of future climate change.*

**RC1.9** Line 104. How is this sensitivity to hydroclimate determined?

**The passage in question (lines 118- 120):**

*To assess the tropical hydroclimate response to the 8.2 ka Event, we developed an updated compilation of published, high-resolution, continuous, well-dated proxy datasets, collating records that span the period 7ka-10ka, cover latitudes from 30°N to 30°S, and which are sensitive to some aspect of hydroclimate variability.*

**We address this in point (ii) of the subsequent paragraph (lines 125-131):**

*To constrain the timing and duration of the abrupt hydroclimate anomaly associated with the 8.2 ka Event, the datasets in this compilation were screened to meet the following criteria: (i) data resolution of 50 years or better over the period of 7 ka-10 ka;* **(ii) based on hydroclimate-sensitive proxy data interpreted by authors as reflecting precipitation amount or intensity, the isotopic compositions of environmental water (including precipitation, lake water, and seawater), effective moisture, lake level, fluvial discharge, or sea surface salinity (SSS)***; and (iii) contain at least three radiometric dates over the 7 ka-10 ka interval. Emphasis was placed on collecting water isotope-based records to enable more direct comparison with isotope-enabled climate model simulations.*

**RC1.10** Line 151. Why 10 years? On what is this based?

**The actR parameter in question (summary.bin.step) controls how the shift detection information is summarized in the output, and has no bearing on the changepoint calculation itself. Because the majority of our records are decadally resolved, and because we wanted to constrain the timing of significant shifts as precisely as possible, this value was chosen for our data processing.**

**RC1.11** Line 166. Why 50 or 100 years? Why not taking just 50 years?

**Thank you for this question. Our parameter choices were based on the following considerations: The large dataset required parameters that would effectively accommodate most time series in our compilation while minimizing spurious shift detections. For all but one record in our compilation, the 100-year minimum segment length optimally**

captured abrupt changes without detecting broader climatic trends or less significant (spurious) shifts. For the PAD07 speleothem record of Cheng et al. (2009; Fig. C5 in our revised manuscript), it was necessary to reduce the minimum segment length to 50 years to capture the clear isotopic depletion near 8.2 ka that was otherwise smoothed over when employing our longer 100-year minimum segment length (see Fig. 4 below). This was the only record in our compilation that necessitated such an adjustment, and we have clarified this in the manuscript (lines 186-192):

> *A minimum segment length of 50 or 100 years was assigned for each record in the proxy compilation to minimize short-lived transitions in the noisy proxy records, with the assumption that the 8.2 ka Event signal in each of the records lasts at least 50 years. For all but one record in our compilation, the 100-year minimum segment length optimally captured the major shifts in the data sets while minimizing the detection of spurious short-lived shifts. The exception was the speleothem record of Cheng et al. (2009; PAD07; Fig. C5), for which it was necessary to reduce the minimum segment length to 50 years to capture the clear isotopic depletion near 8.2 ka that was otherwise missed.*

A comprehensive sensitivity analysis of this parameter would be valuable to systematically determine the optimal minimum segment lengths across different types of paleoclimate records. However, this was outside the scope of our current study.

[Figure]

**Figure 4.** A comparison of shifts in mean (red horizontal lines, thickened for visibility) detected in the ensemble time series for PAD07 (Cheng et al., 2009) using a minimum segment length parameter of 100 (top panel) and 50 (bottom panel).

**RC1.12a** Line 183. In the actR method, the time window for detection (7.9 to 8.3 ka) is different from the window used in the MM method (7.9 to 8.5 ka). Why this difference and what is the consequence?

**The analysis window used in the MM method is the same as the window defined in Morrill et al. (2013), maintained here for some degree of parity with their original analysis, which they chose "to bracket the [8.2 ka] event, while accommodating errors in the age models of several hundred years". The tighter ("significant") window that we employed in our actR method (7.9 ka-8.3 ka) is based on the duration of the 8.2 ka Event in Greenland ice core data within age uncertainties (8.25 ka to 8.09 ka; Thomas et al., 2007). Additionally, we define a second, broader actR window from 7.7 ka-8.5 ka to further account for age uncertainty across which we define our "tentative" events. The**

**actR method allows us to leverage new, robust age modeling tools and the propagation of age uncertainties through our changepoint analyses to better estimate the timing and duration of abrupt events, like the 8.2 ka Event, while the MM method adds supporting evidence to our interpretation of the sign and magnitude of the detected changes.**

**RC1.12b** In addition, is there also a minimum duration for detection in the actR method?
**To your second question, the minimum event duration length using the actR method is 20 years. We have clarified this in our Methods section in our revision, and we thank you for bringing this omission to our attention. Lines 212-214:**

> *The difference between "start" and "end" dates is used to calculate event duration, which we assume to be between a minimum of 20 and a maximum of 300 years.*

**RC1.13** Line 212. Are the forcings for the "ctrl" simulation identical to the 400-year-long 9ka simulation? If not, what are the differences? And is the climate stable in "ctrl" or is there still a trend in the surface conditions, signifying adjustment to the different forcings?
**Yes, the preindustrial control simulation is branched from the 9 ka simulation and all forcings and boundary conditions are identical to the 400-yr 9 ka simulation. Quasi-equilibrium has indeed been reached in the last 100 years of the control run (analyzed in this study), based on the lack of a trend in the global average surface air temperature (see Fig. 5 below).**

[Figure]

**Figure 5.** Time series of simulated global mean surface air temperature (SAT) for the last 100 years of the control run (red) and all 100 years of the hosed run (blue).

**RC1.14** 4.3 Data-Model comparisons. It is noteworthy that the regional structure of the 8.2ka event found in this study resembles the hydroclimatic anomaly in a simulation of the Younger Dryas cold event in which freshwater forcing applied in the North Atlantic also plays an important role (Renssen et al. 2018). The hydroclimatic response for the Younger Dryas shows wetter conditions in the Caribbean, SE South America, Southern Africa and Madagascar, but drier conditions in S Central America, the Arabian Peninsula and SE Asia, broadly consistent with the proxy-based signal of the 8.2ka event provided in the present paper in Figure 2.
**Thank you for pointing this out. There are indeed some interesting similarities between these two data sets and thus, we have added the following sentence to our Results in 3.5 Data-Model Comparisons (lines 461-462):**

> *Similar hydroclimate features also appear in simulations of the Younger Dryas cold event from Renssen et al (2018).*

**However, we don't emphasize the comparisons between our proxy compilation and this study beyond this for several reasons: (1) the Younger Dryas cold event had differing boundary conditions and a much larger amplitude than the 8.2ka event, (2) the majority of the proxy records in Fig. 2 (now Fig. 3 in our revision) are speleothem δ¹⁸O records, a proxy for precipitation δ¹⁸O, while the modeling results of Renssen et al. (2018) show the change in soil moisture, and**

**(3) the fact that the most robust signature in our proxy records is the drying/isotopic enrichment event in East Asia, which does not agree with the simulated increase in soil moisture in that region in Renssen et al. (2018).**

**RC1.15** Line 238. "… a lack of agreement of the sign or presence of an event". Do you mean a lack of agreement between the two detection methods? Please clarify.
**Yes, thank you for the suggestion. The sentence has been updated to make this explicit (lines 286-287):**
> *The remaining 31 records displayed disagreement between the two detection methods and were thus excluded from further analysis.*

**RC1.16** Figures 2-7. The grey symbols are very hard to see, so I suggest improving their visibility.
**Thank you for this suggestion. Because the size of the symbols scales with the magnitude of the z-scores to emphasize the most prominent signals in the records, the grey symbols (indicating no change) are typically small. However, we agree that it is important to identify these records, and so we have added black outlines to the grey symbols in our revised figures, which we believe has made them more visible (e.g. see Fig. 6 in the revised manuscript).**

**RC1.17** Figures 3-7. What do the grey isolines represent? Please explain.
**The grey isolines represent the average oxygen isotopic composition of precipitation from the control simulation, however, they are extraneous and so have been removed from the revised figures (Figs. 4-8 and B4-B8).**

**RC1.18** Figures 4-7. In several cases, there are no proxy records of a specific type in a region (for example marine records in East Asia, Figures 4c and f). It is not very meaningful to show the simulation results in these cases, as there is no data-model comparison possible. So, I propose to only show the maps of key regions if there are proxy records available for the data-model comparison.
**Thank you for the suggestion. We agree that the panels lacking proxy data are unnecessary. These extraneous panels have been removed from the revised figures (Figs. 5-8 and B5-B8).**

**RC1.19** Figure 5. Legend is missing.
**Thank you for finding this. We have corrected it in our revision (Fig. 6).**

**RC1.20** Figure A2a and b. According to the figure caption, the contours in these figures indicate the "range of temperatures in the "ctrl" simulation over the full 100 years". It is not clear what temperatures these isotherms represent, so I do not see how these contours are useful, and I suggest removing them from the figure.
**Thank you for the suggestion. The unfilled contours representing the isotherms in the control simulation have been removed.**
* * *
**RC2: 'Comment on egusphere-2024-3483', Anonymous Referee #2, 17 Jan 2025**

**RC2.1** The study by Moore et al. compiled global hydroclimate proxies for the 8.2ka event in the tropics and subtropics and compared them with the hosing simulation by the state-of-the-art isotope enabled CESM.

Major novelties:
1. They developed an updated compilation of high-resolution, continuous, well-dated proxy datasets. This is important to the broad paleoclimate community.
2. They introduced the use of the Abrupt Change Toolkit in R (actR) for event detection, which better accounts for age model uncertainties in proxy records. As a result, they quantified the starting, ending, and duration of the 8.2ka event.
3. They revealed a more complex, regionally specific hydroclimate response pattern rather than a simple hemispheric dipole in the 8.2ka event.

However, the paper lacks depth in discussing the source of model-data differences, regional hydroclimate mechanisms, and the responses and of $\delta^{18}O_p$, making it feel dry due to excessive qualitative descriptions of proxy and modeling results. In my opinion, it could be published in climate of the past, but a list of concerns should be addressed.

We appreciate your feedback. While a detailed analysis of the mechanisms of the hydroclimate changes in the model is outside the scope of this study, we have performed a decomposition of the isotopic signals in the model and included further discussion of the regional hydroclimate changes in the model, the possible underlying mechanisms, and how they compare with other climate model simulations and proxy data, as detailed below. We hope these additions add depth to our analysis and discussion.

Major concerns:

**RC2.2a** Condense the proxy and model results description while enhancing analysis of model-data differences. Focus on presenting actR results primarily, with MM results moved to supplementary materials for greater conciseness.
**Thank you for this helpful suggestion. We have moved many of the MM results to the Appendices (e.g., Figs. B4-B8) for conciseness.**

**RC2.2b** This paper's unique contribution lies in the detailed regional data-model comparison, but thoughtful discussions lack. For instance, the $\delta^{18}O_p$ responses in East Asia during the 8.2ka event exhibit an east-west dipole pattern, contrasting with the uniform enriched isotopic signal seen in the Heinrich events. Despite similar hosing experiments, it is intriguing to explore why such discrepancies exist between these events.
**We have significantly enhanced the discussion of the regional $\delta^{18}O_p$ responses in the model and elaborated on comparing these features to the proxy records. Regarding the east-west dipole pattern in East Asia, we now point out two other modeling studies that have demonstrated a large zonal asymmetry in the $\delta^{18}O_p$ response in this region under meltwater forcing: Lewis et al. (2010) and Pausata et al. (2011). In Lewis et al. (2010), the response of $\delta^{18}O_p$ to a simulated Heinrich event in GISS ModelE-R changes sign between inland China and the North Pacific, with a pattern very similar to our simulation of the 8.2 ka event, while in Pausata et al. (2011) the simulated response to a Heinrich event in CCSM3 includes a large enrichment signal over South and East Asia and no change over the North Atlantic.**

**RC2.3** The data-model comparisons are necessarily quantitative rather than qualitative, particularly for speleothem $\delta^{18}O_p$ records. Otherwise, why use the isotope-enabled model? It would be beneficial to understand if changes in $\delta^{18}O_p$ are caused by the water isotope or precipitation seasonality.
**Thank you very much for this suggestion. We have added a decomposition of the changes in amount-weighted precipitation $\delta^{18}O$ using the decomposition method performed in Liu and Battisti (2015) in order to assess whether the changes are due to changes in the monthly isotopic composition of precipitation or changes in the seasonality of precipitation (i.e., changes in the amount of monthly precipitation). To this end, we have added Figs. 9-13 to the manuscript, showing the decomposition in precipitation $\delta^{18}O$ for each of the target regions.**
**We have also added the following text to the Methods section (2.5 Decomposition of changes in precipitation $\delta^{18}O$) :**

*We decomposed the changes in amount-weighted $\delta^{18}O_p$ following Liu and Battisti (2015) to assess whether the changes arise from variations in the monthly isotopic composition of precipitation or changes in the seasonality of precipitation (i.e., changes in monthly precipitation amount). The difference in amount-weighted $\delta^{18}O_p$ between the hosing and control simulations is:*

$$\delta^{18}O_{p,hose} - \delta^{18}O_{p,ctrl} = \frac{\sum_j \delta^{18}O_{j,hose} P_{j,hose}}{\sum_j P_{j,hose}} - \frac{\sum_j \delta^{18}O_{j,ctrl} P_{j,ctrl}}{\sum_j P_{j,ctrl}} \qquad (1)$$

*where $\delta^{18}O_j$ is the monthly isotopic composition of precipitation and $P_j$ is the monthly precipitation rate (in mm day$^{-1}$). The importance of changes in precipitation seasonality to changes in $\delta^{18}O_p$ is then given by:*

$$\frac{\sum_j \delta^{18}O_{j,ctrl} P_{j,hose}}{\sum_j P_{j,hose}} - \frac{\sum_j \delta^{18}O_{j,ctrl} P_{j,ctrl}}{\sum_j P_{j,ctrl}} \qquad (2)$$

*and the importance of changes in the monthly isotopic composition of precipitation to changes in total $\delta^{18}O_p$ is given by:*

$$\frac{\sum_j \delta^{18}O_{j,hose} P_{j,ctrl}}{\sum_j P_{j,ctrl}} - \frac{\sum_j \delta^{18}O_{j,ctrl} P_{j,ctrl}}{\sum_j P_{j,ctrl}} \qquad (3)$$

*Note that Eqs. (2) and (3) do not sum to the total change in $\delta^{18}O_p$ due to nonlinearity in the definition of $\delta^{18}O_p$.*

**We also added the following text to the Discussion section (3.4.1 Mechanisms driving the response of precipitation δ¹⁸O to North Atlantic freshwater forcing):**

> *To assess whether the simulated hydroclimate changes are due to changes in the seasonality of $\delta^{18}O_p$ or changes in the seasonality of precipitation amount, we decomposed the changes in amount-weighted $\delta^{18}O_p$ following Liu and Battisti (2015; Fig. 9). In East Asia, the change in amount-weighted $\delta^{18}O_p$, including the east-west dipole pattern with isotopic depletion off the coast of China into the North Pacific and isotopic enrichment inland, is driven by the seasonal changes in the isotopic composition of precipitation (Fig. 10b,c). Under meltwater forcing, $\delta^{18}O_p$ inland is more enriched throughout the year, particularly in the dry season from December to April (Fig. 12c). While $\delta^{18}O_p$ off the coast is more depleted throughout the year, particularly during the wet season from June to November (Fig. 12d). Consistent with previous studies on Heinrich events, these results suggest that the meltwater-induced enrichment in Chinese speleothem $\delta^{18}O$ records is not driven by changes in local precipitation and/or the strength of the EASM, but rather driven by changes in moisture source, circulation, and/or upstream rainout (Chiang et al., 2020; Pausata et al., 2011, Lewis et al., 2010). That the largest changes in $\delta^{18}O_p$ over China occur during the winter season is consistent with the results from Lewis et al. (2010), which found that increased moisture provenance in the Bay of Bengal during winter yielded enriched $\delta^{18}O_p$ over China during Heinrich events. The large zonal asymmetry observed in the $\delta^{18}O_p$ response to meltwater forcing between China and the North Atlantic was also identified in the Heinrich simulations of Lewis et al. (2010) and Pausata et al. (2011).*

> *In northeastern South America and southern Central America, the change in amount-weighted $\delta^{18}O_p$ is also dominated by the seasonal changes in $\delta^{18}O_p$ and not the seasonality of precipitation (Fig. 10e-f,h-i), however the mechanisms of the response seem to differ from those in East Asia. In northeastern Brazil, precipitation increases under meltwater forcing and becomes more isotopically depleted during the wet season from December to July (Fig. 11c,d). These changes are consistent with a Type-1 control on $\delta^{18}O_p$ (Lewis et al., 2010), wherein the local amount effect dominates the $\delta^{18}O_p$ response. In southern Central America, the change in amount-weighted $\delta^{18}O_p$ is characterized by a distinct SW-NE dipole with isotopic enrichment in the northeastern tropical Pacific and over Panama and isotopic depletion over the Caribbean and the remainder of southern Central America. This pattern is also driven by the seasonal changes in $\delta^{18}O_p$ under meltwater forcing (Fig. 10h,i). In the northeastern tropical Pacific, wet season precipitation is substantially weakened and isotopically enriched (Fig. 13a,c), consistent with a Type-1 site (Lewis et al., 2010), wherein the local amount effect dominates the $\delta^{18}O_p$ response. Past studies on the hydroclimate response to Heinrich events have shown that regional precipitation changes in northeastern Brazil and the eastern Pacific are associated with a southward shift of the Atlantic and northeastern tropical Pacific ITCZs (Lewis et al., 2010; Roberts et al., 2020; Atwood et al., 2020). However, the $\delta^{18}O_p$ response over the Caribbean and southern Central America is notably different. In this region, the wet season precipitation decreases under hosing, essentially eliminating the wet season, while the precipitation becomes substantially more isotopically depleted throughout the year (Fig. 13b,d), in association with the strong surface cooling of the tropical Atlantic Ocean and the addition of isotopically depleted meltwater into the North Atlantic. Thus, the $\delta^{18}O_p$ response in this region would be classified as Type-5 according to the categorization of Lewis et al. (2010), with the mechanisms driving the $\delta^{18}O_p$ response governed by processes outside of the local or nonlocal amount effect, moisture source, or seasonality of precipitation.*

Minor:

**RC2.4** Line 6:  "event detention methods" should be "event detection methods"
**Thank you for finding this. It has been corrected (line 11).**

**RC2.5** Line11: "decadal to multi-centennial timescales" should be: "decadal-to-multi-centennial timescales"
**This phrase has been removed from the revised abstract.**

**RC2.6** Line 33: "strong strong" should be "strong"
**Thank you for finding this. It has been corrected (line 42).**

**RC2.7** Line 253: add "A total of 61"
**The final compilation of records used in our analysis of the amplitude, timing, and duration of the 8.2ka event includes only 30 records. As stated in lines 234-239 of the original submission:**

> *The approximate start, end, and duration of hydroclimate anomalies associated with the 8.2ka event were calculated for all records in our compilation in which events of the same sign were detected in both our modified MM and actR event detection methods. This was done to provide a more robust reconstruction of the hydroclimate response to the 8.2ka Event than that which either method would achieve alone. **This final set of records***

*comprises 30 of the 61 records (49%) in our compilation. The remaining 31 records in our compilation displayed a lack of agreement in the sign or presence of an event and are thus excluded from further analysis.*

**To avoid confusion, we have reworded lines 301-302 to clarify:**

*In the final set of 30 records (that agree on the sign of the event between the MM and actR methods), drier and/or isotopically enriched events were detected in 13 of those 30 records[...]*

**RC2.8** Fig. 3. The shading should be difference between control and hosing, right? The caption is confusing.

**Thank you for bringing this to our attention. We have clarified this in the caption for our revised Fig. 3 (now Fig. 4) as follows:**

*[...]the proxy symbols overlaid on contour maps of the simulated anomalous (a) amount-weighted $\delta^{18}O_p$, (b) precipitation amount, and (c) effective moisture (P-E), calculated from the difference between the last 50 years of the iCESM "hose" and "ctrl" experiments, where only anomalies that exceed the 95% confidence level ($p < 0.05$) are plotted.*

**RC2.9** Fig 5. No color bar

**Thank you for finding this. It has been corrected (see Fig. 6 in our revision).**
* * *
**References**

Aguiar, W., Meissner, K. J., Montenegro, A., Prado, L., Wainer, I., Carlson, A. E., and Mata, M. M.: Magnitude of the 8.2 ka event freshwater forcing based on stable isotope modelling and comparison to future Greenland melting, Scientific Reports, 11, 5473, https://doi.org/10.1038/s41598-021-84709-5, 2021.

Atwood, A. R., Donohoe, A., Battisti, D. S., Liu, X., and Pausata, F. S. R.: Robust Longitudinally Variable Responses of the ITCZ to a Myriad of Climate Forcings, Geophysical Research Letters, 47, e2020GL088833, https://doi.org/10.1029/2020GL088833, 2020.

Barber, D. C., Dyke, A., Hillaire-Marcel, C., Jennings, A. E., Andrews, J. T., Kerwin, M. W., Bilodeau, G., McNeely, R., Southon, J., Morehead, M. D., and Gagnon, J.-M.: Forcing of the cold event of 8,200 years ago by catastrophic drainage of Laurentide lakes, Nature, 400, 344–348, https://doi.org/10.1038/22504, 1999.

Carlson, A. E., Clark, P. U., Haley, B. A., and Klinkhammer, G. P.: Routing of western Canadian Plains runoff during the 8.2 ka cold event, Geophysical Research Letters, 36, 2009GL038778, https://doi.org/10.1029/2009GL038778, 2009.

Cheng, H., Fleitmann, D., Edwards, R. L., Wang, X., Cruz, F. W., Auler, A. S., Mangini, A., Wang, Y., Kong, X., Burns, S. J., and Matter, A.: Timing and structure of the 8.2 kyr B.P. event inferred from δ18O records of stalagmites from China, Oman, and Brazil, Geology, 37, 1007–1010, https://doi.org/10.1130/G30126A.1, 2009.

Chiang, J. C. H., Herman, M. J., Yoshimura, K., and Fung, I. Y.: Enriched East Asian oxygen isotope of precipitation indicates reduced summer seasonality in regional climate and westerlies, Proc. Natl. Acad. Sci. U.S.A., 117, 14745–14750, https://doi.org/10.1073/pnas.1922602117, 2020.

Ellison, C. R. W., Chapman, M. R., and Hall, I. R.: Surface and Deep Ocean Interactions During the Cold Climate Event 8200 Years Ago, Science, 312, 1929–1932, https://doi.org/10.1126/science.1127213, 2006.

Lewis, S. C., LeGrande, A. N., Kelley, M., and Schmidt, G. A.: Water vapour source impacts on oxygen isotope variability in tropical precipitation during Heinrich events, Clim. Past, 6, 325–343, https://doi.org/10.5194/cp-6-325-2010, 2010.

Li, Y.-X., Törnqvist, T. E., Nevitt, J. M., and Kohl, B.: Synchronizing a sea-level jump, final Lake Agassiz drainage, and abrupt cooling 8200years ago, Earth and Planetary Science Letters, 315–316, 41–50, https://doi.org/10.1016/j.epsl.2011.05.034, 2012.

Morrill, C., Anderson, D. M., Bauer, B. A., Buckner, R., Gille, E. P., Gross, W. S., Hartman, M., and Shah, A.: Proxy benchmarks for intercomparison of 8.2 ka simulations, Clim. Past, 9, 423–432, https://doi.org/10.5194/cp-9-423-2013, 2013.

Pausata, F. S. R., Battisti, D. S., Nisancioglu, K. H., and Bitz, C. M.: Chinese stalagmite δ18O controlled by changes in the Indian monsoon during a simulated Heinrich event, Nature Geosci, 4, 474–480, https://doi.org/10.1038/ngeo1169, 2011.

Peltier, W. R.: Global glacial isostasy and the surface of the ice-age Earth: The ICE-5G (VM2) model and GRACE, Annu. Rev. Earth Planet. Sci., 32, 111–149, https://doi.org/10.1146/annurev.earth.32.082503.144359, 2004.

Renssen, H., Goosse, H., Roche, D. M., and Seppä, H.: The global hydroclimate response during the Younger Dryas event, Quaternary Science Reviews, 193, 84–97, https://doi.org/10.1016/j.quascirev.2018.05.033, 2018.

Roberts, W. H. G. and Hopcroft, P. O.: Controls on the Tropical Response to Abrupt Climate Changes, Geophys. Res. Lett., 47, https://doi.org/10.1029/2020GL087518, 2020.

Thomas, E. R., Wolff, E. W., Mulvaney, R., Steffensen, J. P., Johnsen, S. J., Arrowsmith, C., White, J. W. C., Vaughn, B., and Popp, T.: The 8.2ka event from Greenland ice cores, Quaternary Science Reviews, 26, 70–81, https://doi.org/10.1016/j.quascirev.2006.07.017, 2007.

Törnqvist, T. E. and Hijma, M. P.: Links between early Holocene ice-sheet decay, sea-level rise and abrupt climate change, Nature Geosci, 5, 601–606, https://doi.org/10.1038/ngeo1536, 2012.